# STRONG INDUCTIVE BIASES PROVABLY PREVENT HARMLESS INTERPOLATION

**Michael Aerni**[*1]**, Marco Milanta**[*1]**, Konstantin Donhauser**[1,2]**, Fanny Yang**[1]
[1]Department of Computer Science, ETH Zurich [2]ETH AI Center

## ABSTRACT

Classical wisdom suggests that estimators should avoid fitting noise to achieve good generalization. In contrast, modern overparameterized models can yield small test error despite interpolating noise — a phenomenon often called "benign overfitting" or "harmless interpolation". This paper argues that the degree to which interpolation is harmless hinges upon the strength of an estimator's inductive bias, i.e., how heavily the estimator favors solutions with a certain structure: while strong inductive biases prevent harmless interpolation, weak inductive biases can even require fitting noise to generalize well. Our main theoretical result establishes tight non-asymptotic bounds for high-dimensional kernel regression that reflect this phenomenon for convolutional kernels, where the filter size regulates the strength of the inductive bias. We further provide empirical evidence of the same behavior for deep neural networks with varying filter sizes and rotational invariance.

## 1 INTRODUCTION

According to classical wisdom (see, e.g., Hastie et al. (2001)), an estimator that fits noise suffers from "overfitting" and cannot generalize well. A typical solution is to prevent interpolation, that is, stopping the estimator from achieving zero training error and thereby fitting less noise. For example, one can use ridge regularization or early stopping for iterative algorithms to obtain a model that has training error close to the noise level. However, large overparameterized models such as neural networks seem to behave differently: even on noisy data, they may achieve optimal test performance at convergence after interpolating the training data (Nakkiran et al., 2021; Belkin et al., 2019a) — a phenomenon referred to as *harmless interpolation* (Muthukumar et al., 2020) or *benign overfitting* (Bartlett et al., 2020) and often discussed in the context of *double descent* (Belkin et al., 2019a).

To date, we lack a general understanding of when interpolation is harmless for overparameterized models. In this paper, we argue that the *strength of an inductive bias* critically influences whether an estimator exhibits harmless interpolation. An estimator with a strong inductive bias heavily favors "simple" solutions that structurally align with the ground truth (such as sparsity or rotational invariance). Based on well-established high-probability recovery results of sparse linear regression (Tibshirani, 1996; Candes, 2008; Donoho & Elad, 2006), we expect that models with a stronger inductive bias generalize better than ones with a weaker inductive bias, particularly from noiseless data. In contrast, the effects of inductive bias are much less studied for interpolators of noisy data.

Recently, Donhauser et al. (2022) provided a first rigorous analysis of the effects of inductive bias strength on the generalization performance of linear max-$\ell_p$-margin/min-$\ell_p$-norm interpolators. In particular, the authors prove that a stronger inductive bias (small $p \to 1$) not solely enhances a model's ability to generalize on noiseless data, but also increases a model's sensitivity to noise — eventually harming generalization when interpolating noisy data. As a consequence, their result suggests that interpolation might not be harmless when the inductive bias is too strong.

In this paper, we confirm the hypothesis and show that strong inductive biases indeed prevent harmless interpolation, while also moving away from sparse linear models. As one example, we consider data where the true labels nonlinearly only depend on input features in a local neighborhood, and vary the strength of the inductive bias via the filter size of convolutional kernels or shallow convolutional neural networks — small filter sizes encourage functions that depend nonlinearly only on local

---

[*]Equal contribution; correspondence to `research@michaelaerni.com`

neighborhoods of the input features. As a second example, we also investigate classification for rotationally invariant data, where we encourage different degrees of rotational invariance for neural networks. In particular,

- we prove a phase transition between harmless and harmful interpolation that occurs by varying the strength of the inductive bias via the filter size of convolutional kernels for kernel regression in the high-dimensional setting (Theorem 1).
- we further show that, for a weak inductive bias, not only is interpolation harmless but partially fitting the observation noise is in fact necessary (Theorem 2).
- we show the same phase transition experimentally for neural networks with two common inductive biases: varying convolution filter size, and rotational invariance enforced via data augmentation (Section 4).

From a practical perspective, empirical evidence suggests that large neural networks not necessarily benefit from early stopping. Our results match those observations for typical networks with a weak inductive bias; however, we caution that strongly structured models must avoid interpolation, even if they are highly overparameterized.

## 2 RELATED WORK

We now discuss three groups of related work and explain how their theoretical results cannot reflect the phase transition between harmless and harmful interpolation for high-dimensional kernel learning.

*Low-dimensional kernel learning:* Many recent works (Bietti et al., 2021; Favero et al., 2021; Bietti, 2022; Cagnetta et al., 2022) prove statistical rates for kernel regression with convolutional kernels in low-dimensional settings, but crucially rely on ridge regularization. In general, one cannot expect harmless interpolation for such kernels in the low-dimensional regime (Rakhlin & Zhai, 2019; Mallinar et al., 2022; Buchholz, 2022); positive results exist only for very specific adaptive spiked kernels (Belkin et al., 2019b). Furthermore, techniques developed for low-dimensional settings (see, e.g., Schölkopf et al. (2018)) usually suffer from a curse of dimensionality, that is, the bounds become vacuous in high-dimensional settings where the input dimension grows with the number of samples.

*High-dimensional kernel learning:* One line of research (Liang et al., 2020; McRae et al., 2022; Liang & Rakhlin, 2020; Liu et al., 2021) tackles high-dimensional kernel learning and proves non-asymptotic bounds using advanced high-dimensional random matrix concentration tools from El Karoui (2010). However, those results heavily rely on a bounded Hilbert norm assumption. This assumption is natural in the low-dimensional regime, but misleading in the high-dimensional regime, as pointed out in Donhauser et al. (2021b). Another line of research (Ghorbani et al., 2021; 2020; Mei et al., 2021; Ghosh et al., 2022; Misiakiewicz & Mei, 2021; Mei et al., 2022) asymptotically characterizes the precise risk of kernel regression estimators in specific settings with access to a kernel's eigenfunctions and eigenvalues. However, these asymptotic results are insufficient to investigate how varying the filter size of a convolutional kernel affects the risk of a kernel regression estimator. In contrast to both lines of research, we prove tight non-asymptotic matching upper and lower bounds for high-dimensional kernel learning which precisely capture the phase transition described in Section 3.2.

*Overfitting of structured interpolators:* Several works question the generality of harmless interpolation for models that incorporate strong structural assumptions. Examples include structures enforced via data augmentation (Nishi et al., 2021), adversarial training (Rice et al., 2020; Kamath et al., 2021; Sanyal et al., 2021; Donhauser et al., 2021a), neural network architectures (Li et al., 2021), pruning-based sparsity (Chang et al., 2021), and sparse linear models (Wang et al., 2022; Muthukumar et al., 2020; Chatterji & Long, 2022). In this paper, we continue that line of research and offer a new theoretical perspective to characterize when interpolation is expected to be harmless.

## 3 THEORETICAL RESULTS

For convolutional kernels, a small filter size induces a strong bias towards estimators that depend nonlinearly on the input features only via small patches. This section analyzes the effect of filter size (as an example inductive bias) on the degree of harmless interpolation for kernel ridge regression.

For this purpose, we derive and compare tight non-asymptotic bias and variance bounds as a function of filter size for min-norm interpolators and optimally ridge-regularized estimators (Theorem 1). Furthermore, we prove for large filter sizes that not only does harmless interpolation occur (Theorem 1), but fitting some degree of noise is even necessary to achieve optimal test performance (Theorem 2).

## 3.1 SETTING

We study kernel regression with a (cyclic) convolutional kernel in a high-dimensional setting where the number of training samples $n$ scales with the dimension of the input data $d$ as $n \in \Theta(d^\ell)$. We use the same setting as in previous works on high-dimensional kernel learning such as Misiakiewicz & Mei (2021): we assume that the training samples $\{x_i, y_i\}_{i=1}^n$ are i.i.d. draws from the distributions $x_i \sim \mathcal{U}(\{-1, 1\}^d)$, and $y_i = f^\star(x_i) + \epsilon_i$ with ground truth $f^\star$ and noise $\epsilon \sim \mathcal{N}(0, \sigma^2)$. For simplicity of exposition, we further assume that $f^\star(x) = x_1 x_2 \cdots x_{L^*}$, with $L^*$ specified in Theorem 1.

While the assumptions on the noise and ground truth can be easily extended by following the proof steps in Section 5, generalizing the feature distribution is challenging. Indeed, existing results that establish precise risk characterizations (see Section 2) crucially rely on hypercontractivity of the feature distribution — an assumption so far only proven for few high-dimensional distributions, including the hypersphere (Beckner, 1992), and the discrete hypercube (Beckner, 1975) which we use in this paper. Hypercontractivity is essential to tightly control the empirical kernel matrix within Lemma 3 in Section 5. Generalizations beyond this assumption require the development of new tools in random matrix theory, which we consider important future work.

We consider (cyclic) convolutional kernels with filter size $q \in \{1, \dots, d\}$ of the form

$$\mathcal{K}(x, x') = \frac{1}{d} \sum_{k=1}^d \kappa \left( \frac{\langle x_{(k,q)}, x'_{(k,q)} \rangle}{q} \right), \tag{1}$$

where $x_{(k,q)} := [x_{\mathrm{mod}(k,d)} \cdots x_{\mathrm{mod}(k+q-1,d)}]$, and $\kappa : [-1, 1] \to \mathbb{R}$ is a nonlinear function that implies standard regularity assumptions (see Assumption 1 in Appendix B) that hold for instance for the exponential function. Decreasing the filter size $q$ restricts kernel regression solutions to depend nonlinearly only on local neighborhoods instead of the entire input $x$.

We analyze the kernel ridge regression (KRR) estimator, which is the minimizer of the following convex optimization problem:

$$\hat{f}_\lambda = \arg\min_{f \in \mathcal{H}} \frac{1}{n} \sum_{i=1}^n (f(x_i) - y_i)^2 + \frac{\lambda}{n} \|f\|_{\mathcal{H}}^2, \tag{2}$$

where $\mathcal{H}$ is the Reproducing Kernel Hilbert space (RKHS) over $\{-1, 1\}^d$ generated by the convolutional kernel $\mathcal{K}$ in Equation (1), $\|\cdot\|_{\mathcal{H}}$ the corresponding norm, and $\lambda > 0$ the ridge regularization penalty.[1] In the interpolation limit ($\lambda \to 0$), we obtain the min-RKHS-norm interpolator

$$\hat{f}_0 = \arg\min_{f \in \mathcal{H}} \|f\|_{\mathcal{H}} \quad \text{s.t.} \quad \forall i : f(x_i) = y_i. \tag{3}$$

For simplicity, we refer to $\hat{f}_0$ as the kernel ridge regression estimator with $\lambda = 0$. We evaluate all estimators with the expected population risk over the noise, defined as

$$\mathbf{Risk}(\hat{f}_\lambda) := \underbrace{\mathbb{E}_x \left[ \left( \mathbb{E}_\epsilon[\hat{f}_\lambda(x)] - f^\star(x) \right)^2 \right]}_{:=\mathbf{Bias}^2(\hat{f}_\lambda)} + \underbrace{\mathbb{E}_{x,\epsilon} \left[ \left( \hat{f}_\lambda(x) - \mathbb{E}_\epsilon[\hat{f}_\lambda(x)] \right)^2 \right]}_{:=\mathbf{Variance}(\hat{f}_\lambda)}.$$

## 3.2 MAIN RESULT

We now present tight upper and lower bounds for the prediction error of kernel regression estimators in the setting from Section 3.1. The resulting rates hold for the high-dimensional regime, that is, when both the ambient dimension $d$ and filter size $q$ scale with $n$.[2] We defer the proof to Section 5.

---

[1] Note that previous works show how early-stopped gradient methods on the square loss behave statistically similarly to kernel ridge regression (Raskutti et al., 2014; Wei et al., 2017).

[2] We hide positive constants that depend at most on $\ell$ and $\beta$ (defined in Theorem 1) using the standard Bachmann–Landau notation $\mathcal{O}(\cdot), \Omega(\cdot), \Theta(\cdot)$, as well as $\lesssim, \gtrsim$, and use $c, c_1, \dots$ as generic positive constants.

**Theorem 1** (Non-asymptotic prediction error rates). *Let $\ell > 0$, $\beta \in (0,1)$, $\ell_\sigma \in \mathbb{R}$. Assume a dataset and a kernel as described in Section 3.1, with the kernel satisfying Assumption 1. Assume further $n \in \Theta(d^\ell)$, the filter size $q \in \Theta\left(d^\beta\right)$, and $\sigma^2 \in \Theta(d^{-\ell_\sigma})$. Lastly, define $\bar{\delta} := \frac{\ell-1}{\beta} - \lfloor \frac{\ell-1}{\beta} \rfloor$ and $\delta := \frac{\ell-\ell_\lambda-1}{\beta} - \lfloor \frac{\ell-\ell_\lambda-1}{\beta} \rfloor$ for any $\ell_\lambda$. Then, with probability at least $1 - cd^{-\beta \min\{\bar{\delta}, 1-\bar{\delta}\}}$ uniformly over all $\ell_\lambda \in [0, \ell-1)$, the KRR estimate $\hat{f}_\lambda$ in Equation (2) with $\max\{\lambda, 1\} \in \Theta(d^{\ell_\lambda})$ satisfies*

$$\mathbf{\textit{Variance}}(\hat{f}_\lambda) \in \Theta\left(n^{\frac{-\ell_\sigma-\ell_\lambda}{\ell} - \frac{\beta}{\ell}\min\{\delta, 1-\delta\}}\right).$$

*Further, for a ground truth $f^\star(x) = x_1 x_2 \cdots x_{L^*}$ with $L^* \leq \left\lceil \frac{\ell-\ell_\lambda-1}{\beta} \right\rceil$, with probability at least $1 - cd^{-\beta \min\{\bar{\delta}, 1-\bar{\delta}\}}$, we have*

$$\mathbf{\textit{Bias}}^2(\hat{f}_\lambda) \in \Theta\left(n^{-2-\frac{2}{\ell}(-\ell_\lambda-1-\beta(L^*-1))}\right).$$

*Finally, by setting $\ell_\lambda = 0$, both rates also hold for the min-RKHS-norm interpolator $\hat{f}_0$ in Eq. (3).*

Note how the theorem reflects the usual intuition for the effects of noise and ridge regularization strength on bias and variance via the parameter $\ell_\lambda$: With increasing ridge regularization $\ell_\lambda$ (and thus increasing $\lambda$), the bias increases and the variance decreases. Similarly, as noise increases (and thus $\ell_\sigma$ decreases), the variance increases.

*Phase transition as a function of $\beta$:* In the following, we focus on the impact of the filter size $q \in \Theta\left(d^\beta\right)$ on the risk (sum of bias and variance) via the growth rate $\beta$. Recalling that a small filter size (small $\beta$) corresponds to a strong inductive bias, and vice versa, Figure 1 demonstrates how the strength of the inductive bias affects generalization. For illustration, we choose the ground truth $f^\star(x) = x_1 x_2$ so that the assumption on $L^*$ is satisfied for all $\beta$. Specifically, Figure 1a shows the rates for the min-RKHS-norm interpolator $\hat{f}_0$ and the optimally ridge-regularized estimator $\hat{f}_{\lambda_{\mathrm{opt}}}$, where we choose $\lambda_{\mathrm{opt}}$ to minimize the expected population risk $\mathbf{Risk}(\hat{f}_{\lambda_{\mathrm{opt}}})$. Furthermore, Figure 1b depicts the (statistical) bias and variance of the interpolator $\hat{f}_0$. At the threshold $\beta^* \in (0,1)$, implicitly defined as the $\beta$ at which the rates of statistical bias and variance in Theorem 1 match, we can observe the following phase transition:

- For $\beta < \beta^*$, that is, for a strong inductive bias, the rates in Figure 1a for the optimally ridge-regularized estimator $\hat{f}_{\lambda_{\mathrm{opt}}}$ are strictly better than the ones for the corresponding interpolator $\hat{f}_0$. In other words, we are observing *harmful interpolation*.
- For $\beta > \beta^*$, that is, for a weak inductive bias, the rates in Figure 1a of the optimally ridge-regularized estimator $\hat{f}_{\lambda_{\mathrm{opt}}}$ and the min-RKHS-norm interpolator $\hat{f}_0$ match. Hence, we observe *harmless interpolation*.

In the following theorem, we additionally show that interpolation is not only harmless for $\beta > \beta^*$, but the optimally ridge-regularized estimator $\hat{f}_{\lambda_{\mathrm{opt}}}$ necessarily fits part of the noise and has a training error strictly below the noise level. In contrast, we show that when interpolation is harmful in Figure 1a, that is, when $\beta < \beta^*$, the training error of the optimally ridge-regularized model approaches the noise level.

**Theorem 2** (Training error (informal)). *Let $\lambda_{opt}$ be such that the expected population risk $\mathbf{Risk}(\hat{f}_{\lambda_{opt}})$ is minimal, and let $\beta^*$ be the unique threshold[3] where the bias and variance bounds in Theorem 1 are of the same order for the interpolator $\hat{f}_0$ (setting $\ell_\lambda = 0$). Then, the expected training error converges in probability:*

$$\lim_{n,d\to\infty} \frac{1}{\sigma^2}\mathbb{E}_\epsilon\left[\frac{1}{n}\sum_i(\hat{f}_{\lambda_{opt}}(x_i) - y_i)^2\right] \quad \begin{cases} = 1 & \beta < \beta^*, \\ \leq c_\beta & \beta \geq \beta^*, \end{cases}$$

*where $c_\beta < 1$ for any $\beta > \beta^*$.*

We refer to Appendix D.2 for the proof and a more general statement.

---

[3]See Theorem 4 for a more general statement that does not rely on a unique $\beta^*$.

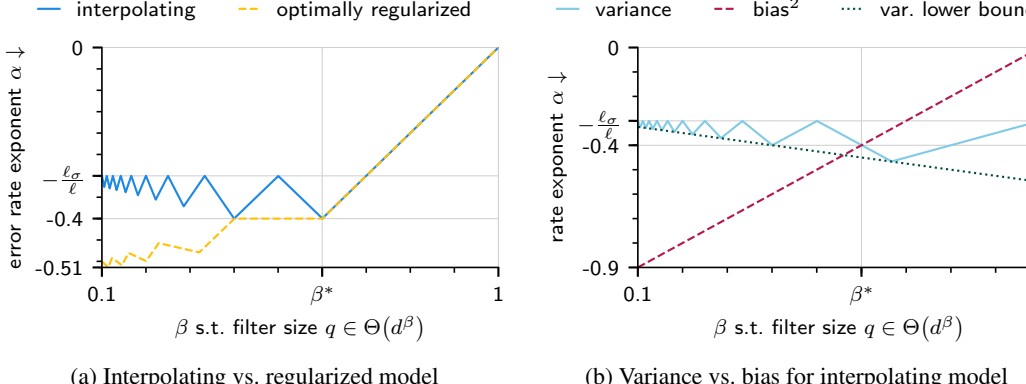

(a) Interpolating vs. regularized model        (b) Variance vs. bias for interpolating model

Figure 1: Illustration of the rates in Theorem 1 for high-dimensional kernel ridge regression as a function of $\beta$ — the rate of the filter size $q \in \Theta(d^\beta)$. (a) Rate exponent $\alpha$ of the **Risk** $\in \Theta(n^\alpha)$ for the interpolator $\hat{f}_0$ vs. the optimally ridge-regularized estimator $\hat{f}_{\lambda_{\text{opt}}}$. (b) Rate exponent of the variance and bias for the interpolator $\hat{f}_0$. For both illustrations, we choose $\hat{f}_0$ with $\ell = 2$, $\ell_\sigma = 0.6$, and the ground truth $f^\star(x) = x_1 x_2$. Lastly, $\beta^*$ denotes the threshold where the bias and variance terms in Theorem 1 match, and where we observe a phase transition between harmless and harmful interpolation. See Appendix D.1 for technical details.

*Bias-variance trade-off:* We conclude by discussing how the phase transition arises from a (statistical) bias and variance trade-off for the min-RKHS-norm interpolator as a function of $\beta$, reflected in Theorem 1 when setting $\ell_\lambda = 0$ and illustrated in Figure 1b. While the statistical bias monotonically decreases with decreasing $\beta$ (i.e., increasing strength of the inductive bias), the variance follows a multiple descent curve with increasing minima as $\beta$ decreases. Hence, analogous to the observations in Donhauser et al. (2022) for linear max-$\ell_p$-margin/min-$\ell_p$-norm interpolators, the interpolator achieves its optimal performance at a $\beta \in (0, 1)$, and therefore at a moderate inductive bias. Finally, we note that Liang et al. (2020) previously observed a multiple descent curve for the variance, but as a function of input dimension and without any connection to structural biases.

## 4 EXPERIMENTS

We now empirically study whether the phase transition phenomenon that we prove for kernel regression persists for deep neural networks with feature learning. More precisely, we present controlled experiments to investigate how the strength of a CNN's inductive bias influences if interpolating noisy data is harmless. In practice, the inductive bias of a neural network varies by way of design choices such as the architecture (e.g., convolutional vs. fully-connected vs. graph networks) or the training procedure (e.g., data augmentation, adversarial training). We focus on two examples: convolutional filter size that we vary via the architecture, and rotational invariance via data augmentation. To isolate the effects of inductive bias and provide conclusive results, we use datasets where we know a priori that the ground truth exhibits a simple structure that matches the networks' inductive bias. See Appendix E for experimental details.

Analogous to ridge regularization for kernels, we use early stopping as a mechanism to prevent noise fitting. Our experiments compare optimally early-stopped CNNs to their interpolating versions. This highlights a trend that mirrors our theoretical results: the stronger the inductive bias of a neural network grows, the more harmful interpolation becomes. These results suggest exciting future work: proving this trend for models with feature learning.

### 4.1 FILTER SIZE OF CNNS ON SYNTHETIC IMAGES

In a first experiment, we study the impact of filter size on the generalization of interpolating CNNs. As a reminder, small filter sizes yield functions that depend nonlinearly only on local neighborhoods of the input features. To clearly isolate the effects of filter size, we choose a special architecture on a synthetic classification problem such that the true label function is indeed a CNN with small filter size. Concretely, we generate images of size $32 \times 32$ containing scattered circles (negative class) and crosses (positive class) with size at most $5 \times 5$. Thus, decreasing filter size down to $5 \times 5$ corresponds to a stronger inductive bias. Motivated by our theory, we hypothesize that interpolating noisy data is harmful with a small filter size, but harmless when using a large filter size.

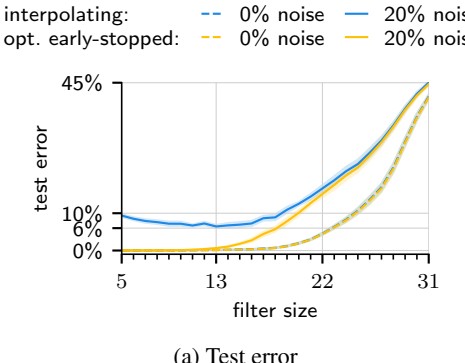
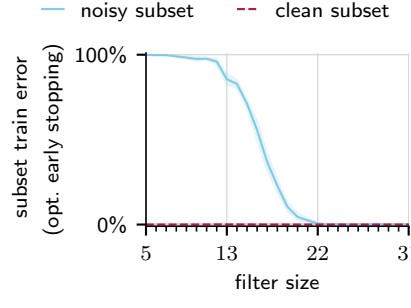

(a) Test error             (b) Training error of optimally early-stopped models

Figure 2: Convolutional neural network experiments with varying filter size on synthetic image data. (a) For noisy data, small filter sizes (strong inductive bias) induce a gap between the generalization performance of interpolating (blue) vs. optimally early-stopped models (yellow). The gap vanishes as the inductive bias decreases (i.e., filter size increases). For noiseless data (dashed), interpolation is always harmless. (b) Training error of the optimally early-stopped model (optimized for test error) on the noisy and clean subsets of a training set with 20% label noise. Under optimal early stopping, models with a strong inductive bias ignore all noisy samples (100% error on the noisy subset), while models with a weak inductive bias fit all noisy training samples (0% error on the noisy subset). All lines show the mean over five random datasets, and shaded areas the standard error; see Section 4.1 for the experiment setup.

*Training setup:* In the experiments, we use CNNs with a single convolutional layer, followed by global spatial max pooling and two dense layers. We train those CNNs with different filter sizes on 200 training samples (either noiseless or with 20% label flips) to minimize the logistic loss and achieve zero training error. We repeat all experiments over 5 random datasets with 15 optimizations per dataset and filter size, and report the average 0-1-error for 100k test samples per dataset. For a detailed discussion on the choice of hyperparameters and more experimental details, see Appendix E.1.

*Results:* First, the noiseless error curves (dashed) in Figure 2a confirm the common intuition that the strongest inductive bias (matching the ground truth) at size 5 yields the lowest test error. More interestingly, for 20% training noise (solid), Figure 2a reveals a similar phase transition as Theorem 1 and confirms our hypothesis: Models with weak inductive biases (large filter sizes) exhibit harmless interpolation, as indicated by the matching test error of interpolating (blue) and optimally early-stopped (yellow) models. In contrast, as filter size decreases, models with a strong inductive bias (small filter sizes) suffer from an increasing gap in test errors when interpolating versus using optimal early stopping. Furthermore, Figure 2b reflects the dual perspective of the phase transition as presented in Theorem 2 under optimal early stopping: models with a small filter size entirely avoid fitting training noise, such that the training error on the noisy training subset equals 100%, while models with a large filter size interpolate the noise.

*Difference to double descent:* One might suspect that our empirical observations simply reflect another form of double descent (Belkin et al., 2019a). As a CNN's filter size increases (inductive bias becomes weaker), so does the number of parameters and degree of overparameterization. Thus, double descent predicts vanishing benefits of regularization due to model size for weak inductive biases. Nevertheless, we argue that the phenomenon we observe here is distinct, and provide an extended discussion in Appendix E.3. In short, we choose sufficiently large networks and tune their hyperparameters to ensure that all models interpolate and yield small training loss, even for filter size 5 and 20% training noise. To justify this approach, we repeat a subset of the experiments while significantly increasing the convolutional layer width. As the number of parameters increases for a fixed filter size, double descent would predict that the benefits of optimal early stopping vanish. However, we observe that our phenomenon persists. In particular, for filter size 5 (strongest inductive bias), the test error gap between interpolating and optimally early-stopped models remains large.

## 4.2 ROTATIONAL INVARIANCE OF WIDE RESIDUAL NETWORKS ON SATELLITE IMAGES

In a second experiment, we investigate rotational invariance as an inductive bias for CNNs whenever true labels are independent of an image's rotation. Our experiments control inductive bias strength by fitting models on multiple rotated versions of an original training dataset, effectively performing

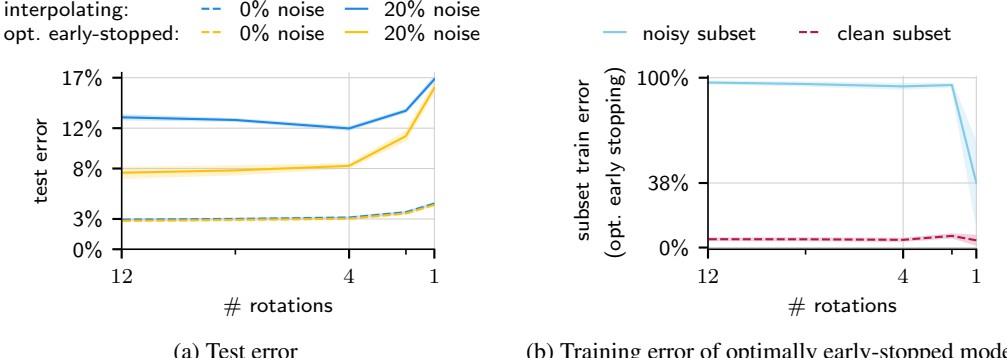

(a) Test error

(b) Training error of optimally early-stopped models

Figure 3: Varying degrees of rotational invariance when fitting Wide Residual Networks on satellite images. (a) For noisy data (solid), a strong bias towards rotational invariance (via the number of augmented rotations) induces a gap between the generalization performance of interpolating (blue) vs. optimally early-stopped models (yellow). The gap decreases as the inductive bias decreases (# rotations decreases). (b) Training error of optimally early-stopped models (w.r.t. the test error) on the noisy and clean subsets of a training set with $20\%$ label noise: For maximum rotational invariance (12 rotations), optimally early-stopped models avoid fitting noisy data (close to $100\%$ training error on noisy subset), yet no rotational invariance (1 rotation) requires fitting noise (less than $50\%$ training error on noisy subset) for optimal generalization. All lines show the mean over five optimization runs, and shaded areas the standard error; see Section 4.2 for the experiment setup.

varying degrees of data augmentation.[4] As an example dataset with a rotationally invariant ground truth, we classify satellite images from the EuroSAT dataset (Helber et al., 2018) into 10 types of land usage. Because the true labels are independent of image orientation, we expect rotational invariance to be a particularly fitting inductive bias for this task.

*Training and test setup:* For computational reasons, we subsample the original EuroSAT training set to 7680 raw training and 10k raw test samples. In the noisy case, we replace $20\%$ of the raw training labels with a wrong label chosen uniformly at random. We then vary the strength of the inductive bias towards rotational invariance by augmenting the dataset with an increasing number of $k$ rotated versions of itself. For each sample, we use $k$ equal-spaced angles spanning $360°$, plus a random offset. Note that training noise applies before rotations, so that all rotated versions of the same image share the same label. We then center-crop all rotated images such that they only contain valid pixels. In all experiments, we fit Wide Residual Networks (Zagoruyko & Komodakis, 2016) on the augmented training set for 5 different network initializations. We evaluate the 0-1-error on the randomly rotated test samples to avoid distribution shift effects from image interpolation. All random rotations are the same for all experiments and stay fixed throughout training. See Appendix E.2 for more experimental details. Lastly, we perform additional experiments with larger models to differentiate from double descent; see Appendix E.3 for the results and further discussions.

*Results:* Similar to the previous subsection, Figure 3a corroborates our hypothesis under rotational invariance: stronger inductive biases result in lower test errors on noiseless data, but an increased gap between the test errors of interpolating and optimally early-stopped models. In contrast to filter size, the phase transition is more abrupt; invariance to 180° rotations already prevents harmless interpolation. Figure 3b confirms this from a dual perspective, since all models with some rotational invariance cannot fit noisy samples for optimal generalization.

## 5 PROOF OF THE MAIN RESULT

The proof of the main result, Theorem 1, proceeds in two steps: First, Section 5.1 presents a fixed-design result that yields matching upper and lower bounds for the prediction error of general kernels under additional conditions. Second, in Section 5.2, we show that the setting of Theorem 1 satisfies those conditions with high probability over dataset draws.

---

[4]Data augmentation techniques can efficiently enforce rotational invariance; see, e.g., Yang et al. (2019).

**Notation** Assuming that inputs are draws from a data distribution $\nu$ (i.e., $x, x' \sim \nu$), we can decompose and divide any continuous, positive semi-definite kernel function as

$$\mathcal{K}(x, x') = \sum_{k=1}^{\infty} \lambda_k \psi_k(x) \psi_k(x') = \underbrace{\sum_{k=1}^{m} \lambda_k \psi_k(x) \psi_k(x')}_{:=\mathcal{K}_{\leq m}(x, x')} + \underbrace{\sum_{k=m+1}^{\infty} \lambda_k \psi_k(x) \psi_k(x')}_{:=\mathcal{K}_{>m}(x, x')}, \quad (4)$$

where $\{\psi_k\}_{k \geq 1}$ is an orthonormal basis of the RKHS induced by $\langle f, g \rangle_\nu := \mathbb{E}_{x \sim \nu}[f(x)g(x)]$ and the eigenvalues $\lambda_k$ are sorted in descending order. In the following, we write $[\cdot]_{i,j}$ to refer to the entry in row $i$ and column $j$ of a matrix. Then, we define the empirical kernel matrix for $\mathcal{K}$ as $\mathbf{K} \in \mathbb{R}^{n \times n}$ with $[\mathbf{K}]_{i,j} := \mathcal{K}(x_i, x_j)$, and analogously the truncated versions $\mathbf{K}_{\leq m}$ and $\mathbf{K}_{>m}$ for $\mathcal{K}_{\leq m}$ and $\mathcal{K}_{>m}$, respectively. Next, we utilize the matrices $\mathbf{\Psi}_{\leq m} \in \mathbb{R}^{n \times m}$ with $[\mathbf{\Psi}_{\leq m}]_{i,l} := \psi_l(x_i)$, and $\mathbf{D}_{\leq m} := \mathrm{diag}(\lambda_1, \ldots, \lambda_m)$. We further use the squared kernel $\mathcal{S}(x, x') := \mathbb{E}_{z \sim \nu}[\mathcal{K}(x, z)\mathcal{K}(z, x')]$, its truncated versions $\mathcal{S}_{\leq m}$ and $\mathcal{S}_{>m}$, as well as the corresponding empirical kernel matrices $\mathbf{S}, \mathbf{S}_{\leq m}, \mathbf{S}_{>m} \in \mathbb{R}^{n \times n}$. Next, for a symmetric positive-definite matrix, we write $\mu_{\min}(\cdot)$ and $\mu_{\max}(\cdot)$ (or $\|\cdot\|$) to indicate the min and max eigenvalue, respectively, and $\mu_i(\cdot)$ for the $i$-th eigenvalue in decreasing order. Finally, we use $\langle \cdot, \cdot \rangle$ for the Euclidean inner product in $\mathbb{R}^d$.

## 5.1 GENERALIZATION BOUND FOR FIXED-DESIGN

First, Theorem 3 provides tight fixed-design bounds for the prediction error.

**Theorem 3** (Generalization bound for fixed-design). *Let $\mathcal{K}$ be a kernel that under a distribution $\nu$ decomposes as $\mathcal{K}(x, x') = \sum_k \lambda_k \psi_k(x) \psi_k(x')$ with $\mathbb{E}_{x \sim \nu}[\psi_k(x)\psi_{k'}(x)] = \delta_{k,k'}$, and $\{(x_i, y_i)\}_{i=1}^{n}$ be a dataset with $y_i = f^\star(x_i) + \epsilon_i$ for zero-mean $\sigma^2$-variance i.i.d. noise $\epsilon_i$ and ground truth $f^\star$. Define $\tau_1 := \min\left\{\frac{n\lambda_m}{\max\{\lambda, 1\}}, 1\right\}$, $\tau_2 := \max\left\{\frac{n\lambda_{m+1}}{\max\{\lambda, 1\}}, 1\right\}$, $r_1 := \frac{\mu_{\min}(\mathbf{K}_{>m}) + \lambda}{\max\{\lambda, 1\}}$, $r_2 := \frac{\|\mathbf{K}_{>m}\| + \lambda}{\max\{\lambda, 1\}}$. Then, for any $m \in \mathbb{N}$ such that $r_1 > 0$ and*

$$\left\| \mathbf{\Psi}_{\leq m}^{\mathsf{T}} \mathbf{\Psi}_{\leq m} / n - \mathbf{I}_m \right\| \leq \frac{1}{2}, \quad (5)$$

*the KRR estimate $\hat{f}_\lambda$ in Equation (2) for any $\lambda \geq 0$ has a variance upper and lower bounded by*

$$\frac{r_1^2 \tau_1^2}{2r_2^2(1.5 + r_1)^2} \frac{m}{n} + \frac{\sum_{i=m+1}^{n} \mu_i(\mathbf{S}_{>m})}{r_2^2 \max\{\lambda, 1\}^2} \leq \textit{Variance}(\hat{f}_\lambda)/\sigma^2 \leq 6\frac{r_2^2}{r_1^2} \frac{m}{n} + \frac{\mathbf{Tr}(\mathbf{S}_{>m})}{r_1^2 \max\{\lambda, 1\}^2}.$$

*Furthermore, for any ground truth that can be expressed as $f^\star(x) = \sum_{k=1}^{m} a_k \psi_k(x)$ with $a \in \mathbb{R}_+^m$ and $\psi_k$ as defined in Equation (4), the bias is upper and lower bounded by*

$$\frac{r_1^2 \tau_1^2}{(1.5 + r_1)^2} \max\{\lambda, 1\}^2 \frac{\|\mathbf{D}_{\leq m}^{-1} a\|^2}{n^2} \leq \textit{Bias}^2(\hat{f}_\lambda) \leq 4\left(r_2^2 + 1.5\frac{r_2^3}{r_1^2}\right) \tau_2 \max\{\lambda, 1\}^2 \frac{\|\mathbf{D}_{\leq m}^{-1} a\|^2}{n^2}.$$

See Appendix A.1 for the proof. Note that this result holds for any fixed-design dataset. We derive the main ideas from the proofs in Bartlett et al. (2020); Tsigler & Bartlett (2020), where the authors establish tight bounds for the min-$\ell_2$-norm interpolator on independent sub-Gaussian features.

**Remark 1** (Comparison with McRae et al. (2022)). *The upper bound for the bias in Theorem 1 from McRae et al. (2022) depends on the suboptimal term $\|\mathbf{D}_{\leq m}^{-1/2} a\|/n$, but also applies to more general ground truths. We improve that upper bound in Theorem 3 and present a matching lower bound.*

## 5.2 PROOF OF THEOREM 1

Throughout the remainder of the proof, all statements hold for the setting in Section 3.1 and under the assumptions of Theorem 1, especially Assumption 1 on $\mathcal{K}$, $n \in \Theta(d^\ell)$, and $\max\{\lambda, 1\} \in \Theta(d^{\ell_\lambda})$. Furthermore, as in Theorem 1, we use $\delta, \bar{\delta} > 0$ with $\delta := \frac{\ell - \ell_\lambda - 1}{\beta} - \left\lfloor \frac{\ell - \ell_\lambda - 1}{\beta} \right\rfloor$ and $\bar{\delta} := \frac{\ell - 1}{\beta} - \left\lfloor \frac{\ell - 1}{\beta} \right\rfloor$. We show that there exists a particular $m$ for which the conditions in Theorem 3 hold and we can control the terms in the bias and variance bounds.

**Step 1: Conditions for the bounds in Theorem 3**    We first derive sufficient conditions on $m$ such that the conditions on $\mathbf{\Psi}_{\leq m}$ and $f^\star$ in Theorem 3 hold with high probability. The following standard concentration result shows that all $m \ll n$ satisfy Equation (5) with high probability.

**Lemma 1** (Corollary of Theorem 5.44 in Vershynin (2012)).  *For $d$ large enough, with probability at least $1 - cd^{-\beta\bar{\delta}}$, all $m \in \mathcal{O}(n \cdot q^{-\bar{\delta}})$ satisfy Equation (5).*

See Appendix C.1 for the proof. Simultaneously, to ensure that $f^\star$ is contained in the span of the first $m$ eigenfunctions, $m$ must be sufficiently large. We formalize this in the following lemma.

**Lemma 2** (Bias bound condition).  *Consider a kernel as in Theorem 1 satisfying Assumption 1, and a ground truth $f^\star(x) = \prod_{j=1}^{L^*} x_j$ with $1 \leq L^* \leq \left\lceil \frac{\ell - \ell_\lambda - 1}{\beta} \right\rceil$. Then, for any $m$ with $\lambda_m \in o\left(\frac{1}{dq^{L^*-1}}\right)$ and $d$ sufficiently large, $f^\star$ is in the span of the first $m$ eigenfunctions and $\left\| \mathbf{D}_{\leq m}^{-1} a \right\| \in \Theta\left(dq^{L^*-1}\right)$.*

See Appendix B.3 for the proof. Note that $L^* \geq 1$ follows from $\ell_\lambda < \ell - 1$ in Theorem 1, and allows us to focus on non-trivial ground truth functions.

**Step 2: Concentration of the (squared) kernel matrix**    In a second step, we show that there exists a set of $m$ that satisfy the sufficient conditions, and for which the spectra of the kernel matrix $\mathbf{K}_{>m}$ and squared kernel matrix $\mathbf{S}_{>m}$ concentrate.

**Lemma 3** (Tight bound conditions).  *With probability at least $1 - cd^{-\beta\min\{\bar{\delta}, 1-\bar{\delta}\}}$ uniformly over all $\ell_\lambda \in [0, \ell - 1)$, for $d$ sufficiently large and any $\lambda \in \mathbb{R}$, $m \in \mathbb{N}$ with $\max\{\lambda, 1\} \in \Theta(d^{\ell_\lambda})$, $n\lambda_m \in \Theta(\max\{\lambda, 1\})$, and $m \in \mathcal{O}\left(\frac{nq^{-\delta}}{\max\{\lambda, 1\}}\right)$, we have*

$$r_1, r_2 \in \Theta(1), \quad \mathbf{Tr}(\mathbf{S}_{>m}) \in \mathcal{O}\left(d^{\ell_\lambda}q^{-\delta} + d^{\ell_\lambda}q^{-(1-\delta)}\right), \quad \sum_{i=m+1}^{n} \mu_i(\mathbf{S}_{>m}) \in \Omega\left(d^{\ell_\lambda}q^{-(1-\delta)}\right).$$

We refer to Appendix C.3 for the proof, which heavily relies on the feature distribution. Note that the results for $m$ in Lemma 3 also imply $\tau_1, \tau_2 \in \Theta(1)$.

**Step 3: Completing the proof**    Finally, we complete the proof by showing the existence of a particular $m$ that simultaneously satisfies all conditions of Lemmas 1 to 3.

**Lemma 4** (Eigendecay).  *There exists an $m$ such that*

$$n\lambda_m \in \Theta(\max\{\lambda, 1\}) \qquad and \qquad m \in \Theta\left(\frac{nq^{-\delta}}{\max\{\lambda, 1\}}\right) \subseteq \mathcal{O}(n \cdot q^{-\bar{\delta}}).$$

*Furthermore, assuming $L^* \leq \left\lceil \frac{\ell - \ell_\lambda - 1}{\beta} \right\rceil$, we have $\lambda_m \in o\left(\frac{1}{d \cdot q^{L^*-1}}\right)$.*

We refer to Appendix B.4 for the proof. As a result, we can use Lemmas 1 to 4 to instantiate Theorem 3 for the setting in Theorem 1, resulting in the following tight high-probability bounds for variance and bias:

$$d^{-\ell_\lambda}q^{-\delta} + d^{-\ell_\lambda}q^{-(1-\delta)} \lesssim \mathbf{Variance}(\hat{f}_\lambda)/\sigma^2 \lesssim d^{-\ell_\lambda}q^{-\delta} + d^{-\ell_\lambda}q^{-(1-\delta)},$$
$$d^{-2(\ell-\ell_\lambda-1-\beta(L^*-1))} \lesssim \mathbf{Bias}^2(\hat{f}_\lambda) \lesssim d^{-2(\ell-\ell_\lambda-1-\beta(L^*-1))}.$$

Reformulating the bounds in terms of $n$ then concludes the proof of Theorem 1.

# 6    SUMMARY AND OUTLOOK

In this paper, we highlight how the strength of an inductive bias impacts generalization. Concretely, we study when the gap in test error between interpolating models and their optimally ridge-regularized or early-stopped counterparts is zero, that is, when interpolation is harmless. In particular, we prove a phase transition for kernel regression using convolutional kernels with different filter sizes: a weak inductive bias (large filter size) yields harmless interpolation, and even requires fitting noise for optimal test performance, whereas a strong inductive bias (small filter size) suffers from suboptimal generalization when interpolating noise. Intuitively, this phenomenon arises from a bias-variance trade-off, captured by our main result in Theorem 1: with increasing inductive bias, the risk on noiseless training samples (bias) decreases, while the sensitivity to noise in the training data (variance) increases. Our empirical results on neural networks suggest that this phenomenon extends to models with feature learning, which opens up an avenue for exciting future work.

ACKNOWLEDGMENTS

K.D. is supported by the ETH AI Center and the ETH Foundations of Data Science. We would further like to thank Afonso Bandeira for insightful discussions.

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

# A  Generalization bound for fixed-design

We prove Theorem 3 by deriving a closed-form expression for the bias and variance, and bounding them individually. Hence, the proof does not rely on any matrix concentration results.

## A.1  Proof of Theorem 3

It is well-known that the KRR problem defined in Equation (2) yields the estimator

$$\hat{f}_\lambda(x) = \mathbf{y}^\mathsf{T}\mathbf{H}^{-1}\mathbf{k}(x),$$

where $x, x_i \sim \nu$, $\mathbf{y} := \mathbf{f} + \epsilon = [f^\star(x_1) + \epsilon_1, \dots, f^\star(x_n) + \epsilon_n]^\mathsf{T}$, $\mathbf{k}(x) := [\mathcal{K}(x_1, x), \dots, \mathcal{K}(x_n, x)]^\mathsf{T}$, and $\mathbf{H} := \mathbf{K} + \lambda\mathbf{I}_n$. For this estimator, both bias and variance exhibit a closed-form expression:

$$
\begin{aligned}
\mathbf{Bias}^2(\hat{f}_\lambda) &= \mathbb{E}_{x\sim\nu}[(f^\star(x) - \mathbb{E}_\epsilon[(\mathbf{f}+\epsilon)^\mathsf{T}\mathbf{H}^{-1}\mathbf{k}(x)])^2] \\
&= \mathbb{E}_{x\sim\nu}[(f^\star(x) - \mathbf{f}^\mathsf{T}\mathbf{H}^{-1}\mathbf{k}(x))^2] \\
&= \mathbb{E}_{x\sim\nu}[f^\star(x)^2] - 2\mathbf{f}^\mathsf{T}\mathbf{H}^{-1}\mathbb{E}_{x\sim\nu}[f^\star(x)\mathbf{k}(x)] + \mathbf{f}^\mathsf{T}\mathbf{H}^{-1}\mathbb{E}_{x\sim\nu}[\mathbf{k}(x)\mathbf{k}(x)^\mathsf{T}]\mathbf{H}^{-1}\mathbf{f} \\
&\overset{(i)}{=} a^\mathsf{T}a - 2a^\mathsf{T}\mathbf{\Psi}_{\le m}^\mathsf{T}\mathbf{H}^{-1}\mathbf{\Psi}_{\le m}\mathbf{D}_{\le m}a + a^\mathsf{T}\mathbf{\Psi}_{\le m}^\mathsf{T}\mathbf{H}^{-1}\mathbf{S}\mathbf{H}^{-1}\mathbf{\Psi}_{\le m}a, \qquad (6)
\end{aligned}
$$

$$
\begin{aligned}
\mathbf{Variance}(\hat{f}_\lambda)/\sigma^2 &= \frac{1}{\sigma^2}\mathbb{E}_{x\sim\nu}\mathbb{E}_\epsilon[((\mathbf{f}+\epsilon)^\mathsf{T}\mathbf{H}^{-1}\mathbf{k}(x) - \mathbb{E}_\epsilon[(\mathbf{f}+\epsilon)^\mathsf{T}\mathbf{H}^{-1}\mathbf{k}(x)])^2] \\
&= \frac{1}{\sigma^2}\mathbb{E}_{x\sim\nu}\mathbb{E}_\epsilon[\epsilon^\mathsf{T}\mathbf{H}^{-1}\mathbf{k}(x)\mathbf{k}(x)^\mathsf{T}\mathbf{H}^{-1}\epsilon] \\
&= \frac{1}{\sigma^2}\sum_{i,j}\mathbb{E}[\epsilon_i\epsilon_j]\left[\mathbf{H}^{-1}\mathbb{E}_{x\sim\nu}[\mathbf{k}(x)\mathbf{k}(x)^\mathsf{T}]\mathbf{H}^{-1}\right]_{i,j} \\
&\overset{(i)}{=} \mathbf{Tr}\left(\mathbf{H}^{-1}\mathbf{S}\mathbf{H}^{-1}\right). \qquad (7)
\end{aligned}
$$

Step $(i)$ uses the definition of the squared kernel $\mathcal{S}$, that $f^\star(x) = \sum_{k=1}^m a_k\psi_k(x)$ as $f^\star$ is in the span of the first $m$ eigenfunctions, and the following consequences of the eigenfunctions' orthonormality:

$$
\mathbb{E}_{x\sim\nu}\left[f^\star(x)^2\right] = \sum_{k,k'=1}^m a_k a_{k'}\mathbb{E}_{x\sim\nu}\left[\psi_k(x)\psi_{k'}(x)\right] = \sum_{k,k'=1}^m a_k a_{k'}\delta_{k,k'} = \|a\|_2^2,
$$

$$
\begin{aligned}
\mathbb{E}_{x\sim\nu}\left[[f^\star(x)\mathbf{k}(x)]_i\right] &= \sum_{k=1}^m\sum_{k'=1}^\infty a_k\lambda_{k'}\mathbb{E}_{x\sim\nu}\left[\psi_k(x)\psi_{k'}(x)\right]\psi_{k'}(x_i) \\
&= \sum_{k=1}^m a_k\lambda_k\psi_k(x_i) = [\mathbf{\Psi}_{\le m}\mathbf{D}_{\le m}a]_i.
\end{aligned}
$$

We now bound the closed-form expressions of bias and variance individually.

**Bias**  Lemma 5 below yields $\mathbf{S} = \mathbf{\Psi}_{\le m}\mathbf{D}_{\le m}^2\mathbf{\Psi}_{\le m}^\mathsf{T} + \mathbf{S}_{>m}$. Hence, the bias decomposes into

$$
\begin{aligned}
\mathbf{Bias}^2(\hat{f}_\lambda) &= a^\mathsf{T}a - 2a^\mathsf{T}\mathbf{\Psi}_{\le m}^\mathsf{T}\mathbf{H}^{-1}\mathbf{\Psi}_{\le m}\mathbf{D}_{\le m}a + a^\mathsf{T}\mathbf{\Psi}_{\le m}^\mathsf{T}\mathbf{H}^{-1}\mathbf{\Psi}_{\le m}\mathbf{D}_{\le m}^2\mathbf{\Psi}_{\le m}^\mathsf{T}\mathbf{H}^{-1}\mathbf{\Psi}_{\le m}a \\
&\quad + a^\mathsf{T}\mathbf{\Psi}_{\le m}^\mathsf{T}\mathbf{H}^{-1}\mathbf{S}_{>m}\mathbf{H}^{-1}\mathbf{\Psi}_{\le m}a \\
&= \underbrace{\left\|\left(\mathbf{I}_m - \mathbf{D}_{\le m}\mathbf{\Psi}_{\le m}^\mathsf{T}\mathbf{H}^{-1}\mathbf{\Psi}_{\le m}\right)a\right\|^2}_{:=B_1} + \underbrace{a^\mathsf{T}\mathbf{\Psi}_{\le m}^\mathsf{T}\mathbf{H}^{-1}\mathbf{S}_{>m}\mathbf{H}^{-1}\mathbf{\Psi}_{\le m}a}_{:=B_2}.
\end{aligned}
$$

First, we rewrite $B_1$ as

$$
\begin{aligned}
B_1 &= \left\|\left(\mathbf{I}_m - \mathbf{D}_{\le m}\mathbf{\Psi}_{\le m}^\mathsf{T}\mathbf{H}^{-1}\mathbf{\Psi}_{\le m}\right)a\right\|^2 \\
&\overset{(i)}{=} \left\|\left(\mathbf{D}_{\le m} - \mathbf{D}_{\le m}\mathbf{\Psi}_{\le m}^\mathsf{T}\left(\mathbf{\Psi}_{\le m}\mathbf{D}_{\le m}\mathbf{\Psi}_{\le m}^\mathsf{T} + \mathbf{H}_{>m}\right)^{-1}\mathbf{\Psi}_{\le m}\mathbf{D}_{\le m}\right)\mathbf{D}_{\le m}^{-1}a\right\|^2 \\
&\overset{(ii)}{=} \left\|\left(\mathbf{D}_{\le m}^{-1} + \mathbf{\Psi}_{\le m}^\mathsf{T}\mathbf{H}_{>m}^{-1}\mathbf{\Psi}_{\le m}\right)^{-1}\mathbf{D}_{\le m}^{-1}a\right\|^2, \qquad (8)
\end{aligned}
$$

where $(i)$ uses the decomposition $\mathbf{H} = \mathbf{K}_{\leq m} + \mathbf{H}_{>m}$ with $\mathbf{H}_{>m} := \mathbf{K}_{>m} + \lambda\mathbf{I}_n$, and $(ii)$ applies the Woodbury matrix identity.

Second, we can upper-bound $B_2$ as follows:

$$
B_2 \leq \|\mathbf{S}_{>m}\| a^{\intercal}\boldsymbol{\Psi}_{\leq m}^{\intercal}\mathbf{H}^{-2}\boldsymbol{\Psi}_{\leq m}a = \|\mathbf{S}_{>m}\| \left(\mathbf{D}_{\leq m}^{-1}a\right)^{\intercal}\mathbf{D}_{\leq m}\boldsymbol{\Psi}_{\leq m}^{\intercal}\mathbf{H}^{-2}\boldsymbol{\Psi}_{\leq m}\mathbf{D}_{\leq m}\left(\mathbf{D}_{\leq m}^{-1}a\right)
$$

$$
\overset{(i)}{=} \|\mathbf{S}_{>m}\| a^{\intercal}\mathbf{D}_{\leq m}^{-1}\left(\mathbf{D}_{\leq m}^{-1} + \boldsymbol{\Psi}_{\leq m}^{\intercal}\mathbf{H}_{>m}^{-1}\boldsymbol{\Psi}_{\leq m}\right)^{-1}\boldsymbol{\Psi}_{\leq m}^{\intercal}\mathbf{H}_{>m}^{-2}\boldsymbol{\Psi}_{\leq m}
$$

$$
\left(\mathbf{D}_{\leq m}^{-1} + \boldsymbol{\Psi}_{\leq m}^{\intercal}\mathbf{H}_{>m}^{-1}\boldsymbol{\Psi}_{\leq m}\right)^{-1}\mathbf{D}_{\leq m}^{-1}a
$$

$$
\leq \underbrace{\|\mathbf{S}_{>m}\|\left\|\boldsymbol{\Psi}_{\leq m}^{\intercal}\mathbf{H}_{>m}^{-2}\boldsymbol{\Psi}_{\leq m}\right\|}_{:=C_1}\underbrace{\left\|\left(\mathbf{D}_{\leq m}^{-1} + \boldsymbol{\Psi}_{\leq m}^{\intercal}\mathbf{H}_{>m}^{-1}\boldsymbol{\Psi}_{\leq m}\right)^{-1}\mathbf{D}_{\leq m}^{-1}a\right\|^2}_{B_1},
$$

where $(i)$ uses Lemma 6. Thus, the bias can be bounded by $B_1(1 + C_1)$ with $C_1 \geq 0$.

We proceed by upper bounding $C_1$:

$$
1 + C_1 \leq 1 + n\|\mathbf{S}_{>m}\|\left\|\frac{\boldsymbol{\Psi}_{\leq m}^{\intercal}\boldsymbol{\Psi}_{\leq m}}{n}\right\|\frac{1}{\mu_{\min}\left(\mathbf{H}_{>m}\right)^2}
$$

$$
\overset{(i)}{\leq} 1 + 1.5\frac{n\lambda_{m+1}\|\mathbf{K}_{>m}\|}{\left(\mu_{\min}\left(\mathbf{K}_{>m}\right) + \lambda\right)^2} \leq 1 + 1.5\frac{n\lambda_{m+1}\left(\|\mathbf{K}_{>m}\| + \lambda\right)}{\left(\mu_{\min}\left(\mathbf{K}_{>m}\right) + \lambda\right)^2}
$$

$$
\leq 1 + 1.5\frac{n\lambda_{m+1}}{\max\{\lambda, 1\}}\frac{\max\{\lambda, 1\}^2}{\left(\mu_{\min}\left(\mathbf{K}_{>m}\right) + \lambda\right)^2}\frac{\|\mathbf{K}_{>m}\| + \lambda}{\max\{\lambda, 1\}}
$$

$$
\leq 1 + 1.5\frac{r_2}{r_1^2}\frac{n\lambda_{m+1}}{\max\{\lambda, 1\}}
$$

$$
\overset{(ii)}{\leq} \left(1 + 1.5\frac{r_2}{r_1^2}\right)\max\left\{\frac{n\lambda_{m+1}}{\max\{\lambda, 1\}}, 1\right\}
$$

$$
= \left(1 + 1.5\frac{r_2}{r_1^2}\right)\tau_2, \tag{9}
$$

where $(i)$ uses Equation (5) to bound $\|\boldsymbol{\Psi}_{\leq m}^{\intercal}\boldsymbol{\Psi}_{\leq m}/n\|$ and Lemma 7 to bound $\|\mathbf{S}_{>m}\|$, and $(ii)$ follows from $cx + 1 \leq (c + 1)\max\{x, 1\}$ for $c \geq 0$.

Hence, to conclude the bias bound, we need to bound $B_1$ in Equation (8) from above and below.

*Upper bound:*

$$
B_1 \leq \left\|\left(\frac{\mathbf{D}_{\leq m}^{-1}}{n} + \frac{\boldsymbol{\Psi}_{\leq m}^{\intercal}\mathbf{H}_{>m}^{-1}\boldsymbol{\Psi}_{\leq m}}{n}\right)^{-1}\right\|^2\frac{\|\mathbf{D}_{\leq m}^{-1}a\|^2}{n^2}
$$

$$
\leq \frac{\|\mathbf{H}_{>m}\|^2}{\mu_{\min}\left(\frac{\boldsymbol{\Psi}_{\leq m}^{\intercal}\boldsymbol{\Psi}_{\leq m}}{n}\right)^2}\frac{\|\mathbf{D}_{\leq m}^{-1}a\|^2}{n^2}
$$

$$
\overset{(i)}{\leq} 4\left(\|\mathbf{K}_{>m}\| + \lambda\right)^2\frac{\|\mathbf{D}_{\leq m}^{-1}a\|^2}{n^2}
$$

$$
= 4r_2^2\max\{\lambda, 1\}^2\frac{\|\mathbf{D}_{\leq m}^{-1}a\|^2}{n^2} \tag{10}
$$

where $(i)$ follows from Equation (5).

Combining Equation (9) in $(i)$ and Equation (10) in $(ii)$ yields the desired upper bound on the bias:

$$
\textbf{Bias}^2 \leq B_1 + B_2 \leq (1 + C_1)B_1 \overset{(i)}{\leq} \left(1 + 1.5\frac{r_2}{r_1^2}\right)\tau_2 B_1 \overset{(ii)}{\leq} 4\left(1 + 1.5\frac{r_2}{r_1^2}\right)r_2^2\tau_2\max\{\lambda, 1\}^2\frac{\|\mathbf{D}_{\leq m}^{-1}a\|^2}{n^2}.
$$

*Lower bound:*

$$B_1 \geq \mu_{\min} \left( \left( \frac{\mathbf{D}_{\leq m}^{-1}}{n} + \frac{\boldsymbol{\Psi}_{\leq m}^{\mathsf{T}} \mathbf{H}_{>m}^{-1} \boldsymbol{\Psi}_{\leq m}}{n} \right)^{-1} \right)^2 \frac{\|\mathbf{D}_{\leq m}^{-1} a\|^2}{n^2}$$

$$\geq \frac{1}{\left( \frac{1}{n\lambda_m} + \frac{1}{\mu_{\min}(\mathbf{H}_{>m})} \left\| \frac{\boldsymbol{\Psi}_{\leq m}^{\mathsf{T}} \boldsymbol{\Psi}_{\leq m}}{n} \right\| \right)^2} \frac{\|\mathbf{D}_{\leq m}^{-1} a\|^2}{n^2}$$

$$\overset{(i)}{\geq} \left( \frac{\frac{n\lambda_m}{\max\{\lambda,1\}}}{1.5 \frac{\max\{\lambda,1\}}{\mu_{\min}(\mathbf{K}_{>m})+\lambda} \frac{n\lambda_m}{\max\{\lambda,1\}} + 1} \right)^2 \max\{\lambda,1\}^2 \frac{\|\mathbf{D}_{\leq m}^{-1} a\|^2}{n^2}$$

$$\overset{(ii)}{\geq} \left( \frac{r_1}{1.5 + r_1} \right)^2 \min\left\{ \frac{n\lambda_m}{\max\{\lambda,1\}}, 1 \right\}^2 \max\{\lambda,1\}^2 \frac{\|\mathbf{D}_{\leq m}^{-1} a\|^2}{n^2},$$

$$= \left( \frac{r_1}{1.5 + r_1} \right)^2 \tau_1^2 \max\{\lambda,1\}^2 \frac{\|\mathbf{D}_{\leq m}^{-1} a\|^2}{n^2},$$

where $(i)$ follows from Equation (5), and $(ii)$ from the fact that $\frac{x}{cx+1} \geq \frac{1}{1+c} \min\{x,1\}$ for $x, c \geq 0$. Since $\mathbf{Bias}^2 \geq B_1$, this concludes the lower bound for the bias.

**Variance**   As for the bias bound, we first apply Lemma 5 to write $\mathbf{S} = \boldsymbol{\Psi}_{\leq m} \mathbf{D}_{\leq m}^2 \boldsymbol{\Psi}_{\leq m}^{\mathsf{T}} + \mathbf{S}_{>m}$ and decompose the variance in Equation (7) into

$$\mathbf{Variance}(\hat{f}_\lambda)/\sigma^2 = \underbrace{\mathbf{Tr}\left( \mathbf{H}^{-1} \boldsymbol{\Psi}_{\leq m} \mathbf{D}_{\leq m}^2 \boldsymbol{\Psi}_{\leq m}^{\mathsf{T}} \mathbf{H}^{-1} \right)}_{:=V_1} + \underbrace{\mathbf{Tr}\left( \mathbf{H}^{-1} \mathbf{S}_{>m} \mathbf{H}^{-1} \right)}_{:=V_2}. \tag{11}$$

Next, we rewrite $V_1$ as follows:

$$V_1 = \mathbf{Tr}\left( \mathbf{H}^{-1} \boldsymbol{\Psi}_{\leq m} \mathbf{D}_{\leq m}^2 \boldsymbol{\Psi}_{\leq m}^{\mathsf{T}} \mathbf{H}^{-1} \right) = \mathbf{Tr}\left( \mathbf{D}_{\leq m} \boldsymbol{\Psi}_{\leq m}^{\mathsf{T}} \mathbf{H}^{-2} \boldsymbol{\Psi}_{\leq m} \mathbf{D}_{\leq m} \right)$$

$$\overset{(i)}{=} \mathbf{Tr}\left( \left( \mathbf{D}_{\leq m}^{-1} + \boldsymbol{\Psi}_{\leq m}^{\mathsf{T}} \mathbf{H}_{>m}^{-1} \boldsymbol{\Psi}_{\leq m} \right)^{-1} \boldsymbol{\Psi}_{\leq m}^{\mathsf{T}} \mathbf{H}_{>m}^{-2} \boldsymbol{\Psi}_{\leq m} \left( \mathbf{D}_{\leq m}^{-1} + \boldsymbol{\Psi}_{\leq m}^{\mathsf{T}} \mathbf{H}_{>m}^{-1} \boldsymbol{\Psi}_{\leq m} \right)^{-1} \right)$$

$$= \frac{1}{n} \mathbf{Tr}\left( \left( \frac{\mathbf{D}_{\leq m}^{-1}}{n} + \frac{\boldsymbol{\Psi}_{\leq m}^{\mathsf{T}} \mathbf{H}_{>m}^{-1} \boldsymbol{\Psi}_{\leq m}}{n} \right)^{-1} \frac{\boldsymbol{\Psi}_{\leq m}^{\mathsf{T}} \mathbf{H}_{>m}^{-2} \boldsymbol{\Psi}_{\leq m}}{n} \left( \frac{\mathbf{D}_{\leq m}^{-1}}{n} + \frac{\boldsymbol{\Psi}_{\leq m}^{\mathsf{T}} \mathbf{H}_{>m}^{-1} \boldsymbol{\Psi}_{\leq m}}{n} \right)^{-1} \right),$$

where $(i)$ follows from Lemma 6.

To bound $V_1$ and $V_2$, we use the fact that the trace is the sum of all eigenvalues. Therefore, the trace is bounded from above and below by the size of the matrix times the largest and smallest eigenvalue, respectively. This yields the following bounds for $V_1$:

$$V_1 \leq \frac{m}{n} \left\| \left( \frac{\mathbf{D}_{\leq m}^{-1}}{n} + \frac{\boldsymbol{\Psi}_{\leq m}^{\mathsf{T}} \mathbf{H}_{>m}^{-1} \boldsymbol{\Psi}_{\leq m}}{n} \right)^{-1} \frac{\boldsymbol{\Psi}_{\leq m}^{\mathsf{T}} \mathbf{H}_{>m}^{-2} \boldsymbol{\Psi}_{\leq m}}{n} \left( \frac{\mathbf{D}_{\leq m}^{-1}}{n} + \frac{\boldsymbol{\Psi}_{\leq m}^{\mathsf{T}} \mathbf{H}_{>m}^{-1} \boldsymbol{\Psi}_{\leq m}}{n} \right)^{-1} \right\|$$

$$\overset{(i)}{\leq} \frac{m}{n} 4r_2^2 \max\{\lambda,1\}^2 \frac{\|\boldsymbol{\Psi}_{\leq m}^{\mathsf{T}} \boldsymbol{\Psi}_{\leq m}/n\|}{\mu_{\min}(\mathbf{H}_{>m})^2} \overset{(ii)}{\leq} 6\frac{m}{n} r_2^2 \frac{\max\{\lambda,1\}^2}{(\mu_{\min}(\mathbf{K}_{>m}) + \lambda)^2}$$

$$= 6 \left( \frac{r_2}{r_1} \right)^2 \frac{m}{n},$$

and

$$V_1 \geq \frac{m}{n} \mu_{\min} \left( \left( \frac{\mathbf{D}_{\leq m}^{-1}}{n} + \frac{\mathbf{\Psi}_{\leq m}^\mathsf{T} \mathbf{H}_{>m}^{-1} \mathbf{\Psi}_{\leq m}}{n} \right)^{-1} \frac{\mathbf{\Psi}_{\leq m}^\mathsf{T} \mathbf{H}_{>m}^{-2} \mathbf{\Psi}_{\leq m}}{n} \left( \frac{\mathbf{D}_{\leq m}^{-1}}{n} + \frac{\mathbf{\Psi}_{\leq m}^\mathsf{T} \mathbf{H}_{>m}^{-1} \mathbf{\Psi}_{\leq m}}{n} \right)^{-1} \right)$$

$$\overset{(iii)}{\geq} \frac{m}{n} \left( \frac{r_1}{1.5 + r_1} \right)^2 \tau_1^2 \max\{\lambda, 1\}^2 \frac{\mu_{\min}\left( \mathbf{\Psi}_{\leq m}^\mathsf{T} \mathbf{\Psi}_{\leq m}/n \right)}{(\|\mathbf{K}_{>m}\| + \lambda)^2}$$

$$\overset{(iv)}{\geq} \frac{1}{2} \frac{m}{n} \left( \frac{r_1}{1.5 + r_1} \right)^2 \tau_1^2 \frac{\max\{\lambda, 1\}^2}{(\|\mathbf{K}_{>m}\| + \lambda)^2} = \frac{1}{2} \frac{m}{n} \left( \frac{r_1}{r_2(1.5 + r_1)} \right)^2 \tau_1^2.$$

For $(i)$ and $(iii)$, we bound the terms analogously to the upper and lower bound of $B_1$, and use Equation (5) in $(ii)$ and $(iv)$.

Next, the upper bound on $V_2$ follows from a special case of Hölder's inequality as follows:

$$V_2 = \mathbf{Tr}\left( \mathbf{H}^{-1} \mathbf{S}_{>m} \mathbf{H}^{-1} \right) \leq \mathbf{Tr}\left( \mathbf{S}_{>m} \right) \|\mathbf{H}^{-2}\|$$

$$= \frac{\mathbf{Tr}\left( \mathbf{S}_{>m} \right)}{\mu_{\min}\left( \mathbf{K}_{\leq m} + \mathbf{H}_{>m} \right)^2} \leq \frac{\mathbf{Tr}\left( \mathbf{S}_{>m} \right)}{\mu_{\min}\left( \mathbf{H}_{>m} \right)^2} = \frac{\mathbf{Tr}\left( \mathbf{S}_{>m} \right)}{r_1^2 \max\{\lambda, 1\}^2}.$$

For the lower bound, we need a more accurate analysis. First, we apply the identity

$$\mathbf{H}^{-1} = (\mathbf{H}_{>m} + \mathbf{K}_{\leq m})^{-1} = \mathbf{H}_{>m}^{-1} - \underbrace{\mathbf{H}_{>m}^{-1} \mathbf{K}_{\leq m} (\mathbf{H}_{>m} + \mathbf{K}_{\leq m})^{-1}}_{:=\mathbf{A}}, \tag{12}$$

which is valid since $\mathbf{H}$ and $\mathbf{H}_{>m}$ are full rank.

Next, note that the rank of $\mathbf{A}$ is bounded by the rank of $\mathbf{K}_{\leq m}$, which can be written as $\mathbf{\Psi}_{\leq m} \mathbf{D}_{\leq m} \mathbf{\Psi}_{\leq m}^\mathsf{T}$ and therefore has itself rank at most $m$. Furthermore, Equation (5) implies that $\mathbf{\Psi}_{\leq m}^\mathsf{T} \mathbf{\Psi}_{\leq m}$ has full rank, and hence $m \leq n$.

Let now $\{v_1, \ldots, v_{\mathrm{rank}(\mathbf{A})}\}$ be an orthonormal basis of $\mathrm{col}(\mathbf{A})$, and let $\{v_{\mathrm{rank}(\mathbf{A})+1}, \ldots, v_n\}$ be an orthonormal basis of $\mathrm{col}(\mathbf{A})^\perp$. Hence, $\{v_1, \ldots, v_n\}$ is an orthonormal basis of $\mathbb{R}^n$, and similarity invariance of the trace yields

$$V_2 = \mathbf{Tr}\left( \mathbf{H}^{-1} \mathbf{S}_{>m} \mathbf{H}^{-1} \right) = \sum_{i=1}^n v_i^\mathsf{T} \mathbf{H}^{-1} \mathbf{S}_{>m} \mathbf{H}^{-1} v_i$$

$$\geq \sum_{i=\mathrm{rank}(\mathbf{A})+1}^n v_i^\mathsf{T} \mathbf{H}^{-1} \mathbf{S}_{>m} \mathbf{H}^{-1} v_i$$

$$\overset{(i)}{=} \sum_{i=\mathrm{rank}(\mathbf{A})+1}^n v_i^\mathsf{T} (\mathbf{H}_{>m}^{-1} - \mathbf{A}) \mathbf{S}_{>m} (\mathbf{H}_{>m}^{-1} - \mathbf{A}) v_i$$

$$\overset{(ii)}{=} \sum_{i=\mathrm{rank}(\mathbf{A})+1}^n v_i^\mathsf{T} \mathbf{H}_{>m}^{-1} \mathbf{S}_{>m} \mathbf{H}_{>m}^{-1} v_i$$

$$\geq \frac{1}{\|\mathbf{H}_{>m}\|^2} \sum_{i=\mathrm{rank}(\mathbf{A})+1}^n v_i^\mathsf{T} \mathbf{S}_{>m} v_i,$$

$$= \frac{1}{\|\mathbf{H}_{>m}\|^2} \mathbf{Tr}\left( \mathbf{P}^\mathsf{T} \mathbf{S}_{>m} \mathbf{P} \right),$$

where $\mathbf{P}$ is the projection matrix of $\mathbb{R}^n$ onto $\mathrm{col}(\mathbf{A})^\perp$, and $(i)$ follows from Equation (12). Step $(ii)$ uses that, for all $i > \mathrm{rank}(\mathbf{A})$, $v_i$ is orthogonal to the column space of $\mathbf{A}$, and hence

$$v_i^\mathsf{T} (\mathbf{H}_{>m}^{-1} - \mathbf{A}) \mathbf{S}_{>m} (\mathbf{H}_{>m}^{-1} - \mathbf{A}) v_i$$
$$= v_i^\mathsf{T} \mathbf{H}_{>m}^{-1} \mathbf{S}_{>m} \mathbf{H}_{>m}^{-1} v_i - \underbrace{v_i^\mathsf{T} \mathbf{A}}_{=0} \mathbf{S}_{>m} \mathbf{H}_{>m}^{-1} v_i - v_i^\mathsf{T} \mathbf{H}_{>m}^{-1} \mathbf{S}_{>m} \underbrace{\mathbf{A} v_i}_{=0} + \underbrace{v_i^\mathsf{T} \mathbf{A} \mathbf{S}_{>m} \mathbf{A} v_i}_{=0}.$$

Finally, let $\mu_i(\cdot)$ be the $i$-th eigenvalue of its argument with respect to a decreasing order. Then, the Cauchy interlacing theorem yields $\mu_{\mathrm{rank}(\mathbf{A})+i}(\mathbf{S}_{\leq m}) \leq \mu_i(\mathbf{P}^\mathsf{T}\mathbf{S}_{>m}\mathbf{P})$ for all $i = 1, \ldots, n - \mathrm{rank}(\mathbf{A})$. This implies

$$\mathbf{Tr}\left(\mathbf{P}^\mathsf{T}\mathbf{S}_{>m}\mathbf{P}\right) = \sum_{i=1}^{n-\mathrm{rank}(\mathbf{A})} \mu_i(\mathbf{P}^\mathsf{T}\mathbf{S}_{>m}\mathbf{P}) \geq \sum_{i=1}^{n-\mathrm{rank}(\mathbf{A})} \mu_{\mathrm{rank}(\mathbf{A})+i}(\mathbf{S}_{>m})$$

$$= \sum_{i=1+\mathrm{rank}(\mathbf{A})}^{n} \mu_i(\mathbf{S}_{>m}) \overset{(i)}{\geq} \sum_{i=1+m}^{n} \mu_i(\mathbf{S}_{>m}),$$

where $(i)$ uses that the rank of $\mathbf{A}$ is bounded by $m$. This concludes the lower bound on $V_2$ as follows:

$$V_2 = \mathbf{Tr}\left(\mathbf{H}^{-1}\mathbf{S}_{>m}\mathbf{H}^{-1}\right) \geq \frac{\sum_{i=m+1}^{n} \mu_i(\mathbf{S}_{\leq m})}{\|\mathbf{H}_{>m}\|^2} = \frac{\sum_{i=m+1}^{n} \mu_i(\mathbf{S}_{\leq m})}{r_2^2 \max\{\lambda, 1\}^2}.$$

## A.2 TECHNICAL LEMMAS

**Lemma 5** (Squared kernel decomposition). *Let $\mathcal{K}$ be a kernel function that under a distribution $\nu$ can be decomposed as $\mathcal{K}(x, x') = \sum_k \lambda_k \psi_k(x)\psi_k(x')$, where $\mathbb{E}_{x\sim\nu}[\psi_k(x)\psi_{k'}(x)] = \delta_{k,k'}$. Then, the squared kernel $\mathcal{S}(x, x') = \mathbb{E}_{z\sim\nu}[\mathcal{K}(x, z)\mathcal{K}(x', z)]$ can be written as*

$$\mathcal{S}(x, x') = \sum_k \lambda_k^2 \psi_k(x)\psi_k(x'),$$

*and for any $m > 0$, the corresponding kernel matrix can be written as $\mathbf{S} = \mathbf{\Psi}_{\leq m}\mathbf{D}_{\leq m}^2\mathbf{\Psi}_{\leq m}^\mathsf{T} + \mathbf{S}_{>m}$.*

*Proof.* The statement simply follows from

$$\mathcal{S}(x, x') = \sum_{k,k'} \lambda_k \lambda_{k'} \psi_k(x) \underbrace{\mathbb{E}_{z\sim\nu}\left[\psi_k(z)\psi_{k'}(z)\right]}_{\delta_{k,k'}} \psi_{k'}(x') = \sum_k \lambda_k^2 \psi_k(x)\psi_k(x').$$

$\square$

**Lemma 6** (Corollary of Lemma 20 in Bartlett et al. (2020)).

$$\mathbf{D}_{\leq m}\mathbf{\Psi}_{\leq m}^\mathsf{T}\mathbf{H}^{-2}\mathbf{\Psi}_{\leq m}\mathbf{D}_{\leq m} = \left(\mathbf{D}_{\leq m}^{-1} + \mathbf{\Psi}_{\leq m}^\mathsf{T}\mathbf{H}_{>m}^{-1}\mathbf{\Psi}_{\leq m}\right)^{-1}\mathbf{\Psi}_{\leq m}^\mathsf{T}\mathbf{H}_{>m}^{-2}\mathbf{\Psi}_{\leq m}$$

$$\left(\mathbf{D}_{\leq m}^{-1} + \mathbf{\Psi}_{\leq m}^\mathsf{T}\mathbf{H}_{>m}^{-1}\mathbf{\Psi}_{\leq m}\right)^{-1}$$

*Proof.*

$$\mathbf{D}_{\leq m}\mathbf{\Psi}_{\leq m}^\mathsf{T}\mathbf{H}^{-2}\mathbf{\Psi}_{\leq m}\mathbf{D}_{\leq m} = \mathbf{D}_{\leq m}^{1/2}\mathbf{D}_{\leq m}^{1/2}\mathbf{\Psi}_{\leq m}^\mathsf{T}\mathbf{H}^{-2}\mathbf{\Psi}_{\leq m}\mathbf{D}_{\leq m}^{1/2}\mathbf{D}_{\leq m}^{1/2}$$

$$\overset{(i)}{=} \mathbf{D}_{\leq m}^{1/2}\left(\mathbf{I}_m + \mathbf{D}_{\leq m}^{1/2}\mathbf{\Psi}_{\leq m}^\mathsf{T}\mathbf{H}_{>m}^{-1}\mathbf{\Psi}_{\leq m}\mathbf{D}_{\leq m}^{1/2}\right)^{-1}\mathbf{D}_{\leq m}^{1/2}\mathbf{\Psi}_{\leq m}^\mathsf{T}\mathbf{H}_{>m}^{-2}\mathbf{\Psi}_{\leq m}\mathbf{D}_{\leq m}^{1/2}$$

$$\left(\mathbf{I}_m + \mathbf{D}_{\leq m}^{1/2}\mathbf{\Psi}_{\leq m}^\mathsf{T}\mathbf{H}_{>m}^{-1}\mathbf{\Psi}_{\leq m}\mathbf{D}_{\leq m}^{1/2}\right)^{-1}\mathbf{D}_{\leq m}^{1/2}$$

$$= \left(\mathbf{D}_{\leq m}^{-1} + \mathbf{\Psi}_{\leq m}^\mathsf{T}\mathbf{H}_{>m}^{-1}\mathbf{\Psi}_{\leq m}\right)^{-1}\mathbf{\Psi}_{\leq m}^\mathsf{T}\mathbf{H}_{>m}^{-2}\mathbf{\Psi}_{\leq m}\left(\mathbf{D}_{\leq m}^{-1} + \mathbf{\Psi}_{\leq m}^\mathsf{T}\mathbf{H}_{>m}^{-1}\mathbf{\Psi}_{\leq m}\right)^{-1},$$

where $(i)$ applies Lemma 20 from Bartlett et al. (2020). $\square$

**Lemma 7** (Squared kernel tail). *For $m > 0$, let $\mathbf{S}_{>m}$ be the kernel matrix of the truncated squared kernel $\mathcal{S}_{>m} = \sum_{k>m} \lambda_k^2 \psi_k(x)\psi_k(x')$, and let $\mathbf{K}_{>m}$ be the kernel matrix of the truncated original kernel $\mathcal{K}_{>m} = \sum_{k>m} \lambda_k \psi_k(x)\psi_k(x')$. Then,*

$$\|\mathbf{S}_{>m}\| \leq \lambda_{m+1}\|\mathbf{K}_{>m}\|.$$

*Proof.* We show that for any vector $v$, $v^\mathsf{T} \mathbf{S}_{>m} v \le \lambda_{m+1} v^\mathsf{T} \mathbf{K}_{>m} v$, which implies the claim. To do so, we define $\mathbf{\Psi}_k \in \mathbb{R}^{n \times n}$ with $[\mathbf{\Psi}_k]_{i,j} = \psi_k(x_i)\psi_k(x_j)$ for all $k > m$. Then we can write $\mathbf{K}_{>m} = \sum_{k>m} \lambda_k \mathbf{\Psi}_k$ and $\mathbf{S}_{>m} = \sum_{k>m} \lambda_k^2 \mathbf{\Psi}_k$. Since the eigenvalues are in decreasing order, we have $\lambda_k \le \lambda_{m+1}$ for any $k > m$, and thus

$$v^\mathsf{T} \mathbf{S}_{>m} v = \sum_{k>m} \lambda_k^2 v^\mathsf{T} \mathbf{\Psi}_k v \le \lambda_{m+1} \sum_{k>m} \lambda_k v^\mathsf{T} \mathbf{\Psi}_k v = \lambda_{m+1} v^\mathsf{T} \mathbf{K}_{>m} v.$$

$\square$

## B  CONVOLUTIONAL KERNELS ON THE HYPERCUBE

First, Appendix B.1 provides a way to decompose general functions for features distributed uniformly on the hypercube. Next, Appendix B.2 uses those results to characterize the eigenfunctions and eigenvalues of cyclic convolutional kernels. Finally, Appendices B.3 and B.4 apply this characterization to prove Lemmas 2 and 4, respectively.

### B.1  GENERAL FUNCTIONS ON THE HYPERCUBE

This subsection focuses on the main setting in our paper: the hypercube domain $\{-1, 1\}^d$ together with the uniform probability distribution, previously studied in Misiakiewicz & Mei (2021). For any $S \subseteq \{1, \ldots, d\}$, we define the polynomial

$$\mathcal{Y}_S(x) := \prod_{j \in S} [x]_j \tag{13}$$

of degree $|S|$, where $[x]_j$ is the $j$-th entry of $x \in \{-1, 1\}^d$. It is easy to see that $\{\mathcal{Y}_S\}_{S \subseteq \{1, \ldots, d\}}$ is set of orthonormal functions with respect to the inner product $\langle f, g \rangle_{\{-1,1\}^d} := \mathbb{E}_{x \sim \mathcal{U}(\{-1,1\}^d)}[f(x)g(x)]$. Those functions play a key role in the remainder of our proof; as it turns out, they are the eigenfunctions of the kernel in Section 3.1. Towards a formal statement, define the polynomials

$$\mathcal{G}_l^{(d)}\left(\frac{\langle x, x' \rangle}{\sqrt{d}}\right) := \frac{1}{\mathcal{B}(l, d)} \sum_{|S|=l} \mathcal{Y}_S(x)\mathcal{Y}_S(x'), \tag{14}$$

$$\text{where } \mathcal{B}(l, d) := |\{S \subseteq \{1, \ldots, d\} \mid |S| = l\}| = \binom{d}{l}. \tag{15}$$

Note that $\mathcal{G}_l^{(d)}$ only depends on the (Euclidean) inner product of $x$ and $x'$. Furthermore, $\{\mathcal{G}_l^{(d)}\}_{l=0}^d$ is a set of orthonormal polynomials with respect to the distribution of $\langle x, x' \rangle / \sqrt{d}$. The following lemma shows how such polynomials form an eigenbasis for functions that only depend on the inner product between points in the unit hypercube.

**Lemma 8** (Local kernel decomposition). *Let $\kappa \colon \mathbb{R} \to \mathbb{R}$ be any function and $d \in \mathbb{N}_{>0}$. Then, for any $x, x' \in \{-1, 1\}^d$, we can decompose $\kappa(\langle x, x' \rangle / d)$ as*

$$\kappa\left(\frac{\langle x, x' \rangle}{d}\right) = \sum_{l=0}^d \xi_l^{(d)} \frac{1}{\mathcal{B}(l, d)} \sum_{|S|=l} \mathcal{Y}_S(x)\mathcal{Y}_S(x'). \tag{16}$$

*Proof.* Note that the decomposition only needs to hold at the evaluation of $\kappa$ in the values that $\langle x, x' \rangle / d$ can take, that is, $\kappa$ computed in $\{-1, -1 + 2/d, \ldots, -2/d + 1, 1\}$. Since that set has a cardinality $d + 1$, we can write $\kappa$ as a linear combination of $d + 1$ uncorrelated functions. In particular, $\{\mathcal{G}_l^{(d)}\}_{l=0}^d$ is a set of such functions with respect to the distribution of $\langle x, x' \rangle / \sqrt{d}$, and hence

$$\kappa\left(\frac{\langle x, x' \rangle}{d}\right) = \sum_{l=0}^d c_l \mathcal{G}_l^{(d)}\left(\frac{\langle x, x' \rangle}{\sqrt{d}}\right)$$

for some (unknown) coefficients $c_l$. Finally, the proof follows by expanding the definition of $\mathcal{G}_l^{(d)}$ in Equation (14) and choosing $\xi_l^{(d)} = c_l$. $\square$

### B.2 CONVOLUTIONAL KERNELS ON THE HYPERCUBE

While the previous subsection considers general functions on the hypercube, we now focus on convolutional kernels and their eigenvalues. This yields the tools to prove Lemmas 2 and 4. In order to characterize eigenvalues, we first follow existing literature such as Misiakiewicz & Mei (2021) and introduce useful quantities.

Let $S \subseteq \{1, \ldots, d\}$. The diameter of $S$ is

$$\gamma(S) := \max_{i,j \in S} \min \{\mathrm{mod}\, (j - i, d) + 1, \mathrm{mod}\, (i - j, d) + 1\}$$

for $S \neq \emptyset$, and $\gamma(\emptyset) = 0$. Furthermore, we define

$$\mathcal{C}(l, q, d) := |\{S \subseteq \{1, \ldots, d\} \mid |S| = l, \gamma(S) \leq q\}|. \tag{17}$$

Intuitively, the diameter of $S$ is the smallest number of contiguous feature indices that fully contain $S$. The following lemma yields an explicit formula for $\mathcal{C}(l, q, d)$, that is, the number of sets of size $l$ with diameter at most $q$.

**Lemma 9** (Number of overlapping sets). *Let $l, q, d \in \mathbb{N}$ with $l \leq q < d/2$. Then,*

$$\mathcal{C}(l, q, d) = \begin{cases} d\binom{q-1}{l-1} & l > 0, \\ 1 & l = 0. \end{cases}$$

*Proof.* Since the result holds trivially for $l = 0$ and $l = 1$, we henceforth focus on $l \geq 2$. Let $\tilde{\mathcal{C}}(l, \gamma, d)$ be the number of subsets $S \subseteq \{1, \ldots, d\}$ of cardinality $|S| = l$ with diameter exactly $\gamma(S) = \gamma$. First, consider $\tilde{\mathcal{C}}(2, \gamma, d)$. For each set, we can choose the first element $i$ from $d$ different values, and the second as $\mathrm{mod}\,((i - 1) \pm (\gamma - 1), d) + 1$. In this way, since $q < d/2$, we count each set exactly twice. Thus,

$$\tilde{\mathcal{C}}(2, \gamma, d) = \frac{d \cdot 2}{2} = d.$$

Next, consider $l > 2$. We can build all the possible sets by starting with one of the $\tilde{\mathcal{C}}(2, \gamma, d) = d$ sets, and adding the remaining $l - 2$ elements from $\gamma - 2$ possible indices. Hence, every fixed set of size 2 and diameter $\gamma$ yields $\binom{\gamma-2}{l-2}$ different sets of size $l$. Furthermore, by construction, every set of size $l$ and diameter $\gamma$ results from exactly one set of size 2 and diameter $\gamma$. Therefore,

$$\tilde{\mathcal{C}}(l, \gamma, d) = d\binom{\gamma - 2}{l - 2}.$$

The result for $l \geq 2$ then follows from summing $\tilde{\mathcal{C}}(l, \gamma, d)$ over all diameters $\gamma \leq q$:

$$\mathcal{C}(l, q, d) = d \sum_{\gamma=l}^{q} \binom{\gamma - 2}{l - 2} \overset{(i)}{=} d\binom{q - 1}{l - 1},$$

where $(i)$ follows from the hockey-stick identity. $\square$

Now, we focus on cyclic convolutional kernels $\mathcal{K}$ as in Equation (1). First, we restate Proposition 1 from Misiakiewicz & Mei (2021). This proposition establishes that $\mathcal{Y}_S$ for $S \subseteq \{1, \ldots, d\}$ are indeed the eigenfunctions of $\mathcal{K}$, and yields closed-form eigenvalues $\lambda_S$ up to factors $\xi_{|S|}^{(q)}$ that depend on the inner nonlinearity $\kappa$. Next, Lemma 10 uses additional regularity assumptions on $\kappa$ to eliminate the dependency on $\xi_{|S|}^{(q)}$. This characterization of the eigenvalues then enables the proof Lemmas 2 and 4.

**Proposition 1** (Proposition 1 from Misiakiewicz & Mei (2021)). *Let $\mathcal{K}$ be a cyclic convolutional kernel over the unit hypercube as defined in Equation (1). Then,*

$$\mathcal{K}(x, x') := \frac{1}{d} \sum_{k=1}^{d} \kappa\left(\frac{\langle x_{(k,q)}, x'_{(k,q)}\rangle}{q}\right) = \sum_{l=0}^{q} \sum_{\substack{\gamma(S) \leq q \\ |S| = l}} \lambda_S \mathcal{Y}_S(x) \mathcal{Y}_S(x'),$$

*with*

$$\lambda_S = \xi_{|S|}^{(q)} \frac{q + 1 - \gamma(S)}{d\mathcal{B}(|S|, q)}$$

*where $\xi_{|S|}^{(q)}$ are the coefficients of the $\kappa$-decomposition (Equation (16)) over $\{-1, 1\}^q$. Alternatively,*

$$\mathcal{K}(x, x') = \sum_k \lambda_{S_k} \mathcal{Y}_{S_k}(x) \mathcal{Y}_{S_k}(x')$$

*where we order all $S_k \subseteq \{1, \ldots, d\}$ with $\gamma(S_k) \leq q$ such that $\lambda_{S_k} \geq \lambda_{S_{k+1}}$. In particular, $\lambda_{S_k} = \lambda_k$.*

We refer to Proposition 1 from Misiakiewicz & Mei (2021) for a formal proof. Intuitively, the result follows from applying Lemma 8 for each subset $S \subseteq \{1, \ldots, d\}$ of contiguous elements. This is possible because crucially, any subset of $q$ features is again distributed uniformly on the $q$-dimensional unit hypercube. Lastly, the factor $q + 1 - \gamma(S)$ stems from the fact that each eigenfunction $\mathcal{Y}_S$ appears as many times as there are contiguous index sets of size $\gamma(S)$ supported in a fixed contiguous index set of size $q$. In other words, the term is the number of shifted instances of $S$ supported in a contiguous subset of $q$ features.

As mentioned before, Proposition 1 characterizes the eigenvalues of cyclic convolutional kernels $\mathcal{K}$ up to factors $\xi_{|S|}^{(q)}$ that depend on the inner nonlinearity $\kappa$. To avoid the additional factors, we require the following regularity assumptions:

**Assumption 1** (Regularity). *Let $T := \left\lceil 4 + \frac{4\ell}{\beta} \right\rceil$. A cyclic convolutional kernel $\mathcal{K}(x, x') = \frac{1}{d} \sum_{k=1}^d \kappa \left( \frac{\langle x_{(k,q)}, x'_{(k,q)} \rangle}{q} \right)$ from the setting of Section 3.1 with inner function $\kappa$ satisfies the regularity assumption if there exist constants $c \geq T$, $c', c'' > 0$ and a series of constants $\{c_l > 0\}_{l=0}^T$ such that, for any $q \geq c$, the decomposition*

$$\mathcal{K}(x, x') = \sum_{l=0}^q \sum_{\substack{\gamma(S) \leq q \\ |S|=l}} \xi_{|S|}^{(q)} \frac{q + 1 - \gamma(S)}{d\mathcal{B}(|S|, q)} \mathcal{Y}_S(x) \mathcal{Y}_S(x')$$

*from Proposition 1 over inputs $x, x' \in \{-1, 1\}^d$ satisfies*

$$\xi_l^{(q)} \geq c_l \quad \forall l \in \{0, \ldots, T\}, \tag{18}$$

$$\xi_l^{(q)} \geq 0 \quad \forall l > T, \tag{19}$$

$$\xi_{q-l}^{(q)} \leq \frac{c'}{q^{T-l+1}} \quad \forall l \in \{0, \ldots, T\}, \tag{20}$$

$$\sum_{l \geq 0} \xi_l^{(q)} \leq c''. \tag{21}$$

For sufficiently high-dimensional inputs $x, x'$, Equations (18) and (19) ensure that the convolutional kernel $\mathcal{K}(x, x')$ in Equation (1) is a valid kernel, and that it can learn polynomials of degree up to $T$. Indeed, if $\xi_l^{(q)} = 0$ for some $l$, then there are no polynomials of degree $l$ among the eigenfunctions of $\mathcal{K}$. Furthermore, Equations (18), (20) and (21) guarantee that the eigenvalue tail is sufficiently bounded. This allows us to bound $\|\mathbf{K}_{>m}\|$ and $\mu_{\min}(\mathbf{K}_{>m})$ in Appendix C.

Our assumption resembles Assumption 1 by Misiakiewicz & Mei (2021): For one, Equations (20) and (21) are equivalent to Equations 43 and 44 in Misiakiewicz & Mei (2021). Furthermore, Equation (18) above is a slightly stronger version of Equation 42, where strengthening is necessary due to the non-asymptotic nature of our results.

We still argue that many standard $\kappa$, for example, the Gaussian kernel, satisfy Assumption 1 with our convolution kernel $\mathcal{K}$. Because such $\kappa$ satisfy Assumption 1 in Misiakiewicz & Mei (2021), we only need to check that they additionally satisfy our Equation (18). If $\kappa$ is a smooth function, we have $\xi_l^{(q)} = \kappa^{(l)}(0) + o(1)$ for all $l \leq T$, where $\kappa^{(l)}$ is the $l$-th derivative of $\kappa$. In particular, all derivatives of the exponential function at 0 are strictly positive, implying Equation (18) for the Gaussian kernel if $d$ is large enough.

The final lemma of this section is a corollary that characterizes the eigenvalues solely in terms of $|S|$, and further shows that, for $d$ large enough, the eigenvalues decay as $|S|$ grows.

**Lemma 10** (Corollary of Proposition 1). *Consider a cyclic convolutional kernel as in Proposition 1 that satisfies Assumption 1 with $q \in \Theta(d^\beta)$ for some $\beta \in (0, 1)$. Then, for any $S \subseteq \{1, \ldots, d\}$ such that $\gamma(S) \leq q$ and $|S| < T$, the eigenvalue $\lambda_S$ corresponding to the eigenfunction $\mathcal{Y}_S(x)$ satisfies*

$$\lambda_S \in \Omega\left(\frac{1}{d \cdot q^{|S|}}\right) \quad \text{and} \quad \lambda_S \in \mathcal{O}\left(\frac{1}{d \cdot q^{|S|-1}}\right). \tag{22}$$

*Furthermore,*

$$\max_{\substack{|S| \geq T \\ \gamma(S) \leq q}} \lambda_S \in \mathcal{O}\left(\frac{1}{d \cdot q^{T-1}}\right). \tag{23}$$

*Proof.* Without loss of generality, assume $d$ is large enough such that $d > q/2 \geq c/2$, where $c$ is a constant from Assumption 1.

Let $S \subseteq \{1, \ldots, d\}$ with $\gamma(S) \leq q$ be arbitrary and define $l := |S|$ and $r := q + 1 - \gamma(S)$. Since $l \leq \gamma(S) \leq q$, we have

$$1 \leq r \leq q + 1 - l.$$

Furthermore, since $\mathcal{B}(l, q) = \binom{q}{l}$, we use the following classical bound on the binomial coefficient throughout the proof:

$$\left(\frac{1}{l}\right)^l q^l \leq \mathcal{B}(l, q) \leq \left(\frac{e}{l}\right)^l q^l.$$

For the first part of the lemma, assume $|S| < T$. Then, using Assumption 1, we have

$$\lambda_S = \frac{\xi_l^{(q)} r}{d\mathcal{B}(l, q)} \overset{(i)}{\leq} \frac{l^l}{q^l} \frac{c'' r}{d} \leq c_{l,1} \frac{1}{dq^{l-1}} \leq c_{T,1} \frac{1}{dq^{l-1}},$$

$$\overset{(ii)}{\geq} \frac{l^l}{q^l e^l} \frac{c_l r}{d} \geq c_{l,2} \frac{1}{dq^l} \geq c_{T,2} \frac{1}{dq^l}$$

for some positive constants $c_{l,1}, c_{l,2}$ that depend on $l$, $c_{T,1} := \max_{l \in \{0,\ldots,T-1\}} c_{l,1}$, and $c_{T,2} := \min_{l \in \{0,\ldots,T-1\}} c_{l,2}$. Step $(i)$ follows from the upper bound in Equation (21) with non-negativity in Equations (18) and (19), and $(ii)$ follows from the lower bound in Equation (18). Since $c_{T,1}$ and $c_{T,2}$ do not depend on $l$, this concludes the first part of the proof.

For the second part of the proof, we consider two cases depending on whether $|S| \in [T, q - T]$ or $|S| > q - T$.

Hence, first assume $T \leq |S| \leq q - T$. Then,

$$\lambda_S \overset{(i)}{=} \xi_{|S|} \frac{q + 1 - \gamma(S)}{d\binom{q}{|S|}} \overset{(ii)}{\leq} \xi_{|S|} \frac{q + 1 - \gamma(S)}{d\binom{q}{T}} \overset{(iii)}{\leq} c'' \frac{q}{d\left(\frac{q}{T}\right)^T} = \frac{c'' T^T}{dq^{T-1}}, \tag{24}$$

where $(i)$ follows from Proposition 1. In step $(ii)$, we use that $\binom{q}{|S|}$ is minimized when $|S|$ is has the largest difference to $q/2$. Lastly, step $(iii)$ applies the upper bound from Equation (21) together with non-negativity of $\xi_{|S|}$ in Assumption 1, and the classical bound on the binomial coefficient.

Now assume $|S| > q - T$. Then,

$$\lambda_S \overset{(i)}{=} \xi_{|S|} \frac{q + 1 - \gamma(S)}{d\binom{q}{|S|}} \leq \xi_{q-(q-|S|)} \frac{q}{d\binom{q}{q-(q-|S|)}}$$

$$\overset{(ii)}{=} \xi_{q-(q-|S|)} \frac{q}{d\binom{q}{q-|S|}}$$

$$\overset{(iii)}{\leq} \xi_{q-(q-|S|)} \frac{(q - |S|)^{q-|S|} q}{dq^{q-|S|}}$$

$$\overset{(iv)}{\leq} c' \frac{(q - |S|)^{q-|S|} q}{dq^{T-(q-|S|)+1} q^{q-|S|}} \overset{(v)}{\leq} \frac{c' T^T}{dq^T}, \tag{25}$$

where $(i)$ follows from Proposition 1, $(ii)$ from the fact that $\binom{n}{n-k} = \binom{n}{k}$, $(iii)$ from the classical bound for the binomial coefficient, $(iv)$ from Equation (20) in Assumption 1, and $(v)$ from the fact that $q - |S| < T$ in the current case.

Combining Equations (24) and (25) from the two cases finally yields

$$
\max_{\substack{|S| \geq T \\ \gamma(S) \leq q}} \lambda_S \leq \max \left\{ \max_{\substack{T \leq |S| \leq q-T \\ \gamma(S) \leq q}} \lambda_S, \max_{\substack{|S| > q-T \\ \gamma(S) \leq q}} \lambda_S \right\}
$$
$$
\leq \max \left\{ \frac{c'' T^T}{dq^{T-1}}, \frac{c' T^T}{dq^T} \right\} \in \mathcal{O}\left( \frac{1}{dq^{T-1}} \right),
$$

which concludes the second part of the proof. □

### B.3 PROOF OF LEMMA 2

First, note that $f^\star = \mathcal{Y}_{S^*}$ for $S^* = \{1, \ldots, L^*\}$. Since $|S^*| = \gamma(S^*) = L^*$, Proposition 1 yields

$$
\lambda_{S^*} = \xi_{L^*}^{(q)} \frac{q + 1 - L^*}{d\mathcal{B}(L^*, q)} \overset{(i)}{\in} \Theta\left( \frac{q}{dq^{L^*}} \right) = \Theta\left( \frac{1}{dq^{L^*-1}} \right) \overset{(ii)}{\subseteq} \omega(\lambda_m),
$$

where $(i)$ uses Equations (18) and (21) in Assumption 1 for $L^* \leq T$ and $d$ large enough to get $\xi_{L^*}^{(q)} \in \Theta(1)$, and $(ii)$ uses $\lambda_m \in o\left( \frac{1}{dq^{L^*-1}} \right)$. Hence, for $d$ sufficiently large, $\lambda_{S^*} > \lambda_m$. Since the eigenvalues are in decreasing order, this implies that $f^\star$ is in the span of the first $m$ eigenfunctions. This further yields

$$
\|\mathbf{D}_{\leq m}^{-1} a\| = \lambda_{S^*}^{-1} \in \Theta\left( dq^{L^*-1} \right),
$$

since the entry of $a$ corresponding to $\mathcal{Y}_{S^*}$ is 1 while all others are 0.

### B.4 PROOF OF LEMMA 4

Before proving Lemma 4, we introduce the following quantity:

$$
L := \left\lfloor \frac{\ell - \ell_\lambda - 1}{\beta} \right\rfloor. \tag{26}
$$

Intuitively, $L$ corresponds to the degree of the largest polynomial that a cyclic convolutional kernel as defined in Equation (1) can learn. This quantity plays a key role throughout the proof of Lemma 4, and Lemma 3 later. Finally, note that $\delta$ as defined in Theorem 1 can be written as $\delta = \frac{\ell - \ell_\lambda - 1}{\beta} - L$.

*Proof of Lemma 4.* First, we use Proposition 1 to write the cyclic convolutional kernel as

$$
\mathcal{K}(x, x') = \sum_{l=0}^{q} \sum_{\substack{\gamma(S) \leq q \\ |S|=l}} \lambda_S \mathcal{Y}_S(x) \mathcal{Y}_S(x') = \sum_k \lambda_{S_k} \mathcal{Y}_{S_k}(x) \mathcal{Y}_{S_k}(x')
$$

where the $\lambda_{S_k}$ are ordered such that $\lambda_{S_{k+1}} \leq \lambda_{S_k}$.

For the first part of the proof, we need to pick an $m \in \mathbb{N}$ such that $n\lambda_m \in \Theta\left(\max\{\lambda, 1\}\right)$ and $m \in \Theta\left( \frac{nq^{-\delta}}{\max\{\lambda, 1\}} \right)$. We will equivalently choose $m \in \Theta(dq^L)$ with $\lambda_m \in \Theta\left( \frac{1}{d \cdot q^{L+\delta}} \right)$; since $n \in \Theta(d^\ell)$ and $\max\{\lambda, 1\} \in \Theta(d^{\ell_\lambda})$, we have

$$
\Theta\left( \frac{\max\{\lambda, 1\}}{n} \right) = \Theta\left( \frac{d^{\ell_\lambda}}{d^\ell} \right) = \Theta\left( \frac{d^{\ell_\lambda}}{d^{1+\ell_\lambda+\beta(L+\delta)}} \right) = \Theta\left( \frac{1}{d \cdot q^{L+\delta}} \right),
$$
$$
\Theta\left( \frac{nq^{-\delta}}{\max\{\lambda, 1\}} \right) = \Theta\left( \frac{d^\ell q^{-\delta}}{d^{\ell_\lambda}} \right) = \Theta\left( \frac{d^{1+\ell_\lambda+\beta(L+\delta)} q^{-\delta}}{d^{\ell_\lambda}} \right) = \Theta(dq^L).
$$

The remainder of the proof proceeds in five steps: we first construct a candidate $S_m \subseteq \{1, \ldots, d\}$ with $\gamma(S_m) \leq q$, show that the rate of the eigenvalue corresponding to $\mathcal{Y}_{S_m}$ satisfies $\lambda_{S_m} = \lambda_m \in \Theta\left( \frac{1}{d \cdot q^{L+\delta}} \right)$, show that the rate of $m \in \Theta(dq^L)$, establish $\Theta\left( \frac{nq^{-\delta}}{\max\{\lambda, 1\}} \right) \subseteq \mathcal{O}(n \cdot q^{-\bar{\delta}})$, and finally show that $\lambda_m \in o\left( \frac{1}{d \cdot q^{L^*-1}} \right)$ for appropriate $L^*$.

**Construction of $m$** We consider two different $S_m$ depending on $\delta$:

$$S_m = \begin{cases} \{1, \ldots, L, \lfloor q + 1 - q^{1-\delta} \rfloor\} & \delta \in (0, 1) \\ \{1, \ldots, L, \lfloor q/2 \rfloor\} & \delta = 0. \end{cases} \tag{27}$$

For $d$—and hence $q \in \Theta(d^\beta)$—large enough, $S_m$ is well-defined, $|S_m| = L + 1$, and the diameter is

$$\gamma(S_m) = \begin{cases} \lfloor q + 1 - q^{1-\delta} \rfloor & \delta \in (0, 1) \\ \lfloor q/2 \rfloor & \delta = 0. \end{cases}$$

For the rest of the proof, assume that $d$ is sufficiently large.

**Rate of $\lambda_{S_m}$** Using Proposition 1 and $|S_m| = L + 1$, we can write

$$\lambda_{S_m} = \xi_{L+1}^{(q)} \frac{q + 1 - \gamma(S_m)}{d\mathcal{B}(L+1, q)}.$$

First, we show that the numerator is in $\Theta\left(q^{1-\delta}\right)$ for both definitions of $S_m$. In the case where $\delta \in (0, 1)$, we have

$$q + 1 - \lfloor q + 1 - q^{1-\delta} \rfloor = -\lfloor -q^{1-\delta} \rfloor = \lceil q^{1-\delta} \rceil \overset{(i)}{\in} \Theta\left(q^{1-\delta}\right),$$

where $(i)$ follows from $\delta < 1$ and $q$ sufficiently large. In the case where $\delta = 0$, we have

$$q + 1 - \lfloor q/2 \rfloor \leq q + 1 \in \mathcal{O}(q),$$
$$q + 1 - \lfloor q/2 \rfloor \geq q/2 \in \Omega(q).$$

Thus, since $\delta = 0$ in this case, the numerator is in $\Theta(q) = \Theta(q^{1-\delta})$.

As the denominator does not depend on $\delta$, we use the same technique for both $\delta = 0$ and $\delta \in (0, 1)$. The classical bound on $\mathcal{B}(L+1, q) = \binom{q}{L+1}$ yields

$$q^{L+1} \lesssim \left(\frac{q}{L+1}\right)^{L+1} \leq \binom{q}{L+1} \leq e^{L+1} \left(\frac{q}{L+1}\right)^{L+1} \lesssim q^{L+1}.$$

Therefore, $d\mathcal{B}(L+1, q) \in \Theta\left(dq^{L+1}\right)$.

Finally, since $L + 1 \leq T = \lceil 4 + 4\ell/\beta \rceil$, we have $\xi_{L+1}^{(q)} \in \Theta(1)$ by Equations (18) and (21) in Assumption 1 for $d$ sufficiently large. Combining all results then yields the desired rate of $\lambda_{S_m}$ as follows:

$$\lambda_{S_m} \in \Theta\left(\frac{q^{1-\delta}}{dq^{L+1}}\right) = \Theta\left(\frac{1}{d \cdot q^{L+\delta}}\right). \tag{28}$$

**Rate of $m$** To establish $m \in \Theta(dq^L)$, we bound $m$ individually from above and below.

*Upper bound:* Since the eigenvalues are in decreasing order, we can bound $m$ from above by counting how many eigenvalues are larger than $\lambda_m$. To do so, we use $|S_m| = L + 1$, and show that for $d$ sufficiently large, all $S_k$ with $|S_k| > L + 1$ correspond to eigenvalues $\lambda_{S_k} < \lambda_{S_m}$. We first decompose

$$\max_{\substack{k:|S_k|>L+1 \\ \gamma(S_k)\leq q}} \lambda_{S_k} = \max\left\{ \underbrace{\max_{\substack{k:L+1<|S_k|<T \\ \gamma(S_k)\leq q}} \lambda_{S_k}}_{=:M_1}, \underbrace{\max_{\substack{k:|S_k|\geq T \\ \gamma(S_k)\leq q}} \lambda_{S_k}}_{=:M_2} \right\}.$$

For $M_1$, let $k$ with $L + 1 < |S_k| < T$ be arbitrary. Then,

$$\lambda_{S_k} \overset{(i)}{\in} \mathcal{O}\left(\frac{1}{dq^{|S_k|-1}}\right) \subseteq \mathcal{O}\left(\frac{1}{dq^{(L+2)-1}}\right) = \mathcal{O}\left(\frac{1}{dq^{L+1}}\right) \overset{(ii)}{\subseteq} o(\lambda_{S_m}),$$

where we apply Equation (22) from Lemma 10 in $(i)$, and use Equation (28) with $\delta < 1$ in $(ii)$. This implies $M_1 \in o(\lambda_{S_m})$.

For $M_2$, we directly get

$$M_2 = \max_{\substack{k:|S_k| \geq T \\ \gamma(S_k) \leq q}} \lambda_{S_k} \overset{(i)}{\in} \mathcal{O}\left(\frac{1}{dq^{T-1}}\right) \overset{(ii)}{\subseteq} \mathcal{O}\left(\frac{1}{dq^{(L+2)-1}}\right) = \mathcal{O}\left(\frac{1}{dq^{L+1}}\right) \overset{(iii)}{\subseteq} o(\lambda_{S_m}),$$

where we apply Equation (23) from Lemma 10 in $(i)$, $(ii)$ follows from $L + 2 \leq T$, and step $(iii)$ uses Equation (28) with $\delta < 1$.

Combined, we have $\max_{k:|S_k|>L+1, \gamma(S_k) \leq q} \lambda_{S_k} = \max\{M_1, M_2\} \in o(\lambda_{S_m})$. Thus, for $d$ sufficiently large and $|S_k| > L + 1$, we have $\lambda_{S_k} < \lambda_{S_m}$. For this reason, $m$ is at most the number of eigenfunctions with degree no larger than $L + 1$:

$$m \leq \sum_{l=0}^{L+1} \mathcal{C}(l, q, d) \overset{(i)}{\in} \mathcal{O}(dq^L),$$

where $(i)$ uses Lemma 9 for $d$ large enough with $\binom{q}{l} \in \Theta(q^l)$.

*Lower bound:* By construction of $S_m$ in Equation (27), we have $\gamma(S_m) \geq \lfloor q/2 \rfloor$. This, combined with Proposition 1, implies that the indices of all polynomials with degree $L + 1$ but diameter at most $\lfloor q/2 \rfloor - 1$ are smaller than $m$. Hence, for large enough $d$, Lemma 9 yields the following lower bound:

$$m \geq \mathcal{C}(L+1, \lfloor q/2 \rfloor - 1, d) = d\binom{\lfloor q/2 \rfloor - 2}{L} \geq d\left(\frac{\lfloor q/2 \rfloor - 2}{L}\right)^L \in \Omega(dq^L).$$

The upper and lower bound together then imply $m \in \Theta(dq^L)$. This concludes the existence of an $m \in \mathbb{N}$ such that $\lambda_m$ and $m$ exhibit the desired rates.

**Rate of $m$ with respect to $n$**    We can write $n$ as

$$n \in \Theta(d^\ell) = \Theta\left(d \cdot d^{\beta \frac{\ell-1}{\beta}}\right) = \Theta\left(dq^{\lfloor \frac{\ell-1}{\beta} \rfloor + \left(\frac{\ell-1}{\beta} - \lfloor \frac{\ell-1}{\beta} \rfloor\right)}\right) = \Theta\left(dq^{\lfloor \frac{\ell-1}{\beta} \rfloor + \bar{\delta}}\right).$$

Combining this with $L \leq \left\lfloor \frac{\ell-1}{\beta} \right\rfloor$, we directly get $\Theta(dq^L) \subseteq \mathcal{O}(dq^{\lfloor \frac{\ell-1}{\beta} \rfloor}) = \mathcal{O}(nq^{-\bar{\delta}})$.

**Rate of $\lambda_m$ for appropriate $L^*$**    Since $n\lambda_m \in \Theta(\max\{\lambda, 1\})$, we have

$$\lambda_m \in \Theta\left(\frac{\max\{\lambda, 1\}}{n}\right) \overset{(i)}{=} \Theta\left(\frac{d^{\ell_\lambda}}{d \cdot d^{\ell_\lambda} q^{L+\delta}}\right) = \Theta\left(\frac{1}{dq^{L+\delta}}\right),$$

where $(i)$ uses the identity $\ell = 1 + \ell_\lambda + \beta(L + \delta)$. Assume now $L^* \leq \left\lceil \frac{\ell - \ell_\lambda - 1}{\beta} \right\rceil$. For the remainder, we need to consider two cases depending on $\delta$.

If $\delta > 0$, then $L^* \leq L + 1$, and we have

$$\lambda_m \in \mathcal{O}\left(\frac{q^{-\delta}}{dq^{L^*-1}}\right) \subseteq o\left(\frac{1}{dq^{L^*-1}}\right).$$

If $\delta = 0$, then $L^* \leq L$, and we have

$$\lambda_m \in \mathcal{O}\left(\frac{q^{-\delta}}{dq^L}\right) \subseteq \mathcal{O}\left(\frac{1}{dq^{L^*}}\right) \subseteq o\left(\frac{1}{dq^{L^*-1}}\right).$$

In both cases, $\lambda_m \in o\left(\frac{1}{dq^{L^*-1}}\right)$, concluding the proof. $\qquad\square$

## C    MATRIX CONCENTRATION

This section considers the random matrix theory part of our main result. First, Appendix C.1 focuses on the large eigenvalues of our kernel, and proves Lemma 1. Next, Appendices C.2 and C.3 focus on the tail of the eigenvalues, culminating in the proof of Lemma 3. Lastly, Appendices C.4 and C.5 establishes some technical tools that we use throughout the proofs.

## C.1  Proof of Lemma 1

In this proof, we show that the matrix $\mathbf{\Psi}_{\leq m}^{\mathsf{T}}\mathbf{\Psi}_{\leq m}/n$ concentrates around the identity matrix for all $m \in \mathcal{O}(nd^{-\beta\bar{\delta}})$, thereby establishing Equation (5). Let $\hat{m}$ be the largest $m \in \mathcal{O}(nq^{-\bar{\delta}})$. The proof consists of applying Theorem 5.44 from Vershynin (2012) to the matrix $\mathbf{\Psi}_{\leq \hat{m}}$, and extending the result to all suitable choices of $m$ simultaneously.

More precisely, let $\tilde{c}$ be the implicit constant of the $\mathcal{O}(nq^{-\bar{\delta}})$-notation, and define $\hat{m}$ to be the largest $m \in \mathbb{N}$ with $m \leq \tilde{c} \cdot nq^{-\bar{\delta}}$. Note that $\hat{m}$ exists, because $d$ is large enough and fixed.

**Bound for $\hat{m}$**  To apply Theorem 5.44 from Vershynin (2012), we need to verify the theorem's conditions on the rows of $\mathbf{\Psi}_{\leq \hat{m}}$. In particular, we show that the rows are independent, have a common second moment matrix, and that their norm is bounded. Let $[\mathbf{\Psi}_{\leq \hat{m}}]_{i,:}$ indicate the $i$-th row of $\mathbf{\Psi}_{\leq \hat{m}} \in \mathbb{R}^{n \times \hat{m}}$. We may write each row entry-wise as

$$[\mathbf{\Psi}_{\leq \hat{m}}]_{i,:} = [\mathcal{Y}_1(x_i)\ \mathcal{Y}_2(x_i)\ \cdots\ \mathcal{Y}_{\hat{m}}(x_i)]^{\mathsf{T}}.$$

First, the rows of $\mathbf{\Psi}_{\leq \hat{m}}$ are independent, since each row depends on a different $x_i$, and we assume the data to be i.i.d..

Second, since the eigenfunctions are orthonormal w.r.t. the data distribution, the second moment of the rows is $\mathbb{E}\left[[\mathbf{\Psi}_{\leq \hat{m}}]_{i,:}[\mathbf{\Psi}_{\leq \hat{m}}]_{i,:}^{\mathsf{T}}\right] = \mathbf{I}_{\hat{m}}$ for all rows $i \in \{1, \ldots, n\}$.

Third, to show that each row has a bounded norm, we use the fact that the eigenfunctions $\mathcal{Y}_k$ in Equation (13) over $\{-1, 1\}^d$ satisfy $\mathcal{Y}_k(x_i)^2 = 1$ for all $k$. Thus, the norm of each row is

$$\|[\mathbf{\Psi}_{\leq \hat{m}}]_{i,:}\|_2 = \sqrt{\sum_{k=1}^{\hat{m}} \mathcal{Y}_k(x_i)^2} = \sqrt{\sum_{k=1}^{\hat{m}} 1} = \sqrt{\hat{m}}.$$

We can now apply Theorem 5.44 from Vershynin (2012). For any $t \geq 0$, this yields the following inequality with probability $1 - \hat{m}\exp\{-ct^2\}$, where $c$ is an absolute constant:

$$\left\|\frac{\mathbf{\Psi}_{\leq \hat{m}}^{\mathsf{T}}\mathbf{\Psi}_{\leq \hat{m}}}{n} - \mathbf{I}_{\hat{m}}\right\| \leq \max\left\{\|\mathbf{I}_{\hat{m}}\|^{\frac{1}{2}}\Delta, \Delta^2\right\}, \quad \text{where } \Delta = t\sqrt{\frac{\hat{m}}{n}}.$$

The choice $t = \frac{1}{2}\sqrt{\frac{n}{\hat{m}}}$ yields $\max\left\{\|\mathbf{I}_{\hat{m}}\|^{\frac{1}{2}}\Delta, \Delta^2\right\} = 1/2$, and the following error probability for large enough $d$:

$$\hat{m}\exp\left\{-c\frac{n}{4\hat{m}}\right\} \overset{(i)}{\lesssim} nq^{-\bar{\delta}}\exp\left\{-c'\frac{n}{nq^{-\bar{\delta}}}\right\} \lesssim q^{-\bar{\delta}} \cdot d^\ell \exp\{-c'q^{\bar{\delta}}\} \lesssim q^{-\bar{\delta}},$$

where $(i)$ follows from $\hat{m} \in \mathcal{O}(nq^{-\bar{\delta}})$.

**Bound for any $m < \hat{m}$**  Note that $\mathbf{\Psi}_{\leq m}^{\mathsf{T}}\mathbf{\Psi}_{\leq m}$ is a submatrix of $\mathbf{\Psi}_{\leq \hat{m}}^{\mathsf{T}}\mathbf{\Psi}_{\leq \hat{m}}$. Thus,

$$\frac{\mathbf{\Psi}_{\leq m}^{\mathsf{T}}\mathbf{\Psi}_{\leq m}}{n} - \mathbf{I}_m \quad \text{is also a submatrix of} \quad \frac{\mathbf{\Psi}_{\leq \hat{m}}^{\mathsf{T}}\mathbf{\Psi}_{\leq \hat{m}}}{n} - \mathbf{I}_{\hat{m}}.$$

Therefore,

$$\left\|\frac{\mathbf{\Psi}_{\leq m}^{\mathsf{T}}\mathbf{\Psi}_{\leq m}}{n} - \mathbf{I}_m\right\| \leq \left\|\frac{\mathbf{\Psi}_{\leq \hat{m}}^{\mathsf{T}}\mathbf{\Psi}_{\leq \hat{m}}}{n} - \mathbf{I}_{\hat{m}}\right\| \leq 1/2$$

with probability at least $1 - cd^{-\beta\bar{\delta}}$ uniformly over all $m \leq \hat{m}$.

## C.2  Further decomposition of the terms after $m$

In this section, we focus on the concentration of the smallest and largest eigenvalue of the kernel matrix $\mathbf{K}_{>m}$ to prove Lemma 3. However, this proof is involved, and requires additional tools. In particular, we further decompose $\mathcal{K}_{>m}$ into two kernels $\mathcal{K}_1$ and $\mathcal{K}_2$.

In the following, we consider the setting of Theorem 1 with a convolutional kernel $\mathcal{K}$ that satisfies Assumption 1. We define the additional notation

$$L := \left\lfloor \frac{\ell - \ell_\lambda - 1}{\beta} \right\rfloor \quad \text{and} \quad \bar{L} := \left\lfloor \frac{\ell - 1}{\beta} \right\rfloor.$$

Intuitively, $L$ is the maximum polynomial degree that $\mathcal{K}$ can learn with regularization, and $\bar{L}$ is the analogue without regularization. Finally, note that $\delta$ and $\bar{\delta}$ as defined in Theorem 1 can be written as $\delta = \frac{\ell - \ell_\lambda - 1}{\beta} - L$ and $\bar{\delta} = \frac{\ell - 1}{\beta} - \bar{L}$, respectively.

We now introduce the two additional kernels, and then show in Lemma 11 that $\mathcal{K}_{>m} = \mathcal{K}_1 + \mathcal{K}_2$. First, applying Proposition 1 to $\mathcal{K}$ yields

$$\mathcal{K}(x, x') = \sum_k \lambda_{S_k} \mathcal{Y}_{S_k}(x) \mathcal{Y}_{S_k}(x'), \tag{29}$$

where $\{S_k\}_{k>0}$ is a sequence of all subsets $S_k \subseteq \{1, \ldots, d\}$ with $\gamma(S_k) \leq q$, ordered such that $\lambda_{S_k} \geq \lambda_{S_{k+1}}$. Next, let $m \in \mathbb{N}$ be such that $n\lambda_{S_m} \in \Theta(\max\{\lambda, 1\})$, and define the index sets

$$\mathcal{I}_1 := \{k \in \mathbb{N} \mid k > m \text{ and } |S_k| \leq \bar{L} + 1\},$$
$$\mathcal{I}_2 := \{k \in \mathbb{N} \mid |S_k| \geq \bar{L} + 2\}.$$

Those sets induce the following kernels:

$$\mathcal{K}_1(x, x') := \sum_{k \in \mathcal{I}_1} \lambda_{S_k} \mathcal{Y}_{S_k}(x) \mathcal{Y}_{S_k}(x'), \qquad \mathcal{S}_1(x, x') := \sum_{k \in \mathcal{I}_1} \lambda_{S_k}^2 \mathcal{Y}_{S_k}(x) \mathcal{Y}_{S_k}(x'),$$

$$\mathcal{K}_2(x, x') := \sum_{k \in \mathcal{I}_2} \lambda_{S_k} \mathcal{Y}_{S_k}(x) \mathcal{Y}_{S_k}(x'), \qquad \mathcal{S}_2(x, x') := \sum_{k \in \mathcal{I}_2} \lambda_{S_k}^2 \mathcal{Y}_{S_k}(x) \mathcal{Y}_{S_k}(x'),$$

where $\mathcal{S}_1$ and $\mathcal{S}_2$ are the squared kernels corresponding to $\mathcal{K}_1$ and $\mathcal{K}_2$, respectively. The empirical kernel matrices $\mathbf{K}_1, \mathbf{K}_2, \mathbf{S}_1, \mathbf{S}_2 \in \mathbb{R}^{n \times n}$ are

$$[\mathbf{K}_1]_{i,j} = \mathcal{K}_1(x_i, x_j), \quad [\mathbf{K}_2]_{i,j} = \mathcal{K}_2(x_i, x_j), \quad [\mathbf{S}_1]_{i,j} = \mathcal{S}_1(x_i, x_j), \quad \text{and} \quad [\mathbf{S}_2]_{i,j} = \mathcal{S}_2(x_i, x_j).$$

Furthermore, as in the original kernel decomposition, we define the matrices

$$\boldsymbol{\Psi}_1 \in \mathbb{R}^{n \times |\mathcal{I}_1|}, \qquad\qquad [\boldsymbol{\Psi}_1]_{i,j} = \mathcal{Y}_{S_{k_j}}(x_i),$$
$$\mathbf{D}_1 \in \mathbb{R}^{|\mathcal{I}_1| \times |\mathcal{I}_1|}, \qquad\qquad \mathbf{D}_1 = \text{diag}(\lambda_{S_{k_1}}, \ldots, \lambda_{S_{k_{|\mathcal{I}_1|}}}),$$

where $\{k_j\}_{j=1}^{|\mathcal{I}_1|}$ is a sequence of all indices in $\mathcal{I}_1$ ordered such that $\lambda_{S_{k_j}} \geq \lambda_{S_{k_{j+1}}}$. Intuitively, $\boldsymbol{\Psi}_1, \mathbf{D}_1$ are the analogue to $\boldsymbol{\Psi}_{\leq m}, \mathbf{D}_{\leq m}$ in the original decomposition $\mathcal{K} = \mathcal{K}_{\leq m} + \mathcal{K}_{>m}$.

Lastly, we define $\bar{m}$ as the largest eigenvalue corresponding to an eigenfunction $\mathcal{Y}_S$ of degree $|S| \geq \bar{L} + 2$, that is,

$$\bar{m} := \min \mathcal{I}_2.$$

Using the previous definitions, the following lemma establishes that $\mathcal{K}_1$ and $\mathcal{K}_2$ indeed constitute a decomposition of $\mathcal{K}_{>m}$.

**Lemma 11** (1-2 decomposition). *For $d$ sufficiently large, we have*

$$\mathcal{K}_{>m}(x, x') = \mathcal{K}_1(x, x') + \mathcal{K}_2(x, x') \quad \text{and} \quad \mathcal{S}_{>m}(x, x') = \mathcal{S}_1(x, x') + \mathcal{S}_2(x, x').$$

*Proof.* For the decomposition of $\mathcal{K}_{>m}$, we have to show that exactly the eigenfunctions with index larger than $m$ appear in either $\mathcal{K}_1$ or $\mathcal{K}_2$, that is, $\mathcal{I}_1 \cup \mathcal{I}_2 = \{k > m\}$, and that no eigenfunction appears in both $\mathcal{K}_1$ or $\mathcal{K}_2$, that is, $\mathcal{I}_1 \cap \mathcal{I}_2 = \emptyset$. Furthermore, since we can write $\mathcal{S}_{>m}(x, x') = \sum_{k>m} \lambda_{S_k}^2 \mathcal{Y}_{S_k}(x) \mathcal{Y}_{S_k}(x')$ by Lemma 5, the same argument implies the 1-2 decomposition of $\mathcal{S}_{>m}$.

First, from the definition of $\mathcal{I}_1$ and $\mathcal{I}_2$, it follows directly that $\mathcal{I}_1 \cap \mathcal{I}_2 = \emptyset$, that $\mathcal{I}_1 \cup \mathcal{I}_2 \supseteq \{k > m\}$, and that $\mathcal{I}_1 \subseteq \{k > m\}$. Hence, to conclude the proof, we only need to show that $\mathcal{I}_2 \subseteq \{k > m\}$. Since the eigenvalues are sorted in decreasing order, we equivalently show that, for $d$ sufficiently large, all eigenvalues $\lambda_{S_k}$ with $k \in \mathcal{I}_2$ are smaller than $\lambda_{S_m} \in \Theta(\max\{\lambda, 1\}/n)$.

More precisely, we show that $\max_{k \in \mathcal{I}_2} n\lambda_{S_k} \in o(n\lambda_{S_m}) = o(\max\{\lambda, 1\})$. Using $T$ from Assumption 1, we have

$$\max_{k \in \mathcal{I}_2} n\lambda_{S_k} = \max \left\{ \underbrace{\max_{k \in \mathcal{I}_2 : |S_k| < T} n\lambda_{S_k}}_{=:M_1}, \underbrace{\max_{k \in \mathcal{I}_2 : |S_k| \geq T} n\lambda_{S_k}}_{=:M_2} \right\}.$$

For $M_1$, we bound a generic $k \in \mathcal{I}_2$ with $|S_k| < T$ as follows:

$$n\lambda_{S_k} \overset{(i)}{\in} \mathcal{O}\left(\frac{n}{dq^{|S_k|-1}}\right) \subseteq \mathcal{O}\left(\frac{n}{dq^{(\bar{L}+2)-1}}\right) = \mathcal{O}\left(\frac{d^{1+\ell_\lambda+\beta(L+\delta)}}{d^{1+\beta(\bar{L}+\delta)}}d^{-\beta(1-\delta)}\right)$$

$$= \mathcal{O}(\max\{\lambda, 1\}d^{\beta(L-\bar{L})}d^{-\beta(1-\delta)}) \overset{(ii)}{\subseteq} o(n\lambda_m),$$

where $(i)$ applies Equation (22) from Lemma 10, and $(ii)$ uses $\bar{L} \geq L$ and $\delta < 1$. In particular, this implies $M_1 \in o(n\lambda_m)$.

For $M_2$ we have

$$\max_{k \in \mathcal{I}_2 : |S_k| \geq T} n\lambda_{S_k} = n \max_{\substack{|S_k| \geq T \\ \gamma(S_k) \leq q}} \lambda_{S_k} \overset{(i)}{\in} \mathcal{O}\left(\frac{n}{dq^{T-1}}\right)$$

$$\overset{(ii)}{\subseteq} \mathcal{O}\left(\frac{n}{dq^{(\bar{L}+2)-1}}\right) = \mathcal{O}\left(\frac{d^{1+\ell_\lambda+\beta(L+\delta)}}{d^{1+\beta(\bar{L}+\delta)}}d^{-\beta(1-\delta)}\right)$$

$$= \mathcal{O}(\max\{\lambda, 1\}d^{\beta(L-\bar{L})}d^{-\beta(1-\delta)}) \overset{(iii)}{\subseteq} o(n\lambda_m),$$

where $(i)$ applies Equation (23) from Lemma 10, $(ii)$ uses that $\bar{L} + 2 \leq T$, and $(iii)$ follows from $\bar{L} \geq L$ and $\delta < 1$.

Combining the bounds on $M_1$ and $M_2$, we have $\max_{k \in \mathcal{I}_2} n\lambda_{S_k} \in o(n\lambda_{S_m}) = o(\max\{\lambda, 1\})$. Hence, for $d$ sufficiently large, all $k \in \mathcal{I}_2$ yield $\lambda_{S_k} < \lambda_{S_m}$ and consequently $k > m$. □

Using the 1-2 decomposition, we now prove Lemma 3. We defer the auxiliary Lemmas 12 to 15 to Appendix C.4, and concentration-results to Appendix C.5.

## C.3 PROOF OF LEMMA 3

Throughout the proof, we assume $d$ to be large enough such that all quantities are well-defined and all necessary lemmas apply. In particular, we assume the conditions of Lemma 11 to be satisfied, and that $c < \lfloor q/2 \rfloor < q < d/2$ for $c$ in Assumption 1. Hence, $L + 2 \leq \bar{L} + 2 < T < \lfloor q/2 \rfloor$, and we can apply Lemmas 9 to 15, the setting of Appendix C.2, as well as Assumption 1 throughout the proof. We will mention additional implicit lower bounds on $d$ as they arise.

The proof proceeds in three steps: we first bound $r_1$ and $r_2$, then bound $\mathbf{Tr}(\mathbf{S}_{>m})$, and finally $\sum_{i=1+m}^{n} \mu_i(\mathbf{S}_{>m})$. We do not establish the required matrix concentration results directly, but apply various auxiliary lemmas. All corresponding statements hold with either probability at least $1 - \tilde{c}q^{-\bar{\delta}}$ or at least $1 - \tilde{c}q^{-(1-\bar{\delta})}$ for context-dependent constants $\tilde{c}$. We hence implicitly choose a $c > 0$ such that collecting all error probabilities yields the statement of Lemma 3 with probability at least $1 - cd^{-\beta \min\{\bar{\delta}, 1-\bar{\delta}\}}$.

To start, let $m \in \mathbb{N}$ as in the statement of Lemma 3, and instantiate Appendix C.2 with that $m$. In particular, Lemma 11 yields the 1-2 decomposition $\mathbf{K}_{>m} = \mathbf{K}_1 + \mathbf{K}_2$ and $\mathbf{S}_{>m} = \mathbf{S}_1 + \mathbf{S}_2$, which we will henceforth use. Finally, we define

$$\mathcal{Q}_l^{(d,q)}(x, x') := \sum_{\substack{\gamma(S) \leq q \\ |S|=l}} \frac{q+1-\gamma(S)}{d\mathcal{B}(l,q)} \mathcal{Y}_S(x)\mathcal{Y}_S(x') \tag{30}$$

with the corresponding kernel matrix $\mathbf{Q}_l^{(d,q)} \in \mathbb{R}^{n \times n}$.

**Bound on $r_1$ and $r_2$**   Remember the definition of $r_1$ and $r_2$:

$$r_1 = \frac{\mu_{\min}(\mathbf{K}_{>m}) + \lambda}{\max\{\lambda, 1\}}, \qquad r_2 = \frac{\|\mathbf{K}_{>m}\| + \lambda}{\max\{\lambda, 1\}}.$$

To bound those quantities, we have to bound $\mu_{\min}(\mathbf{K}_{>m})$ and $\|\mathbf{K}_{>m}\|$. For the upper bound on $\|\mathbf{K}_{>m}\|$, we use the triangle inequality on $\|\mathbf{K}_{>m}\| = \|\mathbf{K}_1 + \mathbf{K}_2\|$, and then bound $\|\mathbf{K}_1\|$ and $\|\mathbf{K}_2\|$ individually.

Note that we can write $\mathbf{K}_1 = \boldsymbol{\Psi}_1 \mathbf{D}_1 \boldsymbol{\Psi}_1^\intercal$ by definition. Hence,

$$\|\mathbf{K}_1\| = \|\boldsymbol{\Psi}_1 \mathbf{D}_1 \boldsymbol{\Psi}_1^\intercal\| = n\|\mathbf{D}_1\|\left\|\frac{\boldsymbol{\Psi}_1^\intercal \boldsymbol{\Psi}_1}{n}\right\| \overset{(i)}{\leq} 1.5n\|\mathbf{D}_1\| \overset{(ii)}{\leq} 1.5n\lambda_m \overset{(iii)}{\in} \mathcal{O}\left(\max\{\lambda, 1\}\right),$$

where $(i)$ follows from Lemma 12 with probability at least $1 - \tilde{c}_1 q^{-\bar{\delta}}$, $(ii)$ uses that all eigenvalues of $\mathcal{K}_1$ are at most $\lambda_m$ by definition, and $(iii)$ follows from $n\lambda_m \in \Theta(\max\{\lambda, 1\})$.

Next, Lemma 13 directly yields with probability at least $1 - \tilde{c}_2 q^{-(1-\bar{\delta})}$ that $\|\mathbf{K}_2\| \in \Theta(1)$ and $\mu_{\min}(\mathbf{K}_2) \in \Theta(1)$. Hence, with probability at least $1 - (\tilde{c}_1 q^{-\bar{\delta}} + \tilde{c}_2 q^{-(1-\bar{\delta})}) \geq 1 - cq^{-\min\{\bar{\delta}, 1-\bar{\delta}\}}$, we have

$$\|\mathbf{K}_1\|, \|\mathbf{K}_2\| \in \mathcal{O}\left(\max\{\lambda, 1\}\right),$$
$$\mu_{\min}(\mathbf{K}_2) \in \Omega(1).$$

This implies

$$\|\mathbf{K}_{>m}\| + \lambda \leq \|\mathbf{K}_1\| + \|\mathbf{K}_2\| + \lambda \in \mathcal{O}(\max\{\lambda, 1\}),$$
$$\mu_{\min}(\mathbf{K}_{>m}) + \lambda \geq \mu_{\min}(\mathbf{K}_2) + \lambda \in \Omega(\max\{\lambda, 1\}),$$

and subsequently

$$r_2 = \frac{\|\mathbf{K}_{>m}\| + \lambda}{\max\{\lambda, 1\}} \in \mathcal{O}(1),$$
$$r_1 = \frac{\mu_{\min}(\mathbf{K}_{>m}) + \lambda}{\max\{\lambda, 1\}} \in \Omega(1).$$

Finally, since $r_1 \leq r_2$, this yields $r_1, r_2 \in \Theta(1)$.

**Bound on $\mathbf{Tr}(\mathbf{S}_{>m})$**   We need to show that

$$\mathbf{Tr}(\mathbf{S}_{>m}) \lesssim d^{\ell_\lambda} q^{-\delta} + d^{\ell_\lambda} q^{\delta-1} \tag{31}$$

with high probability, where the two terms correspond to the 1-2 decomposition $\mathbf{Tr}(\mathbf{S}_{>m}) = \mathbf{Tr}(\mathbf{S}_1) + \mathbf{Tr}(\mathbf{S}_2)$. We differentiate between $\bar{L} = L$ and $\bar{L} > L$. Intuitively, the case $\bar{L} = L$ corresponds to interpolation or weak regularization, because the maximum degree of learnable polynomials with regularization equals the one without regularization. Conversely, $\bar{L} > L$ corresponds to strong regularization.

*Case $\bar{L} = L$ (interpolation or weak regularization):* In this setting,

$$\delta = \frac{\ell - \ell_\lambda - 1}{\beta} - L = \frac{\ell - 1}{\beta} - \bar{L} - \frac{\ell_\lambda}{\beta} = \bar{\delta} - \frac{\ell_\lambda}{\beta}. \tag{32}$$

First, Lemma 14 yields $\mathbf{Tr}(\mathbf{S}_2) \in \Theta(q^{-(1-\bar{\delta})})$ with probability at least $1 - \tilde{c}_3 q^{-(1-\bar{\delta})}$. Therefore,

$$\mathbf{Tr}(\mathbf{S}_2) \in \Theta(q^{-(1-\bar{\delta})}) \overset{(i)}{=} \Theta(q^{-(1-\delta)+\frac{\ell_\lambda}{\beta}}) = \Theta\left(d^{\ell_\lambda} q^{-(1-\delta)}\right),$$

where $(i)$ follows from Equation (32).

We now consider $\mathbf{Tr}(\mathbf{S}_1)$:

$$\mathbf{Tr}(\mathbf{S}_1) = n\mathbf{Tr}\left(\frac{\boldsymbol{\Psi}_1^\intercal \boldsymbol{\Psi}_1}{n}\mathbf{D}_1^2\right) \leq n\left\|\frac{\boldsymbol{\Psi}_1^\intercal \boldsymbol{\Psi}_1}{n}\right\|\mathbf{Tr}(\mathbf{D}_1^2) \overset{(i)}{\leq} 1.5n\sum_{k \in \mathcal{I}_1} \lambda_{S_k}^2,$$

where $(i)$ follows from Lemma 12 with probability at least $1 - \tilde{c}_1 q^{-\bar{\delta}}$. The bound continues as

$$
\begin{aligned}
\mathbf{Tr}(\mathbf{S}_1) &\lesssim n \sum_{k \in \mathcal{I}_1} \lambda_{S_k}^2 \overset{(i)}{\leq} n \lambda_m^2 |\mathcal{I}_1| \\
&\overset{(ii)}{\in} \mathcal{O}\left( \max\{\lambda, 1\}^2 \frac{|\mathcal{I}_1|}{n} \right) \\
&\overset{(iii)}{\subseteq} \mathcal{O}\left( d^{2\ell_\lambda} \frac{dq^{\bar{L}}}{dq^{\bar{L}+\bar{\delta}}} \right) = \mathcal{O}\left( d^{2\ell_\lambda} q^{-\bar{\delta}} \right) \overset{(iv)}{=} \mathcal{O}\left( d^{\ell_\lambda} q^{-\delta} \right),
\end{aligned}
$$

where $(i)$ uses $\lambda_k \leq \lambda_m$ for all $k \in \mathcal{I}_1$ by definition, $(ii)$ uses $n\lambda_m \in \Theta(\max\{\lambda, 1\})$, and $(iv)$ follows from Equation (32). Furthermore, $(iii)$ uses the following bound of $|\mathcal{I}_1|$:

$$
|\mathcal{I}_1| \leq \sum_{l=0}^{\bar{L}+1} \mathcal{C}(l, q, d) \overset{(i)}{=} 1 + d \sum_{l=1}^{\bar{L}+1} \binom{q-1}{l-1} \overset{(ii)}{\leq} 1 + d \sum_{l=1}^{\bar{L}+1} \left( e \frac{q-1}{l-1} \right)^{l-1} \overset{(iii)}{\in} \mathcal{O}(d \cdot q^{\bar{L}}),
$$

where $\mathcal{C}(l, q, d)$ is defined in Equation (17), $(i)$ follows from Lemma 9, $(ii)$ is a classical bound on the binomial coefficient, and $(iii)$ follows from the fact that the term corresponding to $\bar{L} + 1$ dominates the polynomial.

Finally, collecting the upper bounds on $\mathbf{Tr}(\mathbf{S}_1)$ and $\mathbf{Tr}(\mathbf{S}_2)$ yields

$$
\mathbf{Tr}(\mathbf{S}_{\geq m}) = \mathbf{Tr}(\mathbf{S}_1) + \mathbf{Tr}(\mathbf{S}_2) \in \mathcal{O}\left( d^{\ell_\lambda} q^{-\delta} + d^{\ell_\lambda} q^{-(1-\delta)} \right)
$$

with probability at least $1 - (\tilde{c}_3 q^{-(1-\bar{\delta})} + \tilde{c}_1 q^{-\bar{\delta}}) \geq 1 - cq^{-\min\{\bar{\delta}, 1-\bar{\delta}\}}$.

*Case $\bar{L} > L$ (strong regularization):* In this setting, the dominating rate will arise from $\mathbf{Tr}(\mathbf{S}_1)$. We start by linking $\bar{\delta}$ and $\delta$ in analogy to Equation (32):

$$
\delta = \frac{\ell - \ell_\lambda - 1}{\beta} - L = -\frac{\ell_\lambda}{\beta} + \frac{\ell - 1}{\beta} - \bar{L} + \bar{L} - L = \bar{\delta} - \frac{\ell_\lambda}{\beta} + \bar{L} - L. \tag{33}
$$

Next, as in the previous case, Lemma 14 yields $\mathbf{Tr}(\mathbf{S}_2) \in \Theta(q^{-(1-\bar{\delta})})$ with probability at least $1 - \tilde{c}_3 q^{-(1-\bar{\delta})}$, and therefore

$$
\mathbf{Tr}(\mathbf{S}_2) \in \Theta(q^{-(1-\bar{\delta})}) = \Theta(q^{\bar{\delta}-1}) \overset{(i)}{=} \Theta\left( d^{\ell_\lambda} q^{\delta - (\bar{L}-L)-1} \right) \overset{(ii)}{\subseteq} o\left( d^{\ell_\lambda} q^{-(1-\delta)} \right),
$$

where $(i)$ follows from Equation (33), and $(ii)$ from $\bar{L} > L$.

To bound $\mathbf{Tr}(\mathbf{S}_1)$, we start as in the previous case:

$$
\mathbf{Tr}(\mathbf{S}_1) = n\mathbf{Tr}\left( \frac{\mathbf{\Psi}_1^\intercal \mathbf{\Psi}_1}{n} \mathbf{D}_1^2 \right) \leq n \left\| \frac{\mathbf{\Psi}_1^\intercal \mathbf{\Psi}_1}{n} \right\| \mathbf{Tr}(\mathbf{D}_1^2) \overset{(i)}{\leq} 1.5 n \sum_{k \in \mathcal{I}_1} \lambda_{S_k}^2,
$$

where $(i)$ follows from Lemma 12 with probability at least $1 - \tilde{c}_1 q^{-\bar{\delta}}$. We then decompose the sum over all squared eigenvalues with index in $\mathcal{I}_1$ as

$$
n \sum_{k \in \mathcal{I}_1} \lambda_{S_k}^2 = n \underbrace{\sum_{\substack{k \in \mathcal{I}_1 \\ |S_k| \leq L+1}} \lambda_{S_k}^2}_{=:E_1} + n \underbrace{\sum_{\substack{k \in \mathcal{I}_1 \\ |S_k| = L+2}} \lambda_{S_k}^2}_{=:E_2} + n \underbrace{\sum_{\substack{k \in \mathcal{I}_1 \\ |S_k| \geq L+3}} \lambda_{S_k}^2}_{=:E_3},
$$

and bound the three terms individually.

First, we upper-bound $E_1$ as follows:

$$
\begin{aligned}
E_1 &= n \sum_{\substack{k \in \mathcal{I}_1 \\ |S_k| \leq L+1}} \lambda_{S_k}^2 \overset{(i)}{\leq} \frac{n^2 \lambda_m^2}{n} \sum_{\substack{k \in \mathcal{I}_1 \\ |S_k| \leq L+1}} 1 \\
&\leq \frac{n^2 \lambda_m^2}{n} \sum_{l=0}^{L+1} \mathcal{C}(l, q, d) \overset{(ii)}{\in} \mathcal{O}\left( \frac{\max\{\lambda, 1\}^2}{n} dq^L \right) \\
&= \mathcal{O}\left( \frac{d^{2\ell_\lambda}}{d \cdot d^{\ell_\lambda} \cdot q^{L+\delta}} dq^L \right) = \mathcal{O}\left( d^{\ell_\lambda} q^{-\delta} \right),
\end{aligned}
$$

where $(i)$ follows from $\lambda_k \leq \lambda_m$ for all $k > m$ due to the decreasing order of eigenvalues. Step $(ii)$ applies $n\lambda_m \in \Theta(\max\{\lambda, 1\})$, as well as $\sum_{l=0}^{L+1} \mathcal{C}(l, q, d) \in \mathcal{O}(d \cdot q^L)$, which follows as in the other case from Lemma 9 and the classical bound on the binomial coefficient.

Second, the upper bound of $E_2$ arises as follows:

$$
E_2 = n \sum_{\substack{k \in \mathcal{I}_1 \\ |S_k| = L+2}} \lambda_{S_k}^2 \stackrel{(i)}{=} n(\xi_{L+2}^{(q)})^2 \sum_{\substack{k \in \mathcal{I}_1 \\ |S_k| = L+2}} \left( \frac{q + 1 - \gamma(S_k)}{d\mathcal{B}(L+2, q)} \right)^2
$$

$$
\stackrel{(ii)}{\leq} (\xi_{L+2}^{(q)})^2 \frac{n \cdot q}{d\mathcal{B}(L+2, q)} \sum_{\substack{\gamma(S) \leq q \\ |S| = L+2}} \frac{q + 1 - \gamma(S)}{d\mathcal{B}(L+2, q)}
$$

$$
\stackrel{(iii)}{\lesssim} \frac{n \cdot q}{d \cdot q^{L+2}} \sum_{\substack{\gamma(S) \leq q \\ |S| = L+2}} \frac{q + 1 - \gamma(S)}{d\mathcal{B}(L+2, q)}
$$

$$
\stackrel{(iv)}{=} \frac{n \cdot q}{d \cdot q^{L+2}} \mathcal{Q}_{L+2}^{(d,q)}(x, x)
$$

$$
\stackrel{(v)}{\in} \mathcal{O}\left( \frac{d \cdot d^{\ell_\lambda} q^{L+\delta} \cdot q}{d \cdot q^{L+2}} \right) = \mathcal{O}\left( d^{\ell_\lambda} q^{-(1-\delta)} \right),
$$

where $(i)$ follows from Proposition 1, and $(ii)$ uses $q + 1 - \gamma(S_k) \leq q$. Next, $(iii)$ uses that Equations (18) and (21) in Assumption 1 imply $\xi_{L+2}^{(q)} \in \Theta(1)$, and applies the bound $\mathcal{B}(L+2, q) = \binom{q}{L+2} \leq (eq/(L+2))^{L+2}$. Step $(iv)$ uses $\mathcal{Y}_S(x)\mathcal{Y}_S(x) = 1$ for all $S$ and $x \in \{-1, 1\}^d$, together with the definition of $\mathcal{Q}_{L+2}^{(d,q)}$. Lastly, $(v)$ applies Lemma 15 and $n \in \Theta(d^\ell) = \Theta(d \cdot d^{\ell_\lambda} q^{L+\delta})$.

Third, we upper-bound $E_3$:

$$
E_3 = n \sum_{\substack{k \in \mathcal{I}_1 \\ |S_k| \geq L+3}} \lambda_{S_k}^2 \leq n \left( \max_{k \in \mathcal{I}_1, |S_k| \geq L+3} \lambda_S \right) \sum_{\substack{k \in \mathcal{I}_1 \\ |S_k| \geq L+3}} \lambda_{S_k} \lesssim n \max_{k \in \mathcal{I}_1, |S_k| \geq L+3} (\lambda_{S_k}).
$$

The last step follows from

$$
\sum_{\substack{k \in \mathcal{I}_1 \\ |S_k| \geq L+3}} \lambda_{S_k} \leq \sum_{l=L+3}^{\bar{L}+1} \sum_{\substack{\gamma(S) \leq q \\ |S| = l}} \lambda_S
$$

$$
\stackrel{(i)}{=} \sum_{l=L+3}^{\bar{L}+1} \sum_{\substack{\gamma(S) \leq q \\ |S| = l}} \xi_l^{(q)} \frac{q + 1 - \gamma(S)}{d\mathcal{B}(l, q)}
$$

$$
\stackrel{(ii)}{=} \sum_{l=L+3}^{\bar{L}+1} \xi_l^{(q)} \sum_{\substack{\gamma(S) \leq q \\ |S| = l}} \frac{q + 1 - \gamma(S)}{d\mathcal{B}(l, q)} \mathcal{Y}_S(x)\mathcal{Y}_S(x)
$$

$$
\stackrel{(iii)}{=} \sum_{l=L+3}^{\bar{L}+1} \xi_l^{(q)} \stackrel{(iv)}{\lesssim} 1,
$$

where $(i)$ follows from Proposition 1, $(ii)$ uses $\mathcal{Y}_S(x)\mathcal{Y}_S(x) = 1$ for all $S$ and $x \in \{-1, 1\}^d$, $(iii)$ applies the definition of $\mathcal{Q}_l^{(d,q)}$ and Lemma 15, and $(iv)$ follows from Equations (18) and (21) in Assumption 1 since $\bar{L} + 1 \leq T$.

For $\max_{k \in \mathcal{I}_1, |S_k| \geq L+3}(\lambda_{S_k})$, we bound each element individually:

$$
\lambda_{S_k} \stackrel{(i)}{\in} \mathcal{O}\left( \frac{1}{dq^{|S_k|-1}} \right) \subseteq \mathcal{O}\left( \frac{1}{dq^{(L+3)-1}} \right),
$$

where $(i)$ uses Equation (22) in Lemma 10 since $k \leq \bar{L} + 1 < T$ by definition of $\mathcal{I}_1$. Hence, we obtain the following bound on $E_3$:

$$E_3 \lesssim n \max_{k \in \mathcal{I}_1, |S_k| \geq L+3} (\lambda_{S_k}) \in \mathcal{O}\left(\frac{n}{dq^{L+2}}\right) = \mathcal{O}\left(\frac{d \cdot d^{\ell_\lambda} q^{L+\delta}}{dq^{L+2}}\right) \subseteq \mathcal{O}\left(d^{\ell_\lambda} q^{-(1-\delta)}\right).$$

Finally, we can bound $\mathbf{Tr}(\mathbf{S}_1)$ as

$$\mathbf{Tr}(\mathbf{S}_1) \leq 1.5n \sum_{k \in \mathcal{I}_1} \lambda_{S_k}^2 = E_1 + E_2 + E_3 \in \mathcal{O}\left(d^{\ell_\lambda} q^{-\delta} + d^{\ell_\lambda} q^{-(1-\delta)}\right),$$

which yields

$$\mathbf{Tr}(\mathbf{S}_{\geq m}) = \mathbf{Tr}(\mathbf{S}_1) + \mathbf{Tr}(\mathbf{S}_2) \in \mathcal{O}\left(d^{\ell_\lambda} q^{-\delta} + d^{\ell_\lambda} q^{-(1-\delta)}\right)$$

as desired with probability at least $1 - (\tilde{c}_3 q^{-(1-\bar{\delta})} + \tilde{c}_1 q^{-\bar{\delta}}) \geq 1 - cq^{-\min\{\bar{\delta}, 1-\bar{\delta}\}}$.

**Bound on $\sum_{i=1+m}^{n} \mu_i(\mathbf{S}_{>m})$** As before, we differentiate between no/weak and strong regularization, that is, between $\bar{L} = L$ and $\bar{L} > L$:

*Case $\bar{L} = L$ (interpolation or weak regularization):* In this case, we start by directly bounding

$$\sum_{i=1+m}^{n} \mu_i(\mathbf{S}_{>m}) = \sum_{i=1+m}^{n} \mu_i(\mathbf{S}_1 + \mathbf{S}_2) \geq \sum_{i=1+m}^{n} \mu_i(\mathbf{S}_2)$$

$$= \sum_{i=1}^{n} \mu_i(\mathbf{S}_2) - \sum_{i=1}^{m} \mu_i(\mathbf{S}_2) \overset{(i)}{\geq} \mathbf{Tr}(\mathbf{S}_2) - m\|\mathbf{S}_2\| \qquad (34)$$

$$\overset{(ii)}{\in} \Omega\left(q^{-(1-\bar{\delta})} - \frac{m}{dq^{\bar{L}+1}}\right)$$

$$\overset{(iii)}{=} \Omega\left(d^{\ell_\lambda} q^{-(1-\delta)} - \frac{m}{dq^{\bar{L}+1}}\right),$$

where $(i)$ bounds each of the first $m$ eigenvalues of $S_2$ with the largest one, $(ii)$ follows from Lemma 14 with probability at least $1 - \tilde{c}_3 q^{-(1-\bar{\delta})}$, and $(iii)$ from Equation (32) since $\bar{L} = L$.

To conclude the lower bound, it suffices to show that $\frac{m}{dq^{\bar{L}+1}} \in o(d^{\ell_\lambda} q^{-(1-\delta)})$:

$$\frac{m}{dq^{\bar{L}+1}} \overset{(i)}{\in} \mathcal{O}\left(\frac{nq^{-\delta}}{\max\{\lambda, 1\} dq^{\bar{L}+1}}\right) = \mathcal{O}\left(\frac{q^{-\delta} dq^{\bar{\delta}+\bar{L}}}{d^{\ell_\lambda} dq^{\bar{L}+1}}\right) \overset{(ii)}{=} \mathcal{O}\left(q^{-\bar{\delta}} d^{\ell_\lambda} q^{-(1-\delta)}\right) \overset{(iii)}{\subseteq} o(d^{\ell_\lambda} q^{-(1-\delta)}),$$

where $(i)$ follows from $m \in \mathcal{O}\left(\frac{nq^{-\delta}}{\max\{\lambda, 1\}}\right)$, and $(ii)$ from Equation (32). For $(iii)$, note that $\bar{\delta} = 0$ for a sufficiently large $c$ yields a vacuous result. We hence assume without loss of generality that $\bar{\delta} > 0$, which justifies the step. This concludes the proof for the current case with probability at least $1 - \tilde{c}_3 q^{-(1-\bar{\delta})} \geq 1 - cq^{-\min\{\bar{\delta}, 1-\bar{\delta}\}}$.

*Case $\bar{L} > L$ (strong regularization):* In this case, we define the additional index set

$$\mathcal{I}_3 := \{k \in \mathcal{I}_1 \mid |S_k| = L+2\}$$

with $\mathbf{S}_3, \mathbf{\Psi}_3, \mathbf{D}_3$ analogously to $\mathbf{S}_1, \mathbf{\Psi}_1, \mathbf{D}_1$ in Appendix C.2, but using $\mathcal{I}_3$ instead of $\mathcal{I}_1$. Since $\mathcal{I}_3 \subseteq \mathcal{I}_1$, it follows that $\mathbf{\Psi}_3^\intercal \mathbf{\Psi}_3$ is a submatrix of $\mathbf{\Psi}_1^\intercal \mathbf{\Psi}_1$, and thus

$$\frac{\mathbf{\Psi}_3^\intercal \mathbf{\Psi}_3}{n} - \mathbf{I}_{|\mathcal{I}_3|} \quad \text{is a submatrix of} \quad \frac{\mathbf{\Psi}_1^\intercal \mathbf{\Psi}_1}{n} - \mathbf{I}_{|\mathcal{I}_1|}.$$

This particularly implies

$$\left\|\frac{\mathbf{\Psi}_3^\intercal \mathbf{\Psi}_3}{n} - \mathbf{I}_{|\mathcal{I}_3|}\right\| \leq \left\|\frac{\mathbf{\Psi}_1^\intercal \mathbf{\Psi}_1}{n} - \mathbf{I}_{|\mathcal{I}_1|}\right\| \overset{(i)}{\leq} 1/2, \qquad (35)$$

where $(i)$ follows from Lemma 12 with probability at least $1 - \tilde{c}_1 q^{-\bar{\delta}}$.

We now move our focus back to the lower bound of $\sum_{i=1+m}^{n} \mu_i(\mathbf{S}_{>m})$:

$$\sum_{i=1+m}^{n} \mu_i(\mathbf{S}_{>m}) \overset{(i)}{\geq} \sum_{i=1+m}^{n} \mu_i(\mathbf{S}_1) \overset{(ii)}{\geq} \sum_{i=1+m}^{n} \mu_i(\mathbf{S}_3) \overset{(iii)}{\geq} \mathbf{Tr}(\mathbf{S}_3) - m\|\mathbf{S}_3\|, \qquad (36)$$

where $(i)$ follows from the 1-2 decomposition $\mathbf{S}_{>m} = \mathbf{S}_1 + \mathbf{S}_2$, $(ii)$ from the fact that $\mathcal{I}_3 \subseteq \mathcal{I}_1$, and $(iii)$ analogously to Equation (34). Similar to the previous case, we conclude the proof by first showing that $\mathbf{Tr}(\mathbf{S}_3) \in \Omega\left(d^{\ell_\lambda} q^{-(1-\delta)}\right)$, and then $m\|\mathbf{S}_3\| \in o(\mathbf{Tr}(\mathbf{S}_3))$.

For the lower bound of $\mathbf{Tr}(\mathbf{S}_3)$, we start with

$$\mathbf{Tr}(\mathbf{S}_3) = n\mathbf{Tr}\left(\frac{1}{n}\boldsymbol{\Psi}_3 \mathbf{D}_3^2 \boldsymbol{\Psi}_3^\mathsf{T}\right)$$

$$\geq n\mathbf{Tr}\left(\mathbf{D}_3^2\right) \mu_{\min}\left(\frac{\boldsymbol{\Psi}_3^\mathsf{T} \boldsymbol{\Psi}_3}{n}\right)$$

$$\overset{(i)}{\geq} 0.5n \sum_{k \in \mathcal{I}_3} \lambda_k^2$$

$$\overset{(ii)}{=} n(\xi_{L+2}^{(q)})^2 \sum_{\substack{|S|=L+2 \\ \gamma(S) \leq q}} \left(\frac{q+1-\gamma(S)}{d\mathcal{B}(L+2,q)}\right)^2$$

where $(i)$ follows with high probability from Equation (35). Step $(ii)$ applies Proposition 1, and the fact that $\mathcal{I}_3 = \{k \in \mathbb{N} \mid |S_k| = L+2 \text{ and } \gamma(S) \leq q\}$ for $d$ sufficiently large. To show this, we use

$$\lambda_m \in \Theta\left(\frac{d^{\ell_\lambda}}{n}\right) = \Theta\left(\frac{d^{\ell_\lambda}}{d d^{\ell_\lambda} q^{L+\delta}}\right) = \Theta\left(\frac{1}{dq^{L+\delta}}\right).$$

Since the eigenvalues are in decreasing order and $L+2 \leq \bar{L}+1$ in the current case, we only need to show that $\lambda_S < \lambda_m$ for all $S \subseteq \{1, \ldots, d\}$ with $|S| = \bar{L}+2$ and $\gamma(S) \leq q$:

$$\lambda_S \overset{(i)}{\in} \mathcal{O}\left(\frac{1}{dq^{L+1}}\right) = o\left(\frac{1}{dq^{L+\delta}}\right) = o(\lambda_m),$$

where $(i)$ applies Equation (22) in Lemma 10 since $L+2 < T$. Thus, $\lambda_S < \lambda_m$ for all $S \subseteq \{1, \ldots, d\}$ with $|S| = \bar{L}+2$ and $\gamma(S) \leq q$ if $d$ is sufficiently large, which we additionally assume from now on.

The lower bound of $\mathbf{Tr}(\mathbf{S}_3)$ continues as follows:

$$\mathbf{Tr}(\mathbf{S}_3) \geq n(\xi_{L+2}^{(q)})^2 \sum_{\substack{|S|=L+2 \\ \gamma(S) \leq q}} \left(\frac{q+1-\gamma(S)}{d\mathcal{B}(L+2,q)}\right)^2$$

$$\geq n(\xi_{L+2}^{(q)})^2 \sum_{\substack{|S|=L+2 \\ \gamma(S) \leq \lfloor q/2 \rfloor}} \left(\frac{q+1-\gamma(S)}{d\mathcal{B}(L+2,q)}\right)^2$$

$$\overset{(iii)}{\geq} (\xi_{L+2}^{(q)})^2 \frac{n \cdot q/2}{d\mathcal{B}(L+2,q)} \sum_{\substack{|S|=L+2 \\ \gamma(S) \leq \lfloor q/2 \rfloor}} \frac{\lfloor q/2 \rfloor + 1 - \gamma(S)}{d\mathcal{B}(L+2,q)}$$

$$\overset{(iv)}{\gtrsim} (\xi_{L+2}^{(q)})^2 \frac{n \cdot q/2}{d\mathcal{B}(L+2,q)} \sum_{\substack{|S|=L+2 \\ \gamma(S) \leq \lfloor q/2 \rfloor}} \frac{\lfloor q/2 \rfloor + 1 - \gamma(S)}{d\mathcal{B}(L+2,\lfloor q/2 \rfloor)},$$

where $(iii)$ follows from $q \geq \lfloor q/2 \rfloor$ and $q+1-\gamma(S) \geq q/2$. Step $(iv)$ follows from the fact that $\mathcal{B}(L+2,q)$ and $\mathcal{B}(L+2,\lfloor q/2 \rfloor)$ are of the same order; this follows from a classical bound on the binomial coefficient:

$$\mathcal{B}(L+2,q) \leq \left(\frac{eq}{L+2}\right)^{L+2} \lesssim \left(\frac{\lfloor q/2 \rfloor}{L+2}\right)^{L+2} \leq \mathcal{B}(L+2,\lfloor q/2 \rfloor).$$

We conclude the lower bound on $\mathbf{Tr}(\mathbf{S}_3)$ as follows:

$$
\begin{aligned}
\mathbf{Tr}(\mathbf{S}_3) &\gtrsim (\xi_{L+2}^{(q)})^2 \frac{n \cdot q/2}{d\mathcal{B}(L+2, q)} \sum_{\substack{|S|=L+2 \\ \gamma(S) \leq \lfloor q/2 \rfloor}} \frac{\lfloor q/2 \rfloor + 1 - \gamma(S)}{d\mathcal{B}(L+2, \lfloor q/2 \rfloor)} \\
&\stackrel{(v)}{=} (\xi_{L+2}^{(q)})^2 \frac{n \cdot q/2}{d\mathcal{B}(L+2, q)} \mathcal{Q}_{L+2}^{(d, \lfloor q/2 \rfloor)}(x, x) \\
&\stackrel{(vi)}{=} (\xi_{L+2}^{(q)})^2 \frac{n \cdot q/2}{d\mathcal{B}(L+2, q)} \\
&\stackrel{(vii)}{\in} \Omega\left( \frac{n \cdot q}{d\mathcal{B}(L+2, q)} \right) \\
&\stackrel{(viii)}{=} \Omega\left( \frac{d \cdot d^{\ell_\lambda} q^{L+\delta} \cdot q}{d \cdot q^{L+2}} \right) = \Omega\left( d^{\ell_\lambda} q^{-(1-\delta)} \right),
\end{aligned}
\tag{37}
$$

where $(v)$ uses $\mathcal{Y}_S(x)\mathcal{Y}_S(x) = 1$ for all $S$ and $x \in \{-1, 1\}^d$ with the definition of $\mathcal{Q}_{L+2}^{(d, \lfloor q/2 \rfloor)}$ in Equation (30), $(vi)$ applies Lemma 15 with $\lfloor q/2 \rfloor$ as filter size, $(vii)$ follows from Equation (18) in Assumption 1, and $(viii)$ uses the classical bound on the binomial coefficient.

Finally, for the upper bound on $m\|\mathbf{S}_3\|$, we have

$$
\begin{aligned}
m\|\mathbf{S}_3\| &= m\|\mathbf{\Psi}_3 \mathbf{D}_3 \mathbf{\Psi}_3^\intercal\| \leq mn\|\mathbf{D}_3\| \left\| \frac{\mathbf{\Psi}_3^\intercal \mathbf{\Psi}_3}{n} \right\| \stackrel{(i)}{\leq} 1.5mn\|\mathbf{D}_3\| = mn \max_{k \in \mathcal{I}_3} \lambda_k^2 \\
&\stackrel{(ii)}{=} mn \max_{k \in \mathcal{I}_3} \left( \xi_{L+2}^{(q)} \frac{q + 1 - \gamma(S_k)}{d\mathcal{B}(L+2, q)} \right)^2 \leq \frac{m}{n} (\xi_{L+2}^{(q)})^2 \left( \frac{n \cdot q}{d\mathcal{B}(L+2, q)} \right)^2 \\
&\stackrel{(iii)}{\in} \mathcal{O}\left( \frac{m}{n} \left( \frac{dd^{\ell_\lambda} q^{L+\delta} \cdot q}{dq^{L+2}} \right)^2 \right) \stackrel{(iv)}{\subseteq} \mathcal{O}\left( q^{-\delta} d^{-\ell_\lambda} \left( d^{\ell_\lambda} q^{-(1-\delta)} \right)^2 \right) \\
&= \mathcal{O}\left( q^{-\delta} q^{-(1-\delta)} \left( d^{\ell_\lambda} q^{-(1-\delta)} \right) \right) \stackrel{(v)}{\subseteq} o\left( d^{\ell_\lambda} q^{-(1-\delta)} \right) \stackrel{(vi)}{\subseteq} o(\mathbf{Tr}(\mathbf{S}_3)),
\end{aligned}
$$

where $(i)$ follows with high probability from Equation (35), and $(ii)$ from Proposition 1. Step $(iii)$ uses that Equations (18) and (21) in Assumption 1 yield $\xi_{L+2}^{(q)} \in \Theta(1)$, and $\mathcal{B}(L+2, q) = \binom{q}{L+2} \in \mathcal{O}(q^{L+2})$. Furthermore, $(iv)$ follows from $m \in \mathcal{O}\left( \frac{nq^{-\delta}}{\max\{\lambda, 1\}} \right)$, $(v)$ from $q^{-\delta} q^{-(1-\delta)} \in o(1)$, and $(vi)$ from the lower bound on $\mathbf{Tr}(\mathbf{S}_3)$ in Equation (37).

Finally, combining this result with Equations (36) and (37), we have

$$
\sum_{i=1+m}^{n} \mu_i(\mathbf{S}_{>m}) \geq \mathbf{Tr}(\mathbf{S}_3) - m\|\mathbf{S}_3\| \in \Omega(\mathbf{Tr}(\mathbf{S}_3)) \subseteq \Omega\left( d^{\ell_\lambda} q^{-(1-\delta)} \right)
$$

with probability at least $1 - \tilde{c}_1 q^{-\bar\delta} \geq 1 - cq^{-\min\{\bar\delta, 1-\bar\delta\}}$.

## C.4 TECHNICAL LEMMAS

We use the following technical lemmas in the proof of Lemma 3. All results assume the setting of Appendix C.2, particularly a kernel as in Theorem 1 that satisfies Assumption 1.

**Lemma 12** (Bound on $\mathbf{K}_1$). *In the setting of Appendix C.2, for $d/2 > q \geq \bar{L} + 1$, we have*

$$
\left\| \frac{\mathbf{\Psi}_1^\intercal \mathbf{\Psi}_1}{n} - \mathbf{I}_{|\mathcal{I}_1|} \right\| \leq 1/2
$$

*with probability at least $1 - cq^{-\bar\delta}$ uniformly over all choices of $m$ and $\lambda$.*

*Proof.* The proof follows a very similar argument to Lemma 1 with minor modifications. First, define

$$
\hat{\mathcal{I}}_1 := \{k \in \mathbb{N} \mid |S_k| \leq \bar{L} + 1\}.
$$

Note that $\hat{\mathcal{I}}_1$ does not depend on $m$ or $\lambda$, and $\mathcal{I}_1 \subseteq \hat{\mathcal{I}}_1$ for any $m$. Furthermore,

$$|\hat{\mathcal{I}}_1| = \sum_{l=0}^{\bar{L}+1} \mathcal{C}(l,q,d) \overset{(i)}{=} 1 + \sum_{l=1}^{\bar{L}+1} d\binom{q-1}{l-1} \overset{(ii)}{\leq} 1 + d\sum_{l=1}^{\bar{L}+1} \left(\frac{e(q-1)}{l-1}\right)^{l-1} \overset{(iii)}{\in} \mathcal{O}(d \cdot q^{\bar{L}}),$$

where $(i)$ follows from Lemma 9, $(ii)$ is a classical bound on the binomial coefficient, and $(iii)$ follows from the fact that the largest degree monomial dominates the others.

Next, we define $\hat{\boldsymbol{\Psi}}_1 \in \mathbb{R}^{n \times |\hat{\mathcal{I}}_1|}$ with $[\hat{\boldsymbol{\Psi}}_1]_{i,j} = \mathcal{Y}_{S_{k_j}}(x_i)$ where $k_j$ is the $j$-th element in $\hat{\mathcal{I}}_1$. Using the same arguments as in the proof of Lemma 1, it follows that the rows of $\hat{\boldsymbol{\Psi}}_1$ are independent, that their norm is bounded by $|\hat{\mathcal{I}}_1|$, and that they have an expected outer product equal to $\mathbf{I}_{|\hat{\mathcal{I}}_1|}$.

Hence, as in the proof of Lemma 1, we can apply Theorem 5.44 from Vershynin (2012). Choosing $t = \frac{1}{2}\sqrt{\frac{n}{|\hat{\mathcal{I}}_1|}}$ yields

$$\left\| \frac{\hat{\boldsymbol{\Psi}}_1^\intercal \hat{\boldsymbol{\Psi}}_1}{n} - \mathbf{I}_{|\hat{\mathcal{I}}_1|} \right\| \leq 1/2$$

with probability at least $1 - cq^{-\bar{\delta}}$ for some absolute constant $c$.

Finally, since $\mathcal{I}_1 \subseteq \hat{\mathcal{I}}_1$ for all choices of $\lambda$ and $m$,

$$\left\| \frac{\boldsymbol{\Psi}_1^\intercal \boldsymbol{\Psi}_1}{n} - \mathbf{I}_{|\mathcal{I}_1|} \right\| \leq \left\| \frac{\hat{\boldsymbol{\Psi}}_1^\intercal \hat{\boldsymbol{\Psi}}_1}{n} - \mathbf{I}_{|\hat{\mathcal{I}}_1|} \right\| \leq 1/2$$

with probability at least $1 - cq^{-\bar{\delta}}$ uniformly over all $\lambda$ and $m$. $\qquad \square$

**Lemma 13** (Bound on $\mathbf{K}_2$). *In the setting of Appendix C.2, if $c < \lfloor q/2 \rfloor < q < d/2$ for $c$ as in Assumption 1, we have*

$$c_1 \leq \mu_{\min}(\mathbf{K}_2) \leq \|\mathbf{K}_2\| \leq c_2$$

*for some positive constants $c_1, c_2$ with probability at least $1 - \tilde{c}q^{-(1-\bar{\delta})}$.*

*Proof.* First, the condition on $d$ implies $q > T$ and ensures that we can apply Lemmas 16 and 17, and Assumption 1. Furthermore, note that $T - 1 > \bar{L} + 2$.

Proposition 1 and the definition of $\mathcal{K}_2$ yield the following decomposition:

$$\mathcal{K}_2(x, x') = \sum_{l=\bar{L}+2}^{q} \xi_l^{(q)} \sum_{\substack{\gamma(S) \leq q \\ |S| = l}} \frac{q + 1 - \gamma(S)}{d\mathcal{B}(l,q)} \mathcal{Y}_S(x) \mathcal{Y}_S(x') = \sum_{l=\bar{L}+2}^{q} \xi_l^{(q)} \mathcal{Q}_l^{(d,q)}(x, x'),$$

where $\mathcal{Q}_l^{(d,q)}(x, x')$ is defined in Equation (30). Then, using the triangle inequality with non-negativity of the $\xi_l^{(q)}$ from Equations (18) and (19) in Assumption 1, we have

$$
\begin{aligned}
\|\mathbf{K}_2\| &= \left\| \sum_{l=\bar{L}+2}^{q} \xi_l^{(q)} \mathbf{Q}_l^{(d,q)} \right\| = \left\| \sum_{l=\bar{L}+2}^{T-1} \xi_l^{(q)} \mathbf{Q}_l^{(d,q)} + \sum_{l=T}^{q} \xi_l^{(q)} \mathbf{Q}_l^{(d,q)} \right\| \\
&\leq \sum_{l=\bar{L}+2}^{T-1} \xi_l^{(q)} \left\| \mathbf{Q}_l^{(d,q)} \right\| + \left\| \sum_{l=T}^{q} \xi_l^{(q)} \mathbf{Q}_l^{(d,q)} \right\| \\
&\overset{(i)}{\leq} 1.5 \sum_{l=\bar{L}+2}^{T-1} \xi_l^{(q)} + c_3 \overset{(ii)}{\leq} c_2 \quad \text{with probability} \geq 1 - cq^{-(1-\bar{\delta})}.
\end{aligned}
$$

$\mathbf{Q}_l^{(d,q)}$ is the kernel matrix corresponding to $\mathcal{Q}_l^{(d,q)}(x, x')$, $(i)$ uses Lemmas 16 and 17, and $(ii)$ uses non-negativity and additionally Equation (21) in Assumption 1.

The lower bound follows similarly from

$$\mu_{\min}\left(\mathbf{K}_2\right) \geq \sum_{l=\bar{L}+2}^{q} \xi_l^{(q)} \mu_{\min}\left(\mathbf{Q}_l^{(d,q)}\right) \geq \xi_{\bar{L}+2}^{(q)} \mu_{\min}\left(\mathbf{Q}_{\bar{L}+2}^{(d,q)}\right) \overset{(i)}{\geq} \frac{1}{2}\xi_{\bar{L}+2}^{(q)} \overset{(ii)}{\geq} c_1,$$

where $(i)$ follows from Lemma 16 with probability at least $1 - cq^{-(1-\bar{\delta})}$, and $(ii)$ from Equation (18) in Assumption 1.

Since Lemma 16 yields both the upper and lower bound for all $l$ uniformly with probability $1 - cq^{-(1-\bar{\delta})}$, this concludes the proof. $\qquad\square$

**Lemma 14** (Bound on $\mathbf{Tr}(\mathbf{S}_2)$)**.** *In the setting of Appendix C.2, if $c < \lfloor q/2 \rfloor < q < d/2$ for $c$ as in Assumption 1, we have with probability at least $1 - \tilde{c}q^{-(1-\bar{\delta})}$ that*

$$\mathbf{Tr}(\mathbf{S}_2) \in \Theta(q^{-(1-\bar{\delta})}) \quad and \quad \|\mathbf{S}_2\| \in \mathcal{O}\left(\frac{1}{dq^{\bar{L}+1}}\right),$$

*where $\mathbf{S}_2$ is defined in Appendix C.2.*

*Proof.* Throughout the proof, the conditions on $d$ and hence $q \in \Theta(d^\beta)$ ensure that we can apply Assumption 1 and Lemma 16, as well as $\bar{L} + 2 \leq T < \lfloor q/2 \rfloor$.

First, Lemma 13 yields $\|\mathbf{K}_2\| \in \Theta(1)$ with probability at least $1 - c'q^{-(1-\bar{\delta})}$, which we will use throughout the proof. Next, we bound $\|\mathbf{S}_2\|$ in two steps. For this, remember that $\lambda_{\bar{m}}$ is the largest eigenvalue corresponding to an eigenfunction $\mathcal{Y}_S$ of degree $|S| \geq \bar{L} + 2$.

**Proof that $\|\mathbf{S}_2\| \leq \lambda_{\bar{m}}\|\mathbf{K}_2\|$** Define $\tilde{\mathbf{\Psi}}_k \in \mathbb{R}^{n \times n}$ with $[\tilde{\mathbf{\Psi}}_k]_{i,j} := \mathcal{Y}_{S_k}(x_i)\mathcal{Y}_{S_k}(x_j)$ for all $i, j \in \{1, \ldots, n\}$, $k \in \mathcal{I}_2$, and let $v$ be any vector in $\mathbb{R}^n$. Then,

$$\|\mathbf{S}_2 v\| \overset{(i)}{=} \left\|\left(\sum_{k \in \mathcal{I}_2} \lambda_k^2 \tilde{\mathbf{\Psi}}_k\right) v\right\|$$

$$\leq \left\|\max_{k \in \mathcal{I}_2}\{\lambda_k\} \sum_{k \in \mathcal{I}_2} \lambda_k \tilde{\mathbf{\Psi}}_k v\right\|$$

$$\overset{(ii)}{=} \lambda_{\bar{m}} \left\|\left(\sum_{k \in \mathcal{I}_2} \lambda_k \tilde{\mathbf{\Psi}}_k\right) v\right\|$$

$$= \lambda_{\bar{m}}\|\mathbf{K}_2 v\|,$$

where $(i)$ follows from the definition of $\mathcal{S}_2$ and $(ii)$ from the definition of $\bar{m}$.

**Proof that $\lambda_{\bar{m}} \in \Theta\left(\frac{1}{dq^{\bar{L}+1}}\right)$** We show that $\lambda_k \in \mathcal{O}\left(\frac{1}{dq^{\bar{L}+1}}\right)$ for all $k \in \mathcal{I}_2$, and that there exists $\tilde{m} \in \mathcal{I}_2$ with $\lambda_{\tilde{m}} \in \Omega\left(\frac{1}{dq^{\bar{L}+1}}\right)$. Since $\lambda_{\bar{m}} = \max_{k \in \mathcal{I}_2} \lambda_{S_k}$, those two facts imply $\lambda_{\bar{m}} \in \Theta\left(\frac{1}{dq^{\bar{L}+1}}\right)$.

Let $k \in \mathcal{I}_2$ be arbitrary. Lemma 10 yields

$$\lambda_k \in \begin{cases} \mathcal{O}\left(\frac{1}{dq^{|S_k|-1}}\right) & |S_k| < T \\ \mathcal{O}\left(\frac{1}{dq^{T-1}}\right) & |S_k| \geq T \end{cases}$$

$$\overset{(i)}{\subseteq} \mathcal{O}\left(\frac{1}{dq^{\bar{L}+1}}\right),$$

where $(i)$ follows from $|S_k| \geq \bar{L} + 2$ and $T \geq \bar{L} + 2$.

Now we show that there exists $\tilde{m}$ with $\lambda_{\tilde{m}} \in \Omega\left(\frac{1}{dq^{\bar{L}+1}}\right)$. We choose $\tilde{m}$ with $S_{\tilde{m}} = \{1, 2, \ldots, \bar{L}+2\}$. Note that $\bar{L} + 2 = \gamma(S_{\tilde{m}}) \leq q$ and thus $\tilde{m} \in \mathcal{I}_2$. Next, Proposition 1 yields

$$\lambda_{S_{\tilde{m}}} = \xi_{\bar{L}+2}^{(q)} \frac{(q+1-\gamma(S_{\tilde{m}}))}{d\mathcal{B}(\bar{L}+2, q)} \overset{(i)}{\in} \Theta\left(\frac{q}{dq^{\bar{L}+2}}\right) \subseteq \Theta\left(\frac{1}{dq^{\bar{L}+1}}\right),$$

where $(i)$ follows from Equations (18) and (21) in Assumption 1.

Finally, combining the previous two results and $\|\mathbf{K}_2\| \in \Theta(1)$, we have

$$\|\mathbf{S}_2\| \leq \lambda_{\bar{m}}\|\mathbf{K}_2\| \in \mathcal{O}\left(\lambda_{\bar{m}}\right) = \mathcal{O}\left(\frac{1}{dq^{\bar{L}+1}}\right).$$

**Upper bound of $\mathbf{Tr}(\mathbf{S}_2)$**   The upper bound also follows directly from the last two results:

$$\mathbf{Tr}(\mathbf{S}_2) \leq n\|\mathbf{S}_2\| \in \mathcal{O}\left(\frac{n}{dq^{\bar{L}+1}}\right) = \mathcal{O}\left(\frac{dq^{\bar{L}+\bar{\delta}}}{dq^{\bar{L}+1}}\right) = \mathcal{O}(q^{-(1-\bar{\delta})}).$$

**Lower bound of $\mathbf{Tr}(\mathbf{S}_2)$**   The lower bound requires a more refined argument.

$$\begin{aligned}
\mathcal{S}_2(x, x') = \sum_{k \in \mathcal{I}_2} \lambda_{S_k}^2 \mathcal{Y}_{S_k}(x)\mathcal{Y}_{S_k}(x') &\geq \sum_{\substack{\gamma(S) \leq q \\ |S|=\bar{L}+2}} \lambda_S^2 \mathcal{Y}_S(x)\mathcal{Y}_S(x') \\
&\geq \sum_{\substack{\gamma(S) \leq \lfloor q/2 \rfloor \\ |S|=\bar{L}+2}} \lambda_S^2 \mathcal{Y}_S(x)\mathcal{Y}_S(x') \\
&\stackrel{(i)}{=} (\xi_{\bar{L}+2}^{(q)})^2 \sum_{\substack{\gamma(S) \leq \lfloor q/2 \rfloor \\ |S|=\bar{L}+2}} \frac{(q+1-\gamma(S))^2}{d^2 \mathcal{B}(\bar{L}+2, q)^2} \mathcal{Y}_S(x)\mathcal{Y}_S(x') \\
&= \frac{(\xi_{\bar{L}+2}^{(q)})^2}{d\mathcal{B}(\bar{L}+2, q)} \sum_{\substack{\gamma(S) \leq \lfloor q/2 \rfloor \\ |S|=\bar{L}+2}} (q+1-\gamma(S))\frac{q+1-\gamma(S)}{d\mathcal{B}(\bar{L}+2, q)} \mathcal{Y}_S(x)\mathcal{Y}_S(x') \\
&\stackrel{(ii)}{\geq} \frac{(\xi_{\bar{L}+2}^{(q)})^2 q/2}{d\mathcal{B}(\bar{L}+2, q)} \sum_{\substack{\gamma(S) \leq \lfloor q/2 \rfloor \\ |S|=\bar{L}+2}} \frac{\lfloor q/2 \rfloor + 1 - \gamma(S)}{d\mathcal{B}(\bar{L}+2, q)} \mathcal{Y}_S(x)\mathcal{Y}_S(x'),
\end{aligned}$$

where $(i)$ follows from Proposition 1. In $(ii)$, we use that, as long as $\gamma(S) \leq \lfloor q/2 \rfloor$, we have $q+1-\gamma(S) \geq \frac{q}{2}$ and $q \geq \lfloor q/2 \rfloor$. Continuing the bound, we have

$$\begin{aligned}
\mathcal{S}_2(x, x') &\geq \frac{(\xi_{\bar{L}+2}^{(q)})^2 q/2}{d\mathcal{B}(\bar{L}+2, q)} \sum_{\substack{\gamma(S) \leq \lfloor q/2 \rfloor \\ |S|=\bar{L}+2}} \frac{\lfloor q/2 \rfloor + 1 - \gamma(S)}{d\mathcal{B}(\bar{L}+2, q)} \mathcal{Y}_S(x)\mathcal{Y}_S(x') \\
&\stackrel{(iii)}{\geq} c'_{\bar{L}} \frac{(\xi_{\bar{L}+2}^{(q)})^2 q/2}{d\mathcal{B}(\bar{L}+2, q)} \sum_{\substack{\gamma(S) \leq \lfloor q/2 \rfloor \\ |S|=\bar{L}+2}} \frac{\lfloor q/2 \rfloor + 1 - \gamma(S)}{d\mathcal{B}(\bar{L}+2, \lfloor q/2 \rfloor)} \mathcal{Y}_S(x)\mathcal{Y}_S(x') \\
&\stackrel{(iv)}{=} c'_{\bar{L}} \frac{(\xi_{\bar{L}+2}^{(q)})^2 q/2}{d\mathcal{B}(\bar{L}+2, q)} \mathcal{Q}_{\bar{L}+2}^{(d, \lfloor q/2 \rfloor)}(x, x').
\end{aligned}$$

In $(iii)$, we use the classical bound on the binomial coefficient $\binom{q}{\bar{L}+2} = \mathcal{B}(\bar{L}+2, q)$ as follows:

$$\mathcal{B}(\bar{L}+2, q) \leq (2e)^{\bar{L}+2}\left(\frac{q/2}{\bar{L}+2}\right)^{\bar{L}+2} \leq (2e)^{\bar{L}+2}c_{\bar{L}}\left(\frac{\lfloor q/2 \rfloor}{\bar{L}+2}\right)^{\bar{L}+2} = c'_{\bar{L}}\mathcal{B}(\bar{L}+2, \lfloor q/2 \rfloor),$$

where $c'_{\bar{L}}$ is a constant that depends only on $\bar{L}$. Finally, $(iv)$ follows from the definition of $\mathcal{Q}_{\bar{L}+2}^{(d, \lfloor q/2 \rfloor)}$ in Equation (30).

The bound on $\mathcal{S}_2(x, x')$ implies

$$\mu_{\min}(\mathbf{S}_2) \geq c'_{\bar{L}} \frac{(\xi_{\bar{L}+2}^{(q)})^2 q/2}{d\mathcal{B}(\bar{L}+2, q)} \mu_{\min}\left(\mathbf{Q}_{\bar{L}+2}^{(d, \lfloor q/2 \rfloor)}\right),$$

and allows us to ultimately lower-bound $\mathbf{Tr}(\mathbf{S}_2)$ as follows:

$$
\mathbf{Tr}(\mathbf{S}_2) \geq n\mu_{\min}(\mathbf{S}_2) \geq c'_{\bar{L}} n \frac{(\xi^{(q)}_{\bar{L}+2})^2 q/2}{d\mathcal{B}(\bar{L}+2, q)} \mu_{\min}\left(\mathbf{Q}^{(d,\lfloor q/2\rfloor)}_{\bar{L}+2}\right)
$$

$$
\overset{(i)}{\gtrsim} c'_{\bar{L}} n \frac{(\xi^{(q)}_{\bar{L}+2})^2 q/2}{d\mathcal{B}(\bar{L}+2, q)}
$$

$$
\overset{(ii)}{\gtrsim} d^\ell \frac{q}{dq^{\bar{L}+2}} = \frac{dq^{\bar{L}+\bar{\delta}}}{dq^{\bar{L}+1}} = q^{-(1-\bar{\delta})},
$$

where $(i)$ follows from the lower bound on $\mu_{\min}\left(\mathbf{Q}^{(d,\lfloor q/2\rfloor)}_{\bar{L}+2}\right)$ in Lemma 16 with probability at least $1 - c''\lfloor q/2\rfloor^{-(1-\bar{\delta})} \geq 1 - c''q^{-(1-\bar{\delta})}$, and $(ii)$ follows from Equation (18) in Assumption 1 and the classical lower bound on the binomial coefficient. This yields the desired lower bound $\mathbf{Tr}(\mathbf{S}_2) \in \Omega(q^{-(1-\bar{\delta})})$.

Finally, collecting all error probabilities concludes the proof. $\qquad\square$

**Lemma 15** (Diagonal elements of $\mathcal{Q}^{(d,q)}$)**.** *Let $l, q, d \in \mathbb{N}$ with $0 < l \leq q < d/2$ and $x \in \{-1, 1\}^d$. Then,*

$$
\mathcal{Q}^{(d,q)}_l(x, x) = 1.
$$

*Proof.* First,

$$
\mathcal{Q}^{(d,q)}_l(x, x) = \sum_{\substack{\gamma(S)\leq q \\ |S|=l}} \frac{q+1-\gamma(S)}{d\mathcal{B}(l,q)} \mathcal{Y}_S(x)^2
$$

$$
\overset{(i)}{=} \frac{1}{d\mathcal{B}(l,q)} \sum_{\substack{\gamma(S)\leq q \\ |S|=l}} q+1-\gamma(S)
$$

$$
= \frac{1}{d\mathcal{B}(l,q)} \sum_{\gamma=l}^{q} (q+1-\gamma) \sum_{\substack{\gamma(S)=\gamma \\ |S|=l}} 1
$$

where $(i)$ follows from the fact that $\mathcal{Y}_S(x)^2 = 1$.

Note that $\sum_{\substack{\gamma(S)=\gamma \\ |S|=l}} 1$ matches the definition of $\tilde{\mathcal{C}}(l, \gamma, d)$ in the proof of Lemma 9. Next, we use the following recurrence:

$$
\sum_{\gamma=l}^{q} (q+1-\gamma)\tilde{\mathcal{C}}(l,\gamma,d) = \sum_{\gamma=l}^{q} \left(\tilde{\mathcal{C}}(l,\gamma,d) + ((q-1)+1-\gamma)\tilde{\mathcal{C}}(l,\gamma,d)\right)
$$

$$
= \mathcal{C}(l,q,d) + 0 + \sum_{\gamma=l}^{q-1} ((q-1)+1-\gamma)\tilde{\mathcal{C}}(l,\gamma,d),
$$

where the last step uses the fact that $\mathcal{C}(l,q,d) = \sum_{\gamma=l}^{q} \tilde{\mathcal{C}}(l,\gamma,d)$ by definition, and that the term corresponding to $\gamma = q$ in the second sum is zero. Recursively applying this formula $q - l$ times yields

$$
\sum_{\gamma=l}^{q} (q+1-\gamma)\tilde{\mathcal{C}}(l,\gamma,d) = \sum_{\gamma=l}^{q} \mathcal{C}(l,\gamma,d).
$$

Using this identity, we finally get

$$
\mathcal{Q}^{(d,q)}_l(x,x) = \frac{1}{d\mathcal{B}(l,q)} \sum_{\gamma=l}^{q} \mathcal{C}(l,\gamma,d) \overset{(i)}{=} \frac{1}{d\mathcal{B}(l,q)} \sum_{\gamma=l}^{q} d\binom{\gamma-1}{l-1} \overset{(ii)}{=} \frac{1}{d\mathcal{B}(l,q)} d\binom{q}{l} \overset{(iii)}{=} 1,
$$

where $(i)$ follows from Lemma 9, $(ii)$ from the hockey-stick identity, and $(iii)$ from the definition of $\mathcal{B}(l,q)$ in Equation (15). $\qquad\square$

## C.5 RANDOM MATRIX THEORY LEMMAS

We use the following lemmas related to random matrix theory in the proof of Lemma 3. The first two results bound the kernel's intermediate and late eigenvalues.

**Lemma 16** (Bound on the kernel's intermediate tail). *In the setting of Appendix C.2, for $T - 1 \leq q < d/2$, with probability at least $1 - cq^{-(1-\bar{\delta})}$, all $l \in \mathbb{N}$ with $\bar{L} + 2 \leq l < T$ satisfy*

$$\|\mathbf{Q}_l^{(d,q)} - \mathbf{I}_n\| \leq \frac{1}{2},$$

*where $T$ is defined in Assumption 1 and $\mathbf{Q}$ in Equation (30).*

This lemma particularly implies $\|\mathbf{Q}_l^{(d,q)}\| \leq 1.5$ and $\mu_{\min}\left(\mathbf{Q}_l^{(d,q)}\right) \geq 1/2$ for all $\bar{L} + 2 \leq l < T$ with high probability.

**Lemma 17** (Bound on the kernel's late tail). *In the setting of Appendix C.2, if $c \leq \lfloor q/2 \rfloor - 1 < q < d/2$, for $c$ and $T$ as in Assumption 1, we have*

$$\left\| \sum_{l=T}^{q} \xi_l^{(q)} \mathbf{Q}_l^{(d,q)} \right\| \lesssim 1$$

*with probability at least $1 - \tilde{c}d^{-1}$.*

*Proof of Lemma 16.* First, Lemma 15 yields that the diagonal elements of $\mathbf{Q}_l^{(d,q)}$ are just 1. Hence, we define $\mathbf{W}_l^{(d,q)} := \mathbf{Q}_l^{(d,q)} - \mathbf{I}_n$, and want to show that, with high probability, $\|\mathbf{W}_l^{(d,q)}\| \leq 1/2$ for all $\bar{L} + 2 \leq l < T$ at the same time.

The proof makes use of Lemma 18. Therefore, we need to find an appropriate $\mathcal{M}^{(l,q)}$ for each considered $l$, and show that the conditions of the lemma hold.

The first condition follows directly from the construction of $\mathbf{W}_l^{(d,q)}$ and $\mathrm{diag}(\mathbf{Q}_l^{(d,q)}) = \mathbf{I}_n$.

To establish the second condition, we have for all $i \neq j, k \neq j$

$$\left| \mathbb{E}_{x_j} \left[ [\mathbf{W}_l^{(d,q)}]_{i,j} [\mathbf{W}_l^{(d,q)}]_{j,k} \right] \right|$$

$$= \left| \frac{1}{(d\mathcal{B}(l,q))^2} \sum_{\substack{\gamma(S) \leq q \\ |S| = l}} \sum_{\substack{\gamma(S') \leq q \\ |S'| = l}} (q + 1 - \gamma(S))(q + 1 - \gamma(S')) \mathcal{Y}_S(x_i) \mathbb{E}\left[ \mathcal{Y}_S(x_j) \mathcal{Y}_{S'}(x_j) \right] \mathcal{Y}_{S'}(x_k) \right|$$

$$\overset{(i)}{=} \frac{1}{(d\mathcal{B}(l,q))^2} \left| \sum_{\substack{\gamma(S) \leq q \\ |S| = l}} (q + 1 - \gamma(S))^2 \mathcal{Y}_S(x_i) \mathcal{Y}_S(x_k) \right|$$

$$\leq \frac{q}{d\mathcal{B}(l,q)} \frac{1}{d\mathcal{B}(l,q)} \left| \sum_{\substack{\gamma(S) \leq q \\ |S| = l}} (q + 1 - \gamma(S)) \mathcal{Y}_S(x_i) \mathcal{Y}_S(x_k) \right|$$

$$= \left( \frac{d\mathcal{B}(l,q)}{q} \right)^{-1} \left| \left[ \mathbf{W}_l^{(d,q)} \right]_{i,k} \right| \leq \left( \frac{d\mathcal{B}(l,q)}{lq} \right)^{-1} \left| \left[ \mathbf{W}_l^{(d,q)} \right]_{i,k} \right|,$$

where $(i)$ follows from orthogonality of the eigenfunctions. Hence,

$$\mathcal{M}^{(l,q)} = d\mathcal{B}(l,q)/lq$$

satisfies the second condition in Lemma 18 for all $\bar{L} + 2 \leq l < T$.

The extra $l$ factor is necessary for the third condition to hold. As in Lemma 18, let $p \in \mathbb{N}, p \geq 2$. Then, for all $i \neq j$,

$$
\mathbb{E}\left[|[\mathbf{W}_l^{(d,q)}]_{i,j}|^p\right]^{\frac{1}{p}} \overset{(i)}{\leq} (p-1)^{l/2}\sqrt{\mathbb{E}\left[[\mathbf{W}_l^{(d,q)}]_{i,j}^2\right]} \overset{(ii)}{\leq} \sqrt{p^l \sum_{\substack{\gamma(S) \leq q \\ |S|=l}} \frac{(q+1-\gamma(S))^2}{(d\mathcal{B}(l,q))^2}}
$$

$$
\overset{(iii)}{\leq} \sqrt{p^l \frac{q^2}{(d\mathcal{B}(l,q))^2}\mathcal{C}(l,q,d)} = \sqrt{p^l \frac{lq}{d\mathcal{B}(l,q)}}\sqrt{\frac{q\mathcal{C}(l,q,d)}{ld\mathcal{B}(l,q)}} \overset{(iv)}{=} \sqrt{\frac{p^l}{\mathcal{M}^{(l,q)}}},
$$

where $(i)$ follows from hypercontractivity in Lemma 19, $(ii)$ from orthogonality of the eigenfunctions, and $(iii)$ from the definition of $\mathcal{C}(l,q,d)$ as well as $1 \leq \gamma(S) \leq q$. Step $(iv)$ follows from Lemma 9, the definition of $\mathcal{B}(l,q)$ in Equation (15), $\binom{q-1}{l-1}\frac{q}{l} = \binom{q}{l}$, and the definition of $\mathcal{M}^{(l,q)}$.

Since all conditions are satisfied, Lemma 18 yields for all $p \in \mathbb{N}, p > 2$

$$
\Pr\left(\|\mathbf{W}_l^{(d,q)}\| > 1/2\right) \leq c_{l,1}^p p^{3p} n \left(\frac{n}{\mathcal{M}^{(l,q)}}\right)^p + c_{l,2}^p\left(\frac{n}{\mathcal{M}^{(l,q)}}\right)^2,
$$

where $c_{l,1}, c_{l,2}$ are positive constants that depend on $l$. In particular, if $p \geq 2 + \frac{\frac{1}{\beta}+\bar{L}+\bar{\delta}}{1-\delta}$, then we get

$$
\Pr\left(\|\mathbf{W}_l^{(d,q)}\| > 1/2\right) \leq c_l q^{-2(1-\bar{\delta})},
$$

where $c_l$ is a positive constant that depends on $l$. To avoid this dependence, we can take the union bound over all $l = \bar{L}+2, \ldots, T-1$:

$$
\Pr\left(\exists l \in \{\bar{L}+2, \ldots, T-1\}\colon \|\mathbf{W}_l^{(d,q)}\| > 1/2\right) \leq q^{-2(1-\bar{\delta})}\sum_{l=\bar{L}+2}^{T-1} c_l \leq c'_{\bar{L}} q^{-2(1-\bar{\delta})},
$$

where $c'_{\bar{L}}$ only depends on $\bar{L}$ and $T$, which are fixed in our setting. Finally, additionally note that neither $\bar{L}$ nor $T$ depend on $\ell_\lambda$. $\qquad\square$

*Proof of Lemma 17.* First, Lemma 15 yields that the diagonal elements of $\mathbf{Q}_l^{(d,q)}$ are just 1 for all $l \in \{T, \ldots, q\}$. Hence, we define $\mathbf{W}_l^{(d,q)} := \mathbf{Q}_l^{(d,q)} - \mathbf{I}_n$, and decompose the kernel matrix as

$$
\sum_{l=T}^{q}\xi_l^{(q)}\mathbf{Q}_l^{(d,q)} = \sum_{l=T}^{q}\xi_l^{(q)}\mathbf{W}_l^{(d,q)} + \sum_{l=T}^{q}\xi_l^{(q)}\mathbf{I}_n.
$$

We can hence apply the triangle inequality to bound the norm as follows:

$$
\left\|\sum_{l=T}^{q}\xi_l^{(q)}\mathbf{Q}_l^{(d,q)}\right\| \leq \left\|\sum_{l=T}^{q}\xi_l^{(q)}\mathbf{W}_l^{(d,q)}\right\| + \left\|\sum_{l=T}^{q}\xi_l^{(q)}\mathbf{I}_n\right\|
$$

$$
\overset{(i)}{\leq} \sum_{l=T}^{q}\xi_l^{(q)}\|\mathbf{W}_l^{(d,q)}\| + \sum_{l=T}^{q}\xi_l^{(q)}\|\mathbf{I}_n\|
$$

$$
\overset{(ii)}{\leq} \sum_{l=T}^{q}\xi_l^{(q)}\sqrt{\sum_{i\neq j}^{n}[\mathbf{W}_l^{(d,q)}]_{i,j}^2} + \sum_{l=T}^{q}\xi_l^{(q)}
$$

$$
\overset{(iii)}{\leq} \sum_{l=T}^{q}\xi_l^{(q)}\underbrace{\sqrt{n^2 \max_{i\neq j}[\mathbf{W}_l^{(d,q)}]_{i,j}^2}}_{=:\varpi_l} + c''
$$

$$
\overset{(iv)}{\leq} \sum_{l=T}^{q}\frac{c'}{q} + c'' \lesssim 1, \quad \text{with probability} \geq 1 - cd^{-1},
$$

where $(i)$ uses non-negativity of the $\xi_l^{(q)}$ from Equations (18) and (19) in Assumption 1, $(ii)$ bounds the operator norm with the Frobenius norm, and $(iii)$ additionally bounds the sum of the $\xi_l^{(q)}$ using

Equation (21) in Assumption 1. Step $(iv)$ use a bound that we show in the remainder of the proof: with probability at least $1 - cd^{-1}$, we have $\varpi_l \leq c'/q$ uniformly over all $l \in \{T, \dots, q\}$.

We first bound $\varpi_l$ for a fixed $T \leq l \leq q$ as follows:

$$\Pr(\varpi_l > 1/q) = \Pr\left(\xi_l^{(q)}\sqrt{n^2 \max_{i \neq j}[\mathbf{W}_l^{(d,q)}]_{i,j}^2} > 1/q\right)$$

$$= \Pr\left(\max_{i \neq j}[\mathbf{W}_l^{(d,q)}]_{i,j}^2 > \frac{1}{n^2(\xi_l^{(q)})^2 q^2}\right) = \Pr\left(\exists i \neq j \colon [\mathbf{W}_l^{(d,q)}]_{i,j}^2 > \frac{1}{n^2(\xi_l^{(q)})^2 q^2}\right)$$

$$\overset{(i)}{\leq} n(n-1)\Pr\left([\mathbf{W}_l^{(d,q)}]_{1,2}^2 > \frac{1}{n^2(\xi_l^{(q)})^2 q^2}\right)$$

$$\overset{(ii)}{\leq} n^4(\xi_l^{(q)})^2 q^2 \mathbb{E}\left[[\mathbf{W}_l^{(d,q)}]_{1,2}^2\right]$$

$$\overset{(iii)}{=} \frac{n^4(\xi_l^{(q)})^2 q^2}{(d\mathcal{B}(l,q))^2} \sum_{\substack{\gamma(S),\gamma(S') \leq q \\ |S|,|S'|=l}} (q+1-\gamma(S))(q+1-\gamma(S')) \underbrace{\mathbb{E}\left[\mathcal{Y}_S(x_1)\mathcal{Y}_S(x_2)\mathcal{Y}_{S'}(x_1)\mathcal{Y}_{S'}(x_2)\right]}_{\delta_{S,S'}}$$

$$= \frac{n^4(\xi_l^{(q)})^2 q^2}{(d\mathcal{B}(l,q))^2} \sum_{\substack{\gamma(S) \leq q \\ |S|=l}} (q+1-\gamma(S))^2$$

$$\overset{(iv)}{\leq} \frac{(q+1-l)n^4(\xi_l^{(q)})^2 q^2}{d\mathcal{B}(l,q)} \sum_{\substack{\gamma(S) \leq q \\ |S|=l}} \frac{(q+1-\gamma(S))}{d\mathcal{B}(l,q)}\mathcal{Y}_S(x_1)^2$$

$$\leq \frac{n^4(\xi_l^{(q)})^2 q^3}{d\mathcal{B}(l,q)}\mathcal{Q}_l^{(d,q)}(x_1,x_1) \overset{(v)}{=} \frac{n^4 q^3}{d}\frac{(\xi_l^{(q)})^2}{\mathcal{B}(l,q)},$$

where $(i)$ follows from the union bound and the distribution of the off-diagonal entries in $\mathbf{W}^{(d,q)}$, and $(ii)$ from the Markov inequality. In step $(iii)$, we use orthogonality of the eigenfunctions, as well as the fact that $\mathbf{W}_l^{(d,q)}$ and $\mathbf{Q}_l^{(d,q)}$ coincide on off-diagonal entries by construction. Step $(iv)$ follows from $\mathcal{Y}_S(x)^2 = 1$ for all $S$ and $x \in \{-1,1\}^d$. Finally, step $(v)$ applies Lemma 15.

Next, we use the union bound over all $\varpi_l$ of interest:

$$\Pr\left(\exists l \in \{T, \dots, q\} \colon \varpi_l > 1/q\right) \leq \sum_{l=T}^{q} \Pr\left(\varpi_l > 1/q\right)$$

$$\leq \sum_{l=T}^{q} \frac{n^4 q^3}{d}\frac{(\xi_l^{(q)})^2}{\mathcal{B}(l,q)}$$

$$= \frac{n^4 q^3}{d}\left(\sum_{l=T}^{\lceil q/2 \rceil} \frac{(\xi_l^{(q)})^2}{\binom{q}{l}} + \sum_{l=\lceil q/2 \rceil+1}^{q-T} \frac{(\xi_l^{(q)})^2}{\binom{q}{l}} + \sum_{l=q-T+1}^{q} \frac{(\xi_l^{(q)})^2}{\binom{q}{l}}\right)$$

$$\overset{(i)}{=} \frac{n^4 q^3}{d}\left(\sum_{l=T}^{\lceil q/2 \rceil} \frac{(\xi_l^{(q)})^2}{\binom{q}{l}} + \sum_{l'=T}^{q-\lceil q/2 \rceil-1} \frac{(\xi_{q-l'}^{(q)})^2}{\binom{q}{q-l'}} + \sum_{l'=0}^{T-1} \frac{(\xi_{q-l'}^{(q)})^2}{\binom{q}{q-l'}}\right)$$

$$\overset{(ii)}{\leq} \frac{n^4 q^3}{d}\left(\underbrace{\sum_{l=T}^{\lceil q/2 \rceil} \frac{(\xi_l^{(q)})^2 + (\xi_{q-l}^{(q)})^2}{\binom{q}{l}}}_{=:E_1} + \underbrace{\sum_{l'=0}^{T-1} \frac{(\xi_{q-l'}^{(q)})^2}{\binom{q}{l'}}}_{=:E_2}\right),$$

where $(i)$ substitutes $l' = q - l$, and $(ii)$ uses $q - \lceil q/2 \rceil - 1 \leq \lceil q/2 \rceil$ as well as the fact that $\binom{q}{q-l'} = \binom{q}{l'}$. We bound both $E_1$ and $E_2$ using Assumption 1.

For $E_1$ in particular, Equations (18), (19) and (21) imply that all $\xi_l^{(q)} \lesssim 1$. Hence,

$$E_1 = \sum_{l=T}^{\lceil q/2 \rceil} \frac{(\xi_l^{(q)})^2 + (\xi_{q-l}^{(q)})^2}{\binom{q}{l}} \lesssim \sum_{l=T}^{\lceil q/2 \rceil} \frac{1}{\binom{q}{l}} \overset{(i)}{\leq} \sum_{l=T}^{\lceil q/2 \rceil} \frac{1}{\binom{q}{T}} = \frac{\lceil q/2 \rceil - T + 1}{\binom{q}{T}} \overset{(ii)}{\leq} \frac{T^T q}{q^T} \lesssim \frac{1}{q^{T-1}},$$

where $(i)$ exploits that $T$ is the value in $\{T, \ldots, \lceil q/2 \rceil\}$ the furthest away from $q/2$, and thus

$$\min_{l \in \{T, \ldots, \lceil q/2 \rceil\}} \binom{q}{l} = \binom{q}{T},$$

and $(ii)$ follows from the classical lower bound on the binomial coefficient.

For $E_2$, we have

$$E_2 = \sum_{l'=0}^{T-1} \frac{(\xi_{q-l'}^{(q)})^2}{\binom{q}{l'}} \overset{(i)}{\leq} \sum_{l'=0}^{T-1} \frac{l'^{l'}(\xi_{q-l'}^{(q)})^2}{q^{l'}} \overset{(ii)}{\leq} \sum_{l'=0}^{T-1} \frac{l'^{l'} \left( \frac{c'}{q^{T-l'+1}} \right)^2}{q^{l'}} \lesssim \sum_{l'=0}^{T-1} \frac{l'^{l'}}{q^{2T-2l'+2+l'}} \leq T \frac{T^T}{q^{T+2}} \lesssim \frac{1}{q^{T-1}},$$

where $(i)$ uses the classical bound on the binomial coefficient, and $(ii)$ Equation (20) in Assumption 1.

Combining the bounds on $E_1$ and $E_2$ finally yields

$$\Pr\left( \exists l \in \{T, \ldots, q\} \colon \varpi_l > 1/q \right) \leq \sum_{l=T}^{q} \Pr\left( \varpi_l > 1/q \right) \leq \frac{n^4 q^3}{d}(E_1 + E_2)$$

$$\lesssim \frac{n^4}{dq^{T-4}} \lesssim d^{4\ell - 1 - \beta(T-4)} \overset{(i)}{\lesssim} \frac{1}{d},$$

where $(i)$ follows from the definition of $T = \lceil 4 + \frac{4\ell}{\beta} \rceil$ in Assumption 1. $\qquad\square$

The next statement is a non-asymptotic version of Proposition 3 from Ghorbani et al. (2021).

**Lemma 18** (Graph argument). *Let $\mathbf{W} \in \mathbb{R}^{n \times n}$ be a random matrix that satisfies the following conditions:*

1. $[\mathbf{W}]_{i,i} = 0$, $\forall i \in \{1, \ldots, n\}$.

2. *There exists $\mathcal{M} > 0$ such that, for all $i, j, k \in \{1, \ldots, n\}$ with $i \neq j$ and $j \neq k$, we have*

$$\left| \mathbb{E}_{x_j}\left[ [\mathbf{W}]_{i,j}[\mathbf{W}]_{j,k} \right] \right| \leq \frac{1}{\mathcal{M}} |[\mathbf{W}]_{i,k}|.$$

3. *There exists $l \in \mathbb{N}$ such that, for all $p \in \mathbb{N}$, $p \geq 2$ and all $i, j \in \{1, \ldots, n\}$, $i \neq j$, we have*

$$\mathbb{E}\left[ |[\mathbf{W}]_{i,j}|^p \right]^{1/p} \leq \sqrt{\frac{p^l}{\mathcal{M}}}.$$

*Then, for all $p \in \mathbb{N}$, $p > 2$,*

$$\Pr\left( \|\mathbf{W}\| > 1/2 \right) \leq c_1^p p^{3p} n \left( \frac{n}{\mathcal{M}} \right)^p + c_2^p \left( \frac{n}{\mathcal{M}} \right)^2,$$

*where $c_1$ and $c_2$ are positive constants that depend on $l$.*

*Proof.* Repeating the steps in the proof of Proposition 3 from Ghorbani et al. (2021), we get

$$\mathbb{E}[\|\mathbf{W}\|^{2p}] \leq \mathbb{E}\left[ \mathbf{Tr}(\mathbf{W}^{2p}) \right] \leq (cp)^{3p} \frac{n^{p+1}}{\mathcal{M}^p} + c'^p \left( \frac{n}{\mathcal{M}} \right)^2.$$

Note that the proof in Ghorbani et al. (2021) assumes $\mathcal{M}$ to be in the order of $d^l$. We get rid of this assumption and keep $\mathcal{M}$ explicit. Furthermore, Ghorbani et al. (2021) use their Lemma 4 during their proof, but we use our Lemma 19 instead.

Ultimately, we apply the Markov inequality to get a high-probability bound:

$$\Pr(\|\mathbf{W}\| \geq 1/2) = \Pr\left( \|\mathbf{W}\|^{2p} \geq (1/2)^{2p} \right)$$

$$\leq \frac{\mathbb{E}[\|\mathbf{W}\|^{2p}]}{(1/2)^{2p}} = \left( \frac{c^3}{(1/2)^2} \right)^p p^{3p} \frac{n^{p+1}}{\mathcal{M}^p} + \left( \frac{c'}{(1/2)^2} \right)^p \left( \frac{n}{\mathcal{M}} \right)^2.$$

Renaming the constants concludes the proof. $\qquad\square$

**Lemma 19** (Hypercontractivity). *For all $l, q, d \in \mathbb{N}$ and $p \geq 2$, we have*

$$\mathbb{E}_{x,x'\sim\mathcal{U}(\{-1,1\}^d)}\left[|\mathcal{Q}_l^{(d,q)}(x,x')|^p\right]^{1/p} \leq (p-1)^{l/2}\sqrt{\mathbb{E}_{x,x'\sim\mathcal{U}(\{-1,1\}^d)}\left[(\mathcal{Q}_l^{(d,q)}(x,x'))^2\right]},$$

*where $\mathcal{Q}_l^{(d,q)}(x,x')$ is defined in Equation* (30).

*Proof.* Let $x, x' \sim \mathcal{U}(\{-1,1\}^d)$ and let $z$ be the entry-wise product of $x$ and $x'$. Then, for all $S \subseteq \{1,\ldots,d\}$, $\mathcal{Y}_S(x)\mathcal{Y}_S(x')$ depends only on $z$:

$$\mathcal{Y}_S(x)\mathcal{Y}_S(x') = \left(\prod_{i\in S}[x]_i\right)\left(\prod_{i\in S}[x']_i\right) = \prod_{i\in S}[x]_i[x']_i = \prod_{i\in S}[z]_i.$$

Hence, $\mathcal{Q}_l^{(d,q)}(x,x')$ also only depends on $x$ and $x'$ via $z$. Furthermore, note that $z \sim \mathcal{U}(\{-1,1\}^d)$. Therefore, we can use hypercontractivity (Beckner, 1975) as for instance in Lemma 4 from Misiakiewicz & Mei (2021) to conclude the proof. $\qquad\square$

## D  OPTIMAL REGULARIZATION AND TRAINING ERROR

### D.1  OPTIMAL REGULARIZATION

In the main text we often refer to the optimal regularization $\lambda_{\text{opt}}$, defined as the minimizer of the risk $\mathbf{Risk}(\hat{f}_\lambda)$. While we cannot calculate $\lambda_{\text{opt}}$ directly, we only need the rate $\ell_{\lambda_{\text{opt}}}$ such that $\max\{\lambda_{\text{opt}}, 1\} \in \Theta\left(d^{\ell_{\lambda_{\text{opt}}}}\right)$. Furthermore, it is not a priori clear that such a $\ell_{\lambda_{\text{opt}}}$ minimizes the rate exponent of the risk in Theorem 1. The current subsection establishes that this is indeed the case, and provides a way to determine $\ell_{\lambda_{\text{opt}}}$.

We introduce some shorthand notation for the rate exponents in Theorem 1:

$$\eta_v(\ell_\lambda;\ell,\ell_\sigma,\beta) := \frac{-\ell_\sigma-\ell_\lambda}{\ell} - \frac{\beta}{\ell}\min\{\delta, 1-\delta\},$$

$$\eta_b(\ell_\lambda;\ell,L^*,\beta) := -2 - \frac{2}{\ell}(-\ell_\lambda - 1 - \beta(L^*-1)),$$

$$\eta(\ell_\lambda;\ell,\ell_\sigma,L^*,\beta) := \max\left\{\eta_v(\ell_\lambda;\ell,\ell_\sigma,\beta), \eta_b(\ell_\lambda;\ell,L^*,\beta)\right\}.$$

We highlight that $\eta_b$ and $\eta$ depend on $\ell_\lambda$ also through $\delta = \frac{\ell-\ell_\lambda-1}{\beta} - \left\lfloor\frac{\ell-\ell_\lambda-1}{\beta}\right\rfloor$. Hence, in the setting of Theorem 1, we have with high probability that

$$\mathbf{Variance}(\hat{f}_\lambda) \in \Theta(n^{\eta_v(\ell_\lambda;\ell,\ell_\sigma,\beta)}),$$
$$\mathbf{Bias}^2(\hat{f}_\lambda) \in \Theta(n^{\eta_b(\ell_\lambda;\ell,L^*,\beta)}),$$
$$\mathbf{Risk}^2(\hat{f}_\lambda) \in \Theta(n^{\eta(\ell_\lambda;\ell,\ell_\sigma,L^*,\beta)}).$$

In the following, we view those quantities as functions of $\ell_\lambda$, with all other parameters fixed. Next, we additionally define

$$\lambda_{\text{opt}} := \underset{\lambda\geq 0|\max\{\lambda,1\}\in\mathcal{O}(d^{\bar{\ell}})}{\arg\min} \mathbf{Risk}(\hat{f}_\lambda),$$

$$\ell_{\lambda_{\text{min}}} := \underset{\ell_\lambda\in[0,\bar{\ell}]}{\arg\min}\,\eta(\ell_\lambda;\ell,\ell_\sigma,L^*,\beta),$$

$$\eta_{\text{min}} := \underset{\ell_\lambda\in[0,\bar{\ell}]}{\min}\,\eta(\ell_\lambda;\ell,\ell_\sigma,L^*,\beta),$$

$$\bar{\ell} := \ell - 1 - \beta(L^*-1).$$

First, we remark that $\ell_{\lambda_{\text{min}}}$—the set of regularization rates that minimize the risk rate—might have cardinality larger than one. However, it cannot be empty: $[0,\bar{\ell}]$ is a closed set, and Lemma 21 below shows that $\eta$ is a continuous function.

Second, $\bar{\ell}$ defines the scope of the minimization domain, guaranteeing that the constraint on $L^*$ in Theorem 1 holds for all candidate $\ell_\lambda$.

*Rate of optimal regularization $\ell_{\lambda_{opt}}$ vs. optimal rate $\ell_{\lambda_{min}}$:* Let $\ell_{\lambda_{opt}}$ be the rate of the optimal regularization strength such that $\max\{\lambda_{opt}, 1\} \in \Theta(d^{\ell_{\lambda_{opt}}})$. It is a priori not clear that $\ell_{\lambda_{opt}}$ minimizes $\eta$. However, Lemma 20 bridges the two quantities, and guarantees with high probability that the rate of $\lambda_{opt}$ minimizes the rate of the risk.

**Lemma 20** (Optimal regularization and optimal rate). *In the setting of Theorem 1, assume $\ell > 0$, $\beta \in (0,1)$, $\ell_\sigma \geq -\bar{\ell}$, $L^* \in \left[1, \lceil \frac{\ell-1}{\beta} \rceil\right] \cap \mathbb{N}$. Then, for $d$ sufficiently large, with probability at least $1 - cd^{-\beta \min\{\bar{\delta}, 1-\bar{\delta}\}}$ there exists $l \in \ell_{\lambda_{min}}$ such that*

$$\max\{\lambda_{opt}, 1\} \in \Theta\left(d^l\right).$$

Hence, we only need to obtain a minimum rate $l \in \ell_{\lambda_{min}}$ instead of $\ell_{\lambda_{opt}}$. In order to propose a method for this, we first establish properties of $\eta, \eta_b, \eta_v$ in the following lemma.

**Lemma 21** (Properties of $\eta$). *Assume $\ell > 0$, $\beta \in (0,1)$, $\ell_\sigma \geq -\bar{\ell}$, $L^* \in \left[1, \lceil \frac{\ell-1}{\beta} \rceil\right] \cap \mathbb{N}$.*

1. *Over $[0, \bar{\ell}]$, $\eta_v(\cdot; \ell, \ell_\sigma, \beta)$ is continuous and non-increasing, and $\eta_b(\cdot; \ell, L^*, \beta)$ is continuous and strictly increasing.*

2. *$\ell_{\lambda_{min}} := \arg\min_{\ell_\lambda \in [0, \bar{\ell}]} \eta(\ell_\lambda; \ell, \ell_\sigma, L^*, \beta)$ is a closed interval.*

3. *If there exists $\tilde{l} \in [0, \bar{\ell}]$ with $\eta_v(\tilde{l}; \ell, \ell_\sigma, \beta) = \eta_b(\tilde{l}; \ell, L^*, \beta)$, then $\eta_{\min} = \eta(\tilde{l}; \ell, \ell_\sigma, L^*, \beta)$. Otherwise, $\eta_{\min} = \eta(0; \ell, \ell_\sigma, L^*, \beta)$.*

4. *Every $\bar{l} \in [0, \bar{\ell}]$ with $\bar{l} \geq l$ for all $l \in \ell_{\lambda_{min}}$ satisfies*

$$\eta(\bar{l}; \ell, \ell_\sigma, L^*, \beta) = \eta_b(\bar{l}; \ell, L^*, \beta).$$

5. *Let $\underline{l} \in [0, \bar{\ell}]$ with $\underline{l} \leq l$ for all $l \in \ell_{\lambda_{min}}$. If $\eta(\underline{l}; \ell, \ell_\sigma, L^*, \beta) - \eta_{\min} \leq c$ where $c > 0$ is constant [5] and depends only on $\ell, \ell_\sigma, L^*, \beta$, then*

$$\eta(\underline{l}; \ell, \ell_\sigma, L^*, \beta) = \eta_{\min} + \frac{2}{\ell}\left(\min\left(\ell_{\lambda_{min}}\right) - \underline{l}\right).$$

*Finding an optimal rate:* Lemma 21 suggests a simple strategy to find a $l \in \ell_{\lambda_{min}}$ numerically: search the intersection of $\eta_v$ and $\eta_b$ in $[0, \bar{\ell}]$; if found, then the intersection point is optimal, otherwise $l = 0$ is optimal. Note that, if the intersection point exists, it is unique and easy to numerically approximate, since $\eta_v$ is non-increasing, and $\eta_b$ is strictly increasing.

*Calculating numerical solutions:* However, Lemma 21 also shows that $\ell_{\lambda_{min}}$ is an interval and thus might contain multiple values. In that case, the proposed strategy might not necessarily retrieve the rate of $\lambda_{opt}$, but a different $l \in \ell_{\lambda_{min}}$. Yet, Theorem 1 guarantees that both the optimally regularized estimator and any estimator regularized with $\max\{\lambda, 1\} \in d^l$ for any $l \in \ell_{\lambda_{min}}$ have a risk vanishing with the same rate $n^{\eta_{\min}}$ with high probability. In particular, this allows us to exhibit the rate of the optimally regularized estimator in Figure 1a. Finally, because of the multiple descent phenomenon (see, for example, Figure 1), we do not expect either $\ell_{\lambda_{opt}}$ or $\beta^*$ to attain an easily readable closed-form expression. Nevertheless, simple optimization procedures allow us to calculate accurate numerical approximations.

Finally, we prove Lemmas 20 and 21.

*Proof of Lemma 20.* Let $\ell_{\lambda_{opt}}$ be such that

$$\max\{\lambda_{opt}, 1\} \in \Theta\left(d^{\ell_{\lambda_{opt}}}\right).$$

---

[5]We omit an explicit definition of $c$ here for brevity and refer to the proof instead.

Combining the bounds on bias and variance from Theorem 1, we have

$$
\begin{aligned}
\mathbf{Risk}^2(\hat{f}_\lambda) &= \mathbf{Bias}^2(\hat{f}_\lambda) + \mathbf{Variance}^2(\hat{f}_\lambda) \\
&\in \Theta\left(n^{-2-\frac{2}{\ell}(-\ell_\lambda-1-\beta(L^*-1))} + n^{\frac{-\ell_\sigma-\ell_\lambda}{\ell}-\frac{\beta}{\ell}\min\{\delta,1-\delta\}}\right) \\
&\in \Theta\left(n^{\eta(\ell_\lambda;\ell,\ell_\sigma,L^*,\beta)}\right) \quad \text{with probability} \ge 1 - cd^{-\min\{\bar{\delta},1-\bar{\delta}\}}
\end{aligned}
\tag{38}
$$

uniformly for all $\ell_\lambda \in \left[0,\bar{\ell}\right]$ with $\max\{\lambda,1\} \in \Theta(d^{\ell_\lambda})$.

Throughout the remainder of this proof, we drop the dependencies on $\ell, \ell_\sigma, L^*, \beta$ in the notation of $\eta$ and simply write $\eta(\ell_\lambda)$. We further omit repeating that each step is true with probability at least $1 - cd^{-\min\{\bar{\delta},1-\bar{\delta}\}}$, but imply it throughout.

The goal of the proof is to show that there exists $l \in \ell_{\lambda_{\min}}$ sufficiently close to $\ell_{\lambda_{\mathrm{opt}}}$; formally, we need to show that

$$
d^{\ell_{\lambda_{\mathrm{opt}}}-l} \in \Theta(1). \tag{39}
$$

If this is the case, then the definition of $\ell_{\lambda_{\mathrm{opt}}}$ yields the conclusion as follows:

$$
\max\{\lambda_{\mathrm{opt}},1\} \in \Theta\left(d^{\ell_{\lambda_{\mathrm{opt}}}}\right) = \Theta\left(d^l d^{\ell_{\lambda_{\mathrm{opt}}}-l}\right) \stackrel{(i)}{=} \Theta\left(d^l\right),
$$

where $(i)$ uses Equation (39).

Towards an auxiliary result, we first apply Equation (38) to $\ell_{\lambda_{\mathrm{opt}}}$ and $l \in \ell_{\lambda_{\min}}$ with $\max\{\lambda,1\} \in \Theta(d^l)$:

$$
\begin{aligned}
\mathbf{Risk}^2(\hat{f}_\lambda) &\le c_2 n^{\eta(l)} = c_2 n^{\eta_{\min}}, \\
\mathbf{Risk}^2(\hat{f}_{\lambda_{\mathrm{opt}}}) &\ge c_1 n^{\eta(\lambda_{\mathrm{opt}})},
\end{aligned}
$$

where $c_1, c_2 > 0$ are the constants hidden by the $\Theta$-notation in Equation (38). Next, the optimality of $\lambda_{\mathrm{opt}}$ yields $c_1 < c_2$, and

$$
c_1 n^{\eta(\ell_{\lambda_{\mathrm{opt}}})} \le \mathbf{Risk}^2(\hat{f}_{\lambda_{\mathrm{opt}}}) \le \mathbf{Risk}^2(\hat{f}_l) \le c_2 n^{\eta_{\min}}.
$$

This implies

$$
n^{\eta(\ell_{\lambda_{\mathrm{opt}}})-\eta_{\min}} \le \frac{c_2}{c_1} \quad \Rightarrow \quad \eta(\ell_{\lambda_{\mathrm{opt}}}) - \eta_{\min} \le \frac{\log(c_2/c_1)}{\log n}, \tag{40}
$$

where the second implication uses that $\eta(\ell_{\lambda_{\mathrm{opt}}}) - \eta_{\min} \ge 0$, since $\eta_{\min}$ is the minimum rate.

With this result, we finally focus on establishing Equation (39) which yields the claim of this lemma. Lemma 21 shows that $\ell_{\lambda_{\min}}$ is an interval; hence, we distinguish three cases:

*Case $\ell_{\lambda_{opt}} \in \ell_{\lambda_{min}}$:* Picking $\ell_{\lambda_{\mathrm{opt}}} = l \in \ell_{\lambda_{\min}}$ directly yields $d^{\ell_{\lambda_{\mathrm{opt}}}-\ell_{\lambda_{\mathrm{opt}}}} = d^0 \in \Theta(1)$.

*Case $\ell_{\lambda_{opt}} < \min(\ell_{\lambda_{min}})$:* Let $\underline{l} := \min(\ell_{\lambda_{\min}})$. Equation (40) yields $\eta(\ell_{\lambda_{\mathrm{opt}}}) - \eta_{\min} \le c$ for any $c > 0$ if $d$ is large enough. Hence, for $d$ fixed but sufficiently large, Lemma 21 yields

$$
\eta(\ell_{\lambda_{\mathrm{opt}}}) = \eta_{\min} + \frac{2}{\ell}\left(\underline{l} - \ell_{\lambda_{\mathrm{opt}}}\right).
$$

Applying Equation (40), we get

$$
\underline{l} - \ell_{\lambda_{\mathrm{opt}}} = \frac{\ell}{2}(\eta(\ell_{\lambda_{\mathrm{opt}}}) - \eta_{\min}) \le \frac{\ell\log(c_2/c_1)}{2\log n}.
$$

Now, since $\ell_{\lambda_{\mathrm{opt}}} \le \underline{l}$, $d^{\ell_{\lambda_{\mathrm{opt}}}-\underline{l}} \in \mathcal{O}(1)$. Furthermore,

$$
d^{\underline{l}-\ell_{\lambda_{\mathrm{opt}}}} \le d^{\frac{\ell\log(c_2/c_1)}{2\log n}} \stackrel{(i)}{=} d^{\frac{\log(c_2/c_1)}{2\log d}} = \sqrt{c_2/c_1} \in \mathcal{O}(1),
$$

where $(i)$ follows from $n \in \Theta(d^\ell)$. Since both $d^{\underline{l}-\ell_{\lambda_{\mathrm{opt}}}}$ and $\frac{1}{d^{\underline{l}-\ell_{\lambda_{\mathrm{opt}}}}} = d^{\ell_{\lambda_{\mathrm{opt}}}-\underline{l}}$ are in $\mathcal{O}(1)$,

$$
d^{\ell_{\lambda_{\mathrm{opt}}}-\underline{l}} \in \Theta(1),
$$

which establishes Equation (39) for $\bar{l} \in \ell_{\lambda_{\min}}$ and thereby concludes this case.

*Case $\ell_{\lambda_{opt}} > \max(\ell_{\lambda_{min}})$:* Let $\bar{l} := \max(\ell_{\lambda_{\min}})$. Then, Lemma 21 yields

$$
\begin{aligned}
\eta(\ell_{\lambda_{\mathrm{opt}}}) &= -2 - \frac{2}{\ell}(-\ell_{\lambda_{\mathrm{opt}}} - 1 - \beta(L^* - 1)) \\
&= -2 - \frac{2}{\ell}(-\ell_{\lambda_{\mathrm{opt}}} - \bar{l} + \bar{l} - 1 - \beta(L^* - 1)) \\
&= -2 - \frac{2}{\ell}(-\bar{l} - 1 - \beta(L^* - 1)) - \frac{2}{\ell}(\bar{l} - \ell_{\lambda_{\mathrm{opt}}}) \\
&\overset{(i)}{=} \eta_{\min} + \frac{2}{\ell}(\ell_{\lambda_{\mathrm{opt}}} - \bar{l}),
\end{aligned}
$$

where $(i)$ uses that $\bar{l} \in \ell_{\lambda_{\min}}$. Applying Equation (40), we get

$$
\ell_{\lambda_{\mathrm{opt}}} - \bar{l} = \frac{\ell}{2}(\eta(\ell_{\lambda_{\mathrm{opt}}}) - \eta_{\min}) \leq \frac{\ell \log(c_2/c_1)}{2 \log n}.
$$

Analogously to the previous case, this implies $d^{\ell_{\lambda_{\mathrm{opt}}} - \bar{l}} \in \mathcal{O}(1)$, and the fact that $\ell_{\lambda_{\mathrm{opt}}} > \bar{l}$ implies $d^{\ell_{\lambda_{\mathrm{opt}}} - \bar{l}} \in \Omega(1)$. Together, this yields

$$
d^{\ell_{\lambda_{\mathrm{opt}}} - \bar{l}} \in \Theta(1),
$$

which establishes Equation (39) for $\bar{l} \in \ell_{\lambda_{\min}}$ and thereby concludes this case. $\qquad\square$

*Proof of Lemma 21.* Throughout this proof, we drop the dependencies on $\ell, \ell_\sigma, L^*, \beta$ in the notation of $\eta, \eta_v, \eta_b$ and simply write $\eta(l), \eta_v(l), \eta_b(l)$.

**Continuity and monotonicity (Item 1)** We first show that $\eta_v(l)$ and $\eta_b(l)$ are continuous functions. $\eta_b(l)$ is an affine function of $l$, hence continuous. $\eta_v(l)$, however, additionally depends on $l$ via $\delta$, which is not linear. Hence, to show that $\eta_v(l)$ is continuous, we need to show that $\min\{\delta, 1 - \delta\}$ is continuous. Consider the triangle wave function

$$
\varpi(t) := \min\{t - \lfloor t \rfloor, 1 - (t - \lfloor t \rfloor)\},
$$

which is well-known to be continuous. Because $\min\{\delta, 1 - \delta\} = \varpi\left(\frac{\ell - l - 1}{\beta}\right)$, and $\frac{\ell - l - 1}{\beta}$ is a linear function of $l$, we get that $\eta_v(l)$ is also continuous.

For monotonicity, we consider the derivatives of $\eta_v(l)$ and $\eta_b(l)$. For $\eta_b(l)$, we have

$$
\partial_l \eta_b(l) = \frac{2}{\ell}.
$$

Since $\eta_b$ is an affine function, and $\frac{2}{\ell} > 0$, this also implies that $\eta_b$ is strictly increasing. For $\eta_v$, we need to distinguish two cases:

$$
\begin{aligned}
\partial_l \eta_v(l) &= \begin{cases} \partial_l \frac{-\ell_\sigma - l - \beta\delta}{\ell} & \delta < 1 - \delta \\ \partial_l \frac{-\ell_\sigma - l - \beta(1-\delta)}{\ell} & \delta \geq 1 - \delta \end{cases} \\
&= \begin{cases} 0 & \delta < 1 - \delta, \\ -\frac{2}{\ell} & \delta \geq 1 - \delta. \end{cases}
\end{aligned}
$$

Since $\eta_v$ is a continuous function with non-positive derivatives, this implies that $\eta_v$ is non-increasing.

**$\eta$ decomposition (Item 3)** First assume that there exists $\tilde{l} \in [0, \ell]$ with $\eta_v(\tilde{l}) = \eta_b(\tilde{l})$. Since $\eta_v$ is non-increasing and $\eta_b$ is strictly increasing,

$$
\eta(l) := \max\{\eta_b(l), \eta_v(l)\} = \begin{cases} \eta_v(l) & l < \tilde{l}, \\ \eta_b(l) & \text{otherwise.} \end{cases}
$$

In particular, $\eta(l) > \eta(\tilde{l})$ for all $l > \tilde{l}$ as $\eta_b$ is strictly increasing, and $\eta(l) \geq \eta(\tilde{l})$ for all $l < \tilde{l}$ since $\eta_v$ is non-increasing. Combined, this yields $\eta_{\min} = \eta(\tilde{l}; \ell, \ell_\sigma, L^*, \beta)$.

Next, assume $\tilde{l}$ does not exist, that is, $\eta_v(l) \neq \eta_b(l)$ for all $l \in [0, \bar{\ell}]$. Then, due to continuity, either $\eta_b(l) > \eta_v(l)$ for all $l \in [0, \bar{\ell}]$, or $\eta_v(l) > \eta_b(l)$ for all $l \in [0, \bar{\ell}]$. However, a closer analysis shows that the latter is not possible: the rates at the boundary $\bar{\ell}$ are

$$\eta_b(\bar{\ell}) = -2 - \frac{2}{\ell}\left(-(\ell - 1 - \beta(L^* - 1)) - 1 - \beta(L^* - 1)\right) = 0,$$

$$\eta_v(\bar{\ell}) = -\frac{\ell_\sigma + (\ell - 1 - \beta(L^* - 1))}{\ell} - \frac{\beta}{\ell}\min\{\delta, 1 - \delta\} \overset{(i)}{\leq} 0,$$

where $(i)$ follows from the assumption $\ell_\sigma \geq -\bar{\ell}$ and the fact that $\min\{\delta, 1 - \delta\} \geq 0$. Hence, if $\tilde{l}$ does not exist, $\eta_v(l) < \eta_b(l), \forall l \in [0, \bar{\ell}]$. In particular, $\eta_{\min}$ is the minimum of the strictly increasing function $\eta_b(l)$ over $[[0, \bar{\ell}]$, and therefore attained only at $0$.

Lastly, combining both cases of $\tilde{l}$ yields the following convenient expression:

$$\eta(l) = \begin{cases} \eta_v(l) & \tilde{l} \text{ exists and } l < \tilde{l}, \\ \eta_b(l) & \text{otherwise.} \end{cases} \tag{41}$$

**Closed interval of solutions (Item 2)**    We again differentiate whether $\tilde{l}$ as in the previous step exists or not. If $\tilde{l}$ does not exist, then the previous step already yields $\ell_{\lambda_{\min}} = \{0\}$, which is a closed interval. Next, assume $\tilde{l}$ exists. Then, all $l \in \ell_{\lambda_{\min}}$ satisfy $l \leq \tilde{l}$, since $\tilde{l} \in \ell_{\lambda_{\min}}$ and $\eta_b$ is strictly increasing. Since further $\eta_v$ is continuous and non-increasing over $[0, \tilde{l}] \supseteq \ell_{\lambda_{\min}}$, $\ell_{\lambda_{\min}}$ is an interval. Finally, $\eta(l) = \max\{\eta_b(l), \eta_v(l)\}$ is the maximum of two continuous functions, hence itself also continuous. Therefore, the minimizers $\ell_{\lambda_{\min}}$ of $\eta$ are a closed set. This concludes that $\ell_{\lambda_{\min}}$ is a closed interval.

**Proof of Items 4 and 5**    Item 4 follows straightforwardly from Equation (41): If $\tilde{l}$ does not exist, then $\eta(l) = \eta_b(l)$ for all $l \in [0, \bar{\ell}]$. Similarly, if $\tilde{l}$ exists, $\eta(l) = \eta_b(l)$ for all $l \geq \tilde{l}$, in particular for $\bar{l} \geq \tilde{l} \in \ell_{\lambda_{\min}}$.

Item 5 requires additional considerations. In the case where $\tilde{l}$ does not exist, we have $\ell_{\lambda_{\min}} = \{0\}$, and the result follows directly. Otherwise, using Equation (41), we have $\eta(l) = \eta_v(l)$ for any $l \leq \underline{l}$, as $\underline{l} \leq \min(\ell_{\lambda_{\min}}) \leq \tilde{l}$. As shown in the proof of Item 1, $\eta_v(l)$ alternates between derivatives $0$ and $-2/\ell$. We claim that there exists a left neighborhood of $\min(\ell_{\lambda_{\min}})$ where the derivative is $-2/\ell$. Assume towards a contradiction that no such left neighborhood exists. Then, there must be a left neighborhood of $\min(\ell_{\lambda_{\min}})$ where the derivative is $0$, since $\eta_v(l)$ alternates between only two derivatives. However, in that left neighborhood, $\eta_v(l)$ is constant with all values equal to $\eta_{\min}$. Hence, there exists $l < \min(\ell_{\lambda_{\min}})$ with $\eta(l) = \eta_v(l) = \eta_{\min}$ and thus $l \in \ell_{\lambda_{\min}}$, which is a contradiction.

Thus, there exists a left neighborhood of $\min(\ell_{\lambda_{\min}})$ with diameter $\varepsilon > 0$ throughout which the derivative is $-\frac{2}{\ell}$. Then, for all $l \in [\min(\ell_{\lambda_{\min}}) - \varepsilon, \min(\ell_{\lambda_{\min}})]$,

$$\eta_v(l) = \eta_{\min} - \frac{2}{\ell}(l - \min(\ell_{\lambda_{\min}})).$$

Finally, since $\eta_v(l)$ is non-increasing, as long as $\eta_v(\underline{l}) \leq \eta_{\min} + \frac{2}{\ell}\varepsilon$ we have $\underline{l} \geq \min(\ell_{\lambda_{\min}}) - \varepsilon$. Hence, choosing $c = \frac{2}{\ell}\varepsilon$ yields the statement of Item 5. $\qquad\square$

## D.2   PROOF OF THEOREM 2

The informal Theorem 2 in the main text relies on a $\beta^*$, defined as the intersection of the variance and bias rates from Theorem 1 for the interpolator $\hat{f}_0$ (setting $\ell_\lambda = 0$). Whenever $\beta^*$ is unique, the fact that $\mathbf{Bias}^2(\hat{f}_0)$ in Theorem 1 strictly increases as a function of $\beta$ induces a phase transition: for $\beta > \beta^*$, the bias dominates the rate of the risk in Theorem 1, while for $\beta \leq \beta^*$, the variance dominates. In particular, Lemmas 20 and 21 imply that interpolation is harmless whenever the bias dominates, and harmful if the variance dominates.

Intuitively, Theorem 2 considers optimally regularized estimators, and varies the inductive bias strength via $\beta$. The formal Theorem 4 below presents a different perspective: it considers $\beta$ fixed,

and instead differentiates whether the optimal risk rate in Theorem 1 results from $\ell_\lambda > 0$ (harmful interpolation) or $\ell_\lambda = 0$ (harmless interpolation).

Whenever $\beta^*$ is well-defined and unique, as for example in the setting of Figure 1, the two perspectives coincide. However, one can construct pathological edge cases where variance and bias rates intersect on an interval of $\beta$ values, or the dominating quantity of the risk rate in Theorem 1 as a function of $\beta$ alternates between the variance and bias. While Theorem 2 fails to capture such settings, Theorem 4 still applies. We hence present Theorem 4 as a more general result.

**Theorem 4** (Formal version of Theorem 2). *In the setting of Theorem 1 using the notation from Appendix D.1, let $\ell > 0$, $\beta \in (0,1)$, $\ell_\sigma \geq -\bar{\ell}$, and $L^* \in \left[1, \lceil \frac{\ell-1}{\beta} \rceil \right] \cap \mathbb{N}$. Let further $\lambda_{opt} = \arg\min_{\lambda \geq 0 \mid \max\{\lambda,1\} \in \mathcal{O}(d^{\bar{\ell}})} \mathbf{Risk}(\hat{f}_\lambda)$. Then, the expected training error behaves as follows:*

1. *If all $l \in \arg\min_{l \in [0,\bar{\ell}]} \eta(l; \ell, \ell_\sigma, L^*, \beta)$ satisfy $\eta(l; \ell, \ell_\sigma, L^*, \beta) < \eta(0; \ell, \ell_\sigma, L^*, \beta)$, then*

$$\left| \mathbb{E}_\epsilon \left[ \frac{1}{n} \sum_i (\hat{f}_{\lambda_{opt}}(x_i) - y_i)^2 \right] - \sigma^2 \right| \in \mathcal{O}(d^{-l} + \mathbf{Risk}^2(\hat{f}_{\lambda_{opt}})), \quad w.p. \; \geq 1 - cd^{-\beta \min\{\bar{\delta}, 1-\bar{\delta}\}}.$$

2. *If all $l \in (0, \bar{\ell}]$ satisfy $\eta(l; \ell, \ell_\sigma, L^*, \beta) > \eta(0; \ell, \ell_\sigma, L^*, \beta)$, then*

$$\mathbb{E}_\epsilon \left[ \frac{1}{n} \sum_i (\hat{f}_{\lambda_{opt}}(x_i) - y_i)^2 \right] \leq \tilde{c}\sigma^2 + \mathcal{O}\left( \mathbf{Risk}^2(\hat{f}_{\lambda_{opt}}) \right), \quad w.p. \; \geq 1 - cd^{-\beta \min\{\bar{\delta}, 1-\bar{\delta}\}}$$

*for some constant $\tilde{c} < 1$.*

Intuitively, the two cases in Theorem 4 correspond to harmful and harmless interpolation, where the optimal rate in Theorem 1 is for $\ell_\lambda > 0$ and $\ell_\lambda = 0$, respectively. Then, Lemma 20 yields with high probability that also $\ell_{\lambda_{opt}} > 0$ and $\ell_{\lambda_{opt}} = 0$ in the first and second case, respectively. Finally, we remark that Theorem 4 lacks an edge case: if both some $\ell_\lambda > 0$ and $\ell_\lambda = 0$ minimize the risk rate in Theorem 1 simultaneously, Lemma 20 fails to differentiate whether $\ell_{\lambda_{opt}}$ is zero or positive. However, that edge case corresponds to either interpolation or very weak regularization. Hence, we conjecture the corresponding model's training error to behave similar to the second case in Theorem 4.

*Proof of Theorem 4.* First, Lemma 20 yields with probability at least $1 - cd^{-\beta \min\{\bar{\delta}, 1-\bar{\delta}\}}$ that $\max\{\lambda_{opt}, 1\} \in \Theta(d^l)$, where $l$ is a minimizer of $\eta(l; \ell, \ell_\sigma, L^*, \beta)$. The condition in Item 1 guarantees that all optimal $l$ are positive, while the condition in Item 2 ensures that the optimal $l = 0$.

**Harmful interpolation setting (Item 1)** In this case, $l > 0$, and thus $\lambda_{opt} \in \Theta(d^l)$ for $d$ sufficiently large. We start by applying Theorem 1 with $\ell_\lambda = l$ in this setting.

Within the proof of Theorem 1 in Section 5.2, we pick a $m \in \mathbb{N}$ such that for $d$ sufficiently large, with probability at least $1 - cd^{-\beta \min\{\bar{\delta}, 1-\bar{\delta}\}}$, $m$ satisfies the conditions of Lemmas 1 to 4 and Theorem 3. For the remainder of this proof, let $m$ be the same as in the proof of Theorem 1 for $\ell_\lambda = l$. This $m$ satisfies the conditions of Lemma 23, which hence yields

$$\frac{\lambda_{opt}^2 \sigma^2}{n} \mathbf{Tr}\left(\mathbf{H}^{-2}\right) \leq \mathbb{E}_\epsilon \left[ \frac{1}{n} \sum_i (\hat{f}_{\lambda_{opt}}(x_i) - y_i)^2 \right] \leq \frac{\lambda_{opt}^2 \sigma^2}{n} \mathbf{Tr}\left(\mathbf{H}^{-2}\right) + 6\lambda_{opt}^2 \frac{r_2^2}{r_1^2} \frac{\|\mathbf{D}_{\leq m}^{-1} a\|^2}{n^2},$$

where $\mathbf{H} := \mathbf{K} + \lambda \mathbf{I}_n$. Then, using that $a \leq b \leq a + d$ implies $|b - c| \leq |a - c| + d$, we further get

$$\left| \mathbb{E}_\epsilon \left[ \frac{1}{n} \sum_i (\hat{f}_{\lambda_{opt}}(x_i) - y_i)^2 \right] - \sigma^2 \right| \leq \sigma^2 \underbrace{\left| \frac{\lambda_{opt}^2}{n} \mathbf{Tr}\left(\mathbf{H}^{-2}\right) - 1 \right|}_{:=T_1} + \underbrace{6\lambda_{opt}^2 \frac{r_2^2}{r_1^2} \frac{\|\mathbf{D}_{\leq m}^{-1} a\|^2}{n^2}}_{:=T_2}. \quad (42)$$

We first bound $T_2$ as follows:

$$
\begin{aligned}
T_2 = 6\lambda_{\text{opt}}^2 \frac{r_2^2}{r_1^2} \frac{\|\mathbf{D}_{\leq m}^{-1} a\|^2}{n^2} &\overset{(i)}{\lesssim} \frac{d^{2l}}{d^{2\ell}} \frac{r_2^2}{r_1^2} \|\mathbf{D}_{\leq m}^{-1} a\|^2 \\
&\overset{(ii)}{\lesssim} \frac{d^{2l}}{d^{2\ell}} \cdot d^2 q^{2(L^*-1)} \\
&= \frac{d^{2l}}{d^2 d^{2l} d^{2(\ell-l-1)}} \cdot d^2 q^{2(L^*-1)} \\
&\lesssim d^{-2(\ell-l-1-\beta(L^*-1))} \\
&\overset{(iii)}{\in} \mathcal{O}\left(\mathbf{Bias}^2(\hat{f}_{\lambda_{\text{opt}}})\right) \subseteq \mathcal{O}\left(\mathbf{Risk}^2(\hat{f}_{\lambda_{\text{opt}}})\right),
\end{aligned}
\tag{43}
$$

where $(i)$ uses $n \in \Theta(d^\ell)$ and $\lambda_{\text{opt}} \in \Theta(d^l)$, $(ii)$ applies Lemma 2 for the rate of $\|\mathbf{D}_{\leq m}^{-1} a\|$ and Lemma 3 for $r_2^2, r_1^2 \in \Theta(1)$, and $(iii)$ matches the expression to the rate of the bias in Theorem 1.

For $T_1$, we will bound $\mathbf{Tr}\left(\mathbf{H}^{-2}\right)$ from above and below using Lemma 22. We hence introduce the following notation:

$$
\mathbf{K}_{-2} := \mathbf{K}_{\leq m} + \mathbf{K}_1, \quad \mathbf{H}_2 := \mathbf{K}_2 + \lambda_{\text{opt}} \mathbf{I}_n,
$$

so that $\mathbf{H} = \mathbf{H}_2 + \mathbf{K}_{-2}$, and where $\mathbf{K}_1$ and $\mathbf{K}_2$ are defined in Appendix C.2. Furthermore, the rank of $\mathbf{K}_{-2}$ is at most $|\mathbb{N} \setminus \mathcal{I}_2| = |\{k \in \mathbb{N} \mid |S_k| < \bar{L} + 2\}|$, that is, the number of eigenfunctions that contribute to $\mathbf{K}_{-2}$. The rate of $|\mathbb{N} \setminus \mathcal{I}_2|$ is

$$
|\mathbb{N} \setminus \mathcal{I}_2| \overset{(i)}{=} \sum_{l=0}^{\bar{L}+1} \mathcal{C}(l, q, d) \overset{(ii)}{=} 1 + \sum_{l=1}^{\bar{L}+1} d \binom{q-1}{l-1} \overset{(iii)}{\in} \mathcal{O}(d \cdot q^{\bar{L}}),
$$

where $(i)$ uses the definition of $\mathcal{C}(l, q, d)$ in Equation (17), $(ii)$ applies Lemma 9 with $d$ sufficiently large, and $(iii)$ uses the classical bound $\binom{q-1}{l-1} \leq \left(e\frac{q-1}{l-1}\right)^{l-1}$ as well as the fact that the largest monomial dominates the sum. Finally, since $n \in \Theta(d \cdot q^{\bar{L}+\bar{\delta}})$, we have

$$
\text{rank}(\mathbf{K}_{-2}) \leq |\mathbb{N} \setminus \mathcal{I}_2| \in \mathcal{O}(nq^{-\bar{\delta}}) \subseteq o(n).
\tag{44}
$$

Therefore, for $d$ and hence $n \in \Theta(d^\ell)$ sufficiently large, $\text{rank}(\mathbf{K}_{-2}) < n$, and Lemma 13 yields $c_1 \leq \mu_{\min}(\mathbf{H}_2) \leq \|\mathbf{H}_2\| \leq c_2$ for some constants $c_1, c_2 > 0$ with probability at least $1 - c'q^{-(1-\bar{\delta})}$. We can thus instantiate Lemma 22 with $\mathbf{M} = \mathbf{H}, \mathbf{M}_1 = \mathbf{K}_{-2}, \mathbf{M}_2 = \mathbf{H}_2$. This implies that:

$$
\mathbf{Tr}\left(\mathbf{H}^{-2}\right) \geq \frac{n - \text{rank}(\mathbf{K}_{-2})}{\|\mathbf{H}_2\|^2} \geq \frac{n - \text{rank}(\mathbf{K}_{-2})}{(c_2 + \lambda_{\text{opt}})^2}.
$$

The upper bound on $\mathbf{Tr}\left(\mathbf{H}^{-2}\right)$ simply follows from

$$
\begin{aligned}
\mathbf{Tr}\left(\mathbf{H}^{-2}\right) &\leq n\|\mathbf{H}^{-2}\| \\
&= \frac{n}{\mu_{\min}\left(\mathbf{K}_{\leq m} + \mathbf{K}_1 + \mathbf{K}_2 + \lambda_{\text{opt}} \mathbf{I}_n\right)^2} \\
&\leq \frac{n}{\mu_{\min}\left(\mathbf{K}_2 + \lambda_{\text{opt}} \mathbf{I}_n\right)^2} \\
&\overset{(i)}{\leq} \frac{n}{(c_1 + \lambda_{\text{opt}})^2},
\end{aligned}
\tag{45}
$$

where $(i)$ uses the previous lower bound on $\mu_{\min}(\mathbf{K}_2)$ from Lemma 13.

Combining the upper and lower bounds on $\mathbf{Tr}\left(\mathbf{H}^{-2}\right)$ yields

$$\frac{n-\operatorname{rank}(\mathbf{K}_{-2})}{(c_2+\lambda_{\mathrm{opt}})^2} \leq \mathbf{Tr}\left(\mathbf{H}^{-2}\right) \leq \frac{n}{(c_1+\lambda_{\mathrm{opt}})^2}$$

$$\frac{n-\operatorname{rank}(\mathbf{K}_{-2})}{n}\frac{\lambda_{\mathrm{opt}}^2}{(c_2+\lambda_{\mathrm{opt}})^2} \leq \frac{\lambda_{\mathrm{opt}}^2}{n}\mathbf{Tr}\left(\mathbf{H}^{-2}\right) \leq \frac{\lambda_{\mathrm{opt}}^2}{(c_1+\lambda_{\mathrm{opt}})^2}$$

$$-\frac{\operatorname{rank}(\mathbf{K}_{-2})}{n}\frac{\lambda_{\mathrm{opt}}^2}{(c_2+\lambda_{\mathrm{opt}})^2}+\frac{\lambda_{\mathrm{opt}}^2}{(c_2+\lambda_{\mathrm{opt}})^2}-1 \leq \frac{\lambda_{\mathrm{opt}}^2}{n}\mathbf{Tr}\left(\mathbf{H}^{-2}\right)-1 \leq \frac{\lambda_{\mathrm{opt}}^2}{(c_1+\lambda_{\mathrm{opt}})^2}-1$$

$$-\frac{\operatorname{rank}(\mathbf{K}_{-2})}{n}\frac{\lambda_{\mathrm{opt}}^2}{(c_2+\lambda_{\mathrm{opt}})^2}-\frac{2c_2\lambda_{\mathrm{opt}}+c_2^2}{(c_2+\lambda_{\mathrm{opt}})^2} \leq \frac{\lambda_{\mathrm{opt}}^2}{n}\mathbf{Tr}\left(\mathbf{H}^{-2}\right)-1 \leq -\frac{2c_1\lambda_{\mathrm{opt}}+c_1^2}{(c_1+\lambda_{\mathrm{opt}})^2}.$$

Taking absolute values, a simple case distinction and $c_1 < c_2$ yields the following bound:

$$\left|\frac{\lambda_{\mathrm{opt}}^2}{n}\mathbf{Tr}\left(\mathbf{H}^{-2}\right)-1\right| \leq \frac{2c_2+\lambda_{\mathrm{opt}}}{(c_2+\lambda_{\mathrm{opt}})^2}+\frac{\operatorname{rank}(\mathbf{K}_{-2})}{n}\frac{\lambda_{\mathrm{opt}}^2}{(c_2+\lambda_{\mathrm{opt}})^2}$$

$$\overset{(i)}{\lesssim} \frac{d^l}{d^{2l}}+\frac{nq^{-\bar{\delta}}}{n}\frac{d^{2l}}{d^{2l}} = d^{-l}+q^{-\bar{\delta}} \in \mathcal{O}(d^{-l}),$$

where $(i)$ uses the rate of $\operatorname{rank}(\mathbf{K}_{-2})$ from Equation (44) and $\lambda_{\mathrm{opt}} \in \Theta\left(d^l\right)$. Combining this bound on $T_1$ with the bound on $T_2$ in Equation (43) and collecting all error probabilities concludes the current case.

**Harmless interpolation setting (Item 2)** In this setting, the only minimizer of the risk rate is $l = 0$, and hence

$$\lambda_{\mathrm{opt}} \leq \max\{\lambda_{\mathrm{opt}}, 1\} \in \mathcal{O}(d^0) = \mathcal{O}(1).$$

As for the previous case, let $m$ be the same as in the proof of Theorem 1 for $\ell_\lambda = 0$. With probability at least $1 - cd^{-\beta\min\{\bar{\delta},1-\bar{\delta}\}}$, this $m$ again satisfies the conditions of Lemma 23, which yields

$$\mathbb{E}_\epsilon\left[\frac{1}{n}\sum_i(\hat{f}_{\lambda_{\mathrm{opt}}}(x_i)-y_i)^2\right] \leq \underbrace{\frac{\lambda_{\mathrm{opt}}^2\sigma^2}{n}\mathbf{Tr}\left(\mathbf{H}^{-2}\right)}_{:=T_3}+\underbrace{6\lambda_{\mathrm{opt}}^2\frac{r_2^2}{r_1^2}\frac{\|\mathbf{D}_{\leq m}^{-1}a\|^2}{n^2}}_{T_2}.$$

Furthermore, we apply the same steps with $l = 0$ as in the previous case (Equation (43)) to bound $T_2$:

$$T_2 \in \mathcal{O}\left(\mathbf{Risk}^2(\hat{f}_{\lambda_{\mathrm{opt}}})\right).$$

For $T_3$, we use the same bound on $\mathbf{Tr}\left(\mathbf{H}^{-2}\right)$ as in Equation (45) with the same probability as follows:

$$T_3 = \frac{\lambda_{\mathrm{opt}}^2\sigma^2}{n}\mathbf{Tr}\left(\mathbf{H}^{-2}\right) \leq \frac{\lambda_{\mathrm{opt}}^2\sigma^2}{n}\frac{n}{(c_1+\lambda_{\mathrm{opt}})^2}$$

$$= \sigma^2\frac{\lambda_{\mathrm{opt}}^2}{(c_1+\lambda_{\mathrm{opt}})^2}$$

$$\overset{(i)}{\leq} \sigma^2\frac{(c'')^2}{(c_1+c'')^2},$$

where $(i)$ follows for $d$ sufficiently large from $\lambda_{\mathrm{opt}} \in \mathcal{O}(1)$ with $c'' > 0$. Since $c_1 > 0$, we have $\frac{(c'')^2}{(c_1+c'')^2} < 1$. Hence, combining the bounds on $T_2$ and $T_3$, as well as collecting all probabilities, we get the desired result for this case. □

### D.3 Technical Lemmas

**Lemma 22** (Trace of the inverse). *Let $\mathbf{M}, \mathbf{M}_1, \mathbf{M}_2 \in \mathbb{R}^{n\times n}$ be symmetric positive semi-definite matrices with $\mathbf{M} = \mathbf{M}_1 + \mathbf{M}_2$. Furthermore, assume that $\mu_{\min}(\mathbf{M}_2) > 0$ and $\operatorname{rank}(\mathbf{M}_1) < n$. Then,*

$$\mathbf{Tr}(\mathbf{M}^{-2}) \geq \frac{n-\operatorname{rank}(\mathbf{M}_1)}{\|\mathbf{M}_2\|^2}.$$

*Proof.* First, we apply the identity

$$\mathbf{M}^{-1} = (\mathbf{M}_2 + \mathbf{M}_1)^{-1} = \mathbf{M}_2^{-1} - \underbrace{\mathbf{M}_2^{-1}\mathbf{M}_1(\mathbf{M}_2 + \mathbf{M}_1)^{-1}}_{:=\mathbf{A}},$$

which holds since $\mathbf{M}_2$, and thus $\mathbf{M}$, are full rank. Next, $\mathbf{A}$ is a product of matrices including $\mathbf{M}_1$; hence, the rank of $\mathbf{A}$ is bounded by $\mathrm{rank}(\mathbf{M}_1) < n$. Let now $\{v_1, \dots, v_{\mathrm{rank}(\mathbf{A})}\}$ be an orthonormal basis of $\mathrm{col}(\mathbf{A})$, and let $\{v_{\mathrm{rank}(\mathbf{A})+1}, \dots, v_n\}$ be an orthonormal basis of $\mathrm{col}(\mathbf{A})^\perp$. Thus, $\{v_1, \dots, v_n\}$ is an orthonormal basis of $\mathbb{R}^n$, and similarity invariance of the trace yields

$$\mathbf{Tr}(\mathbf{M}^{-2}) = \sum_{i=1}^n v_i^\mathsf{T} \mathbf{M}^{-2} v_i = \sum_{i=1}^{\mathrm{rank}(\mathbf{A})} v_i^\mathsf{T} \mathbf{M}^{-2} v_i + \sum_{i=\mathrm{rank}(\mathbf{A})+1}^n v_i^\mathsf{T} \mathbf{M}^{-2} v_i$$

$$\geq \sum_{i=\mathrm{rank}(\mathbf{A})+1}^n v_i^\mathsf{T} \left(\mathbf{M}_2^{-1} - \mathbf{A}\right)^2 v_i$$

$$\stackrel{(i)}{=} \sum_{i=\mathrm{rank}(\mathbf{A})+1}^n \left( v_i^\mathsf{T} \mathbf{M}_2^{-2} v_i - \underbrace{v_i^\mathsf{T} \mathbf{A} \, \mathbf{M}_2^{-1} v_i}_{=0} - v_i^\mathsf{T} \mathbf{M}_2^{-1} \underbrace{\mathbf{A} v_i}_{=0} + \underbrace{v_i^\mathsf{T} \mathbf{A} \mathbf{A} v_i}_{=0} \right)$$

$$= \sum_{i=\mathrm{rank}(\mathbf{A})+1}^n v_i^\mathsf{T} \mathbf{M}_2^{-2} v_i$$

$$\stackrel{(ii)}{\geq} (n - \mathrm{rank}(\mathbf{A}))\mu_{\min}\left(\mathbf{M}_2^{-2}\right) = (n - \mathrm{rank}(\mathbf{A}))\frac{1}{\|\mathbf{M}_2\|^2}$$

$$\stackrel{(iii)}{\geq} \frac{n - \mathrm{rank}(\mathbf{M}_1)}{\|\mathbf{M}_2\|^2},$$

where $(i)$ uses that, for all $i > \mathrm{rank}(\mathbf{A})$, $v_i$ is orthogonal to the column space of $\mathbf{A}$, and $\mathbf{A}$ is symmetric. Furthermore, $(ii)$ uses that all $v_i$ have norm 1, and $(iii)$ that $\mathrm{rank}(\mathbf{A}) \leq \mathrm{rank}(\mathbf{M}_1)$. $\qquad\square$

**Lemma 23** (Fixed-design training error). *In the setting of Theorem 3, let $m \in \mathbb{N}$ such that $r_1 > 0$, Equation (5) holds, and the ground truth satisfies $f^\star(x) = \sum_{k=1}^m a_k \psi_k(x)$. Then,*

$$\frac{\lambda^2 \sigma^2}{n} \mathbf{Tr}\left(\mathbf{H}^{-2}\right) \leq \mathbb{E}_\epsilon\left[ \frac{1}{n} \sum_i (\hat{f}_\lambda(x_i) - y_i)^2 \right] \leq \frac{\lambda^2 \sigma^2}{n} \mathbf{Tr}\left(\mathbf{H}^{-2}\right) + 6\lambda^2 \frac{r_2^2}{r_1^2} \frac{\|\mathbf{D}_{\leq m}^{-1} a\|^2}{n^2},$$

*where $\mathbf{H} \coloneqq \mathbf{K} + \lambda \mathbf{I}_n$.*

*Proof.* The kernel ridge regression estimator is

$$\hat{f}_\lambda(x^*) = \mathbf{y}^\mathsf{T} \mathbf{H}^{-1} \mathbf{k}(x^*),$$

where $\mathbf{y} \coloneqq [y_1, \dots, y_n]^\mathsf{T}$ with $y_i = f^\star(x_i) + \epsilon_i$, and $\mathbf{k}(x^*) \coloneqq [\mathcal{K}(x_1, x^*), \dots, \mathcal{K}(x_n, x^*)]^\mathsf{T}$. Thus, the estimator evaluated at $x_1, \dots, x_n$ is $\mathbf{K}\mathbf{H}^{-1}\mathbf{y}$, which yields the following training error:

$$\frac{1}{n} \sum_i (\hat{f}_\lambda(x_i) - y_i)^2 = \frac{1}{n} \|\left(\mathbf{I}_n - \mathbf{K}\mathbf{H}^{-1}\right)\mathbf{y}\|^2 \stackrel{(i)}{=} \frac{\lambda^2}{n} \|\mathbf{H}^{-1}\mathbf{y}\|^2,$$

where $(i)$ follows from

$$\mathbf{I}_n - \mathbf{K}\mathbf{H}^{-1} = \mathbf{I}_n - \mathbf{K}(\mathbf{K} + \lambda\mathbf{I}_n)^{-1} = \mathbf{I}_n - (\mathbf{K} + \lambda\mathbf{I}_n - \lambda\mathbf{I}_n)(\mathbf{K} + \lambda\mathbf{I}_n)^{-1} = \lambda(\mathbf{K} + \lambda\mathbf{I}_n)^{-1}.$$

Next, by the assumptions on the ground truth, we can write $\mathbf{y} = \boldsymbol{\epsilon} + \boldsymbol{\Psi}_{\leq m} a$. Thus, the expected training error with respect to the noise is

$$\mathbb{E}_\epsilon\left[ \frac{1}{n} \sum_i (\hat{f}_\lambda(x_i) - y_i)^2 \right] = \frac{\lambda^2}{n} \mathbb{E}_\epsilon\left[ (\boldsymbol{\epsilon} + \boldsymbol{\Psi}_{\leq m} a)^\mathsf{T} \mathbf{H}^{-2} (\boldsymbol{\epsilon} + \boldsymbol{\Psi}_{\leq m} a) \right]$$

$$= \underbrace{\frac{\lambda^2 \sigma^2}{n} \mathbf{Tr}\left(\mathbf{H}^{-2}\right)}_{:=T_1} + \underbrace{\frac{\lambda^2}{n} a^\mathsf{T} \boldsymbol{\Psi}_{\leq m}^\mathsf{T} \mathbf{H}^{-2} \boldsymbol{\Psi}_{\leq m} a}_{:=T_2}.$$

First, $T_2 > 0$ since $\mathbf{H}$ is positive semi-definite. Therefore, $T_1$ already yields the desired lower bound on the expected training error. For the upper bound, we bound $T_2$ as follows:

$$
\begin{aligned}
T_2 &= \frac{\lambda^2}{n}(\mathbf{D}_{\leq m}^{-1}a)^{\intercal}\mathbf{D}_{\leq m}\mathbf{\Psi}_{\leq m}^{\intercal}\mathbf{H}^{-2}\mathbf{\Psi}_{\leq m}\mathbf{D}_{\leq m}(\mathbf{D}_{\leq m}^{-1}a) \\
&\overset{(i)}{=} \frac{\lambda^2}{n}(\mathbf{D}_{\leq m}^{-1}a)^{\intercal}\left(\mathbf{D}_{\leq m}^{-1}+\mathbf{\Psi}_{\leq m}^{\intercal}\mathbf{H}_{>m}^{-1}\mathbf{\Psi}_{\leq m}\right)^{-1}\mathbf{\Psi}_{\leq m}^{\intercal}\mathbf{H}_{>m}^{-2}\mathbf{\Psi}_{\leq m} \\
&\qquad \left(\mathbf{D}_{\leq m}^{-1}+\mathbf{\Psi}_{\leq m}^{\intercal}\mathbf{H}_{>m}^{-1}\mathbf{\Psi}_{\leq m}\right)^{-1}(\mathbf{D}_{\leq m}^{-1}a) \\
&= \frac{\lambda^2}{n}\left\|\mathbf{H}_{>m}^{-1}\mathbf{\Psi}_{\leq m}\left(\mathbf{D}_{\leq m}^{-1}+\mathbf{\Psi}_{\leq m}^{\intercal}\mathbf{H}_{>m}^{-1}\mathbf{\Psi}_{\leq m}\right)^{-1}\mathbf{D}_{\leq m}^{-1}a\right\|^2 \\
&\leq \lambda^2\frac{\left\|\mathbf{\Psi}_{\leq m}^{\intercal}\mathbf{H}_{>m}^{-2}\mathbf{\Psi}_{\leq m}\right\|}{n}\left\|\left(\mathbf{D}_{\leq m}^{-1}+\mathbf{\Psi}_{\leq m}^{\intercal}\mathbf{H}_{>m}^{-1}\mathbf{\Psi}_{\leq m}\right)^{-1}\mathbf{D}_{\leq m}^{-1}a\right\|^2 \\
&\overset{(ii)}{\leq} \frac{1.5\lambda^2}{(\mu_{\min}(\mathbf{K}_{>m})+\lambda)^2}B_1. \\
&\overset{(iii)}{\leq} 6\lambda^2\frac{r_2^2\max\{\lambda,1\}^2}{(\mu_{\min}(\mathbf{K}_{>m})+\lambda)^2}\frac{\|\mathbf{D}_{\leq m}^{-1}a\|^2}{n^2} = 6\lambda^2\frac{r_2^2}{r_1^2}\frac{\|\mathbf{D}_{\leq m}^{-1}a\|^2}{n^2},
\end{aligned}
$$

where $\mathbf{H}_{>m} := \mathbf{K}_{>m} + \lambda\mathbf{I}_n$. Step $(i)$ follows from Lemma 6, step $(ii)$ uses Equation (5) and matches the term $B_1$ from the proof of Theorem 3, and step $(iii)$ applies the bound on $B_1$ achieved in Theorem 3. This upper-bounds the expected training error and thereby concludes the proof. $\square$

# E  EXPERIMENTAL DETAILS

This section describes our experimental setup and includes additional details. We provide the code to replicate all experiments and plots in https://github.com/michaelaerni/iclr23-InductiveBiasesHarmlessInterpolation.

## E.1  SETUP FOR FILTER SIZE EXPERIMENTS

The following describes the main filter size experiments presented in Section 4.1.

**Network architecture**  We use a special CNN architecture that amplifies the role of filter size as an inductive bias. Each model of the main filter size experiments in Figure 2 has the following architecture:

1. Convolutional layer with 128 filters of varying size and no padding

2. Global max pooling over the spatial feature dimensions

3. ReLU activation

4. Linear layer with 256 output features

5. ReLU activation

6. Linear layer with 1 output feature

All convolutional and linear layers use a bias term. Since we employ a single convolutional layer before global max pooling, the convolutional filter size directly determines the maximum size of an input patch that can influence the CNN's output. Note that this architecture reduces to an MLP if the filter size equals the input image size.

**Optimization procedure**  We use the same training procedure for all settings in Figure 2. Optimization minimizes the logistic loss for 300 epochs of mini-batch SGD with momentum 0.9 and batch size 100. We linearly increase the learning rate from $10^{-6}$ to a peak value of 0.2 during the first 50

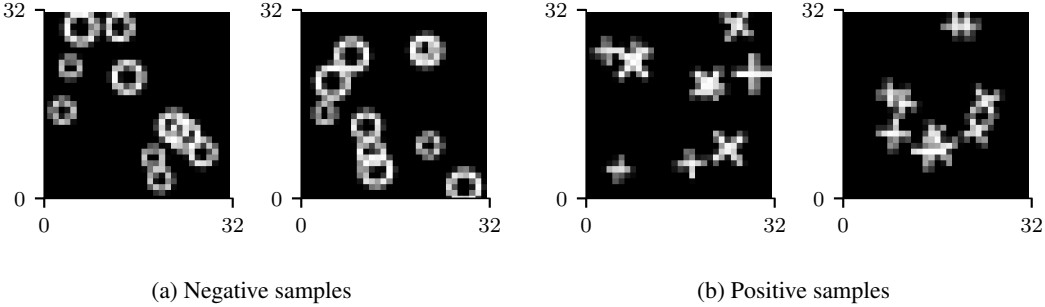

(a) Negative samples                    (b) Positive samples

Figure 4: Example synthetic images used in the filter size experiments.

epochs, and then reduce the learning rate according to an inverse square-root decay every 20 epochs. For peak learning rate $\gamma_0$, a decay rate $L$, the inverse square-root decay schedule at epoch $t \geq 0$ is

$$\frac{\gamma_0}{\sqrt{1 + \lfloor t/L \rfloor}}. \tag{46}$$

Learning rate warm-up helps to capture the early-stopped test error more precisely. Whenever possible, we use deterministic training algorithms, so that our results are as reproducible as possible. We selected all hyperparameters to minimize the training loss of the strongest inductive bias (filter size 5) on noisy training data, with the constraint that all other settings still converge and interpolate. Note that we do not use data augmentation, dropout, or weight decay.

**Evaluation**    We observed that all models achieved their minimum test error either at the beginning or very end of training. Hence, our experiments evaluate the test error every 2 epochs during the first 150 epochs, and every 10 epochs afterwards to save computation time. We use an oracle, that is, the true test error, to determine the optimal early stopping epoch in retrospective. The optimal early stopping training error is always over the entire training set (including potential noise) for a fixed model, not an average over mini-batches. To mitigate randomness in both the training data and optimization procedure, we average over multiple dataset and training seeds. More precisely, we sample 5 different pairs of training and test datasets. For each dataset, we fit 15 randomly initialized models per filter size on the same dataset, and calculate average metrics. The plots then display the mean and standard error over the 5 datasets.

**Dataset**    All filter size experiments use synthetic images. For a fixed seed, the experiments generate 200 training and 100k test images, both having an equal amount of positive and negative classes. Given a class, the sampling procedure iteratively scatters 10 shapes on a black $32 \times 32$ image. A single shape is either a circle (negative class) or a cross (positive class), has a uniformly random size in $[3, 5]$, and a uniformly random center such that all shapes end up completely inside the target image. We use a conceptual line width of $0.5$ pixels, but discretize the shapes into a grid. See Figure 4 for examples. A single dataset seed fully determines the training data, test data, and all scattered shapes.

**Noise model**    In the noisy case, we select $20\%$ of all training samples uniformly at random without replacement, and flip their label. The noise is deterministic per dataset seed and does not change between different optimization runs. Note that we never apply noise to the test data.

### E.2    SETUP FOR ROTATIONAL INVARIANCE EXPERIMENTS

The following describes the rotational invariance experiments presented in Section 4.2.

**Dataset**    We use the EuroSAT (Helber et al., 2018) training split and subsample it into 7680 raw training and 10k raw test samples in a stratified way. For a fixed number of rotations $k$, we generate a training dataset as follows:

1. In the noisy case, select a random $20\%$ subset of all training samples without replacement; for each, change the label to one of the other 9 classes uniformly at random.

2. For the $i$-th training sample ($i \in \{1, \ldots, 7680\}$):

   (a) Determine a random offset angle $\alpha_i$.

   (b) Rotate the original image by each angle in $\{\alpha_i + j \cdot (360°/k) \mid j = 0, \ldots, k-1\}$.

   (c) Crop each of the $k$ rotated $64 \times 64$ images to $44 \times 44$ so that no network sees black borders from image interpolation.

3. Concatenate all $k \times 7680$ samples into a single training dataset.

4. Shuffle this final dataset (at the beginning of training and every epoch).

To generate the actual test dataset, we apply a random rotation to each raw test sample independently, and crop the rotated images to the same size as the training samples. This procedure rotates every image exactly once, and uses random angle offsets to avoid distribution shift effects from image interpolation. Note that all random rotations are independent of the label noise and the number of training rotations. Hence, all experiments share the same test dataset. Furthermore, since we apply label noise before rotating images, all rotations of an image consistently share the same label.

**Network architecture** All experiments use a Wide Residual Network (Zagoruyko & Komodakis, 2016) with 16 layers, widen factor 6, and default PyTorch weight initialization. We chose the width and depth such that all networks are sufficiently overparameterized while still being manageable in terms of computational cost.

**Optimization procedure** We use the same training procedure for all settings in Figure 3. Optimization minimizes the softmax cross-entropy loss using mini-batch SGD with momentum 0.9 and batch size 128. Since the training set size grows in the number of rotations, all experiments fix the number of gradient updates to 144k. This corresponds to 200 epochs over a dataset with 12 rotations. Similar to the filter size experiments, we linearly increase the learning rate from zero to a peak value of 0.15 during the first 4800 steps, and then reduce the learning rate according to an inverse square-root decay (Equation (46)) every 960 steps. Whenever possible, we use deterministic training algorithms, so that our results are as reproducible as possible. We selected all hyperparameters to minimize the training loss of the strongest inductive bias (12 rotations) on noisy training data, with the constraint that all other settings still converge and interpolate. As for all experiments in this paper, we do not use additional data augmentation, dropout, or weight decay.

**Evaluation** Similar to the filter size experiments, we evaluate the test error more frequently during early training iterations: every 480 steps for the first 9600 steps, every 1920 steps afterwards. The experiments again use the actual test error to determine the best step for early-stopping, and calculate the corresponding training error over the entire training dataset, including all rotations and potential noise. Due to the larger training set size and increased computational costs, we only sample a single training and test dataset, and report the mean and standard error of all metrics over five training seeds.

### E.3 DIFFERENCE TO DOUBLE DESCENT

As mentioned in Section 4.1, our empirical observations resemble the double descent phenomenon. This subsection expands on the discussion and provides additional details about how this paper's phenomenon differs from double descent.

While all models in all experiments interpolate the training data, we observe that both noisy labels and stronger inductive biases increase the final training loss of an interpolating model: Smaller filter size results in a decreasing number of model parameters. Enforcing invariance to more rotations requires a model to interpolate more (correlated) training samples. Thus, in both cases, increasing inductive bias strength decreases a model's overparameterization in relation to the number of training samples — shifting the setting closer to the interpolation threshold.

We argue that our choice of architecture and hyperparameter tuning ensures that no model in any experiment is close to the corresponding interpolation threshold. If that is the case, then double descent predicts that increasing the number of model parameters has a negligible effect on whether regularization benefits generalization, and does therefore not explain our observations.

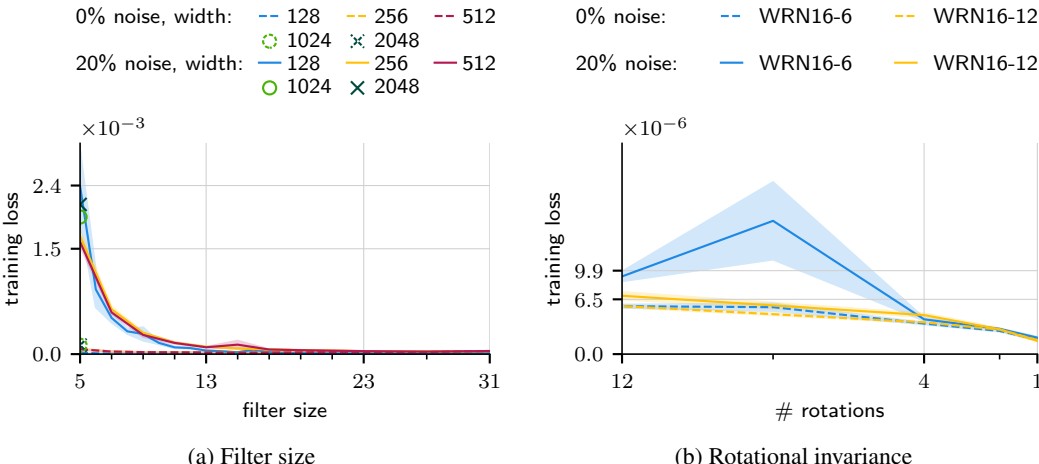

Figure 5: Training losses of the models in this paper's experiments as a function of (a) filter size and (b) the number of training set rotations. Models with a stronger inductive bias generally exhibit larger losses. However, in all instances, the numerical difference is small. Lines show the mean loss over 5 training set samples in (a) and 5 different optimization runs in (b), shaded areas the corresponding standard error.

In the following, we first describe how our hyperparameter and model selection procedure ensures that all models in all experiments are sufficiently overparameterized, so that double descent predicts negligible effects from increasing the number of parameters. Then, we provide additional experimental evidence that supports our argument: We repeat a subset of the experiments in Section 4 while upscaling the number of parameters in all models. For a fixed model scale and varying inductive bias, we observe that all phenomena in Section 4 persist. For a fixed inductive bias strength, we further see that the test error of interpolating models saturates at a value that matches our hypothesis. In particular, for strong inductive biases, the gap in test error between interpolating models and their optimally early-stopped version — harmful interpolation — persists.

**Hyperparameter tuning** We mitigate differences in model complexity for different inductive bias strengths by tuning all hyperparameters on worst-case settings, that is, maximum inductive bias with noisy training samples. To avoid optimizing on test data, we tune on dataset seeds and network initializations that differ from the ones used in actual experiments. Figure 5 displays the final training loss for all empirical settings in this paper. While models with a stronger inductive bias exhibit larger training losses, all values are close to zero, and the numerical difference is small. Finally, we want to stress again that this discussion is only about the training *loss*; all models in all experiments have zero training *error* and perfectly fit the corresponding training data.

**Increasing model complexity for varying filter size** As additional evidence, we repeat the main filter size experiments from Figure 2 in Section 4.1 using the same setup as before (see Appendix E.1), but increase the convolutional layer width to 256, 512, 1024, and 2048. For computational reasons, we evaluate a reduced number of filter sizes for widths 256 and 512, and only the smallest filter size 5 for widths 1024 and 2048. Since we found the original learning rate 0.2 to be too unstable for the larger model sizes, we use a decreased peak learning rate 0.13 for widths 256 and 512, and 0.1 for widths 1024 and 2048.

Figures 6a and 6b show the test errors for 20% and 0% training noise, respectively. With noisy training data (Figure 6a), larger interpolating models yield a slightly smaller test error, but the overall trends remain: the gap in test error between converged and optimally early-stopped models increases with inductive bias strength, and the phase transition between harmless and harmful interpolation persists. In particular, Figure 6a shows strong evidence that the number of model parameters does not influence our phenomenon: for example, models with filter size 5 (strong inductive bias) and width 512 (red) have more parameters than models with filter size 27 (weak inductive bias) and width 128 (blue). Nevertheless, models with filter size 5 benefit significantly from early stopping, while interpolation for models with filter size 27 is harmless. In the noiseless case (Figure 6b),

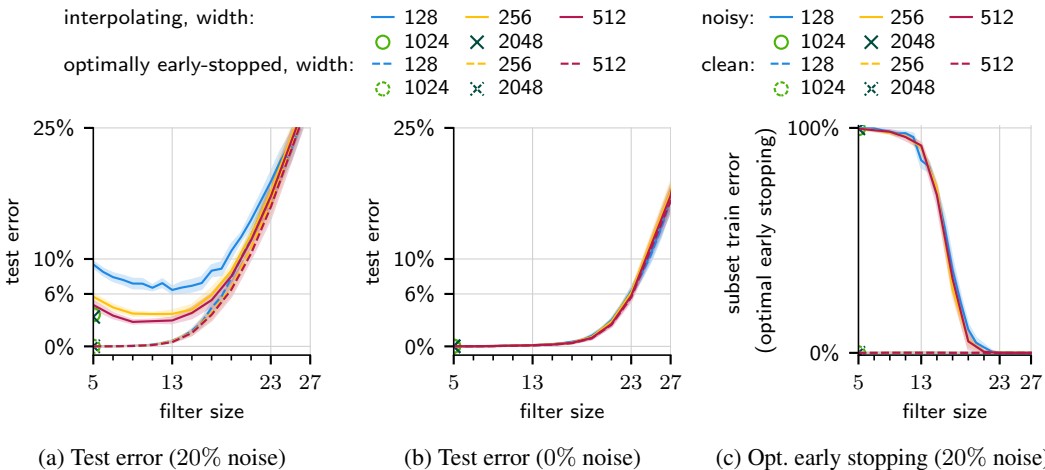

(a) Test error (20% noise)     (b) Test error (0% noise)     (c) Opt. early stopping (20% noise)

Figure 6: An increase in convolutional layer width by factors 2 to 16 does not significantly alter the behavior of (a) the test error when training on 20% label noise, (b) the test error when training on 0% label noise, and (c) the training error when using optimal early stopping on 20% label noise. Despite significantly larger model size, the phase transition between harmless and harmful interpolation persists. Lines show the mean over five random datasets, shaded areas the standard error.

increasing model complexity does neither harm nor improve generalization, and all models achieve their optimal performance after interpolating the entire training dataset. Similarly, Figure 6c reveals that the fraction of training noise that optimally early-stopped models fit stays the same for larger models. Finally, for a fixed inductive bias strength, the test errors saturate as model size increases, making a different trend for models with more than 2048 filters unlikely. To increase legibility, we present the numerical results for the largest two filter sizes in Table 1.

Table 1: Test errors for filter size 5 (strongest inductive bias) and very large width under 20% training noise.

|                          | width 1024 | width 2048 |
| --- | --- | --- |
| early-stopped test error | 0.0062%    | 0.0049%    |
| interpolating test error | 3.6251%    | 3.3664%    |
| # parameters             | 289281     | 578049     |

**Increasing model capacity for varying rotational invariance**     For completeness, we also repeat the rotation invariance experiments from Figure 3 in Section 4.2 with twice as wide Wide Residual Networks on a reduced number of rotations. More precisely, we increase the network widen-factor from 6 to 12, and otherwise use the same setting as the main experiments (see Appendix E.2). Note that this corresponds to a parameter increase from around 6 million to around 24 million parameters.

The results in Figure 7 provide additional evidence that our phenomenon is distinct from double descent: both the test error (Figures 7a and 7b) and fraction of fitted noise under optimal early stopping (Figure 7c) exhibit the same trend, despite the significant difference in number of parameters.

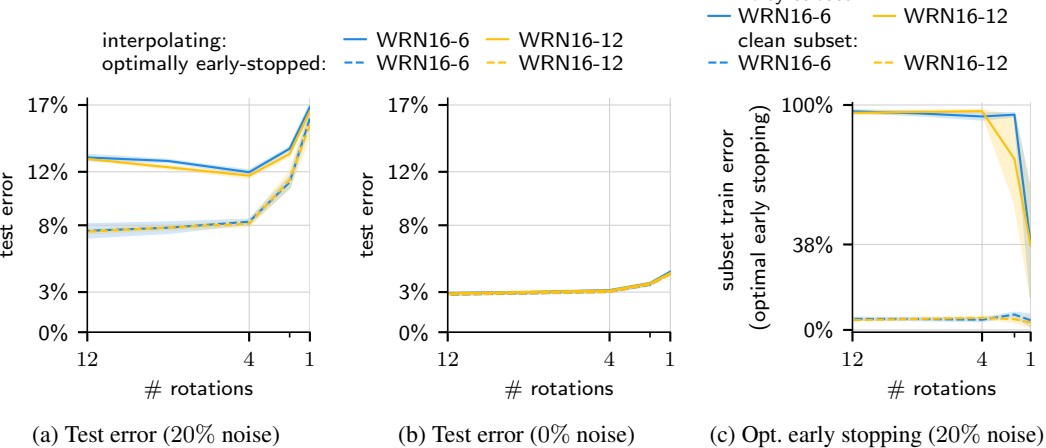

(a) Test error (20% noise)  (b) Test error (0% noise)  (c) Opt. early stopping (20% noise)

Figure 7: Doubling Wide Residual Network width does not significantly alter the behavior of (a) the test error when training on 20% label noise, (b) the test error when training on 0% label noise, and (c) the training error when using optimal early stopping on 20% label noise. Despite significantly larger model size, the phase transition between harmless and harmful interpolation persists. Lines show the mean over five random network initializations, shaded areas the standard error.

