# OpenReview forum: "Strong inductive biases provably prevent harmless interpolation"
_ICLR.cc/2023/Conference — ICLR 2023 poster_

### Official Review · Reviewer_KVHo · 2022-10-23

**Confidence:** 3
**Correctness:** 3
**Technical Novelty And Significance:** 4
**Empirical Novelty And Significance:** 3
**Recommendation:** 8

**Clarity, Quality, Novelty And Reproducibility:**

The paper is well-written and is of a high quality. Both the theoretical and experimental results are insightful and novel in my opinion. Also, the authors describe the experimental settings in sufficient length for reproducibility.

**Strength And Weaknesses:**

This is a well-polished paper that studies the important topic of generalization in modern neural networks. The authors present an interesting insight in terms of inductive bias that is quite novel and is supported both theoretically and experimentally. The experiments are also well-executed.

The primary weakness is that the settings studied theoretically and experimentally are contrived but I don’t think this is a major issue given the insight this paper provides. It would have been valuable to figure out how to quantify inductive bias in typical training settings, e.g. in standard benchmark image datasets with convolutional neural networks, but this is quite challenging given the lack of knowledge about the ground truth.

Another weakness is already acknowledged by the authors. The main theoretical results make strong assumptions about the distribution of the data that cannot be easily relaxed.

Finally, while the authors prove that models with weak inductive bias interpolate some of the noise, this does not mean that interpolating noise is necessary since it is not clear whether the optimal bias is below or above the threshold $\beta^\star$. Please correct me if I have miss-interpreted this part.

I have two minor suggestions:

- Please clarify the meaning of the $\Theta$ notation for quantities that are bounded from below by zero and go to zero such as writing $\sigma^2 = \Theta(d^{-\ell_\sigma})$. I personally read this as $O$ notation but if there is more than that, please clarify in the paper.
- The statement of the theorems should be self-contained. Currently, there are hidden assumptions on the ground truth and eigenfunctions in the proof that are not stated in the main theorems. Please mention them explicitly.

Two typos:

- Line 32: “interpolators interpolators”
- Line 208: “double descentpredicts”


**Summary Of The Paper:**

The paper studies the impact of the strength of inductive bias on the generalization performance of the trained model. First, the authors prove that in kernel learning with convolutional kernels, a phase transition occurs in the noisy data setting. Specifically, when the level of inductive bias exceeds a particular threshold, interpolation is harmful. But, it is harmless when the inductive bias is sufficiently weak. For the noiseless data setting, no similar transition occurs and interpolation is provably harmless. In addition, the authors prove that in the noisy data setting, the model with the weak inductive bias fits some of the noise (variance on the training sample is below the data variance). However, the latter statement does not really prove that interpolating noise is necessary since nothing is mentioned about the optimal inductive bias. Finally, the authors present two experiments, which suggest that such conclusions seem to hold in deep neural networks as well.


**Summary Of The Review:**

This is a nice, insightful paper that is well-executed. However, I haven't verified the correctness of the proofs.

---

> ### Author Response · Authors · 2022-11-18
> **Response to Reviewer KVHo**
>
> We would like to thank Reviewer KVHo for appreciating our work and the profound comments, which we would like to address in detail.
>
> **Comment on the summary and weaknesses, part 1:**
> >“In addition, the authors prove that in the noisy data setting, the model with the weak inductive bias fits some of the noise (variance on the training sample is below the data variance). However, the latter statement does not really prove that interpolating noise is necessary since nothing is mentioned about the optimal inductive bias.“
>
> >“Finally, while the authors prove that models with weak inductive bias interpolate some of the noise, this does not mean that interpolating noise is necessary since it is not clear whether the optimal bias is below or above the threshold $\\beta^\\star$. Please correct me if I have miss-interpreted this part.”
>
> We thank the reviewer for the detailed points. First, we would like to clarify the statement of Theorem 2: we prove that for **fixed, weak inductive biases**, interpolating noise is necessary to achieve the smallest risk. We do not make theoretical statements about an “optimal inductive bias”. That being said, in the examples in the paper, the smallest risk is achieved at a strong inductive bias where regularization helps (that is, not fitting noise yields a smaller risk than interpolation).
> We further tried to resolve this potential point of ambiguity by modifying the second bullet point in the contribution list as follows:
> > “we further show that, for weak inductive biases, not only is interpolation harmless but partially fitting the observation noise is in fact necessary (Theorem 2).”
>
> Finally, we might have misinterpreted some of the reviewer’s points. Thus, we would like to kindly ask the reviewer whether we addressed all of their concerns, and will gladly answer any follow-up questions.
>
> **Comment on the weaknesses, part 2:**
> > “Another weakness is already acknowledged by the authors. The main theoretical results make strong assumptions about the distribution of the data that cannot be easily relaxed.”
>
> We thank the reviewer for noticing the challenges of our theoretical analysis, and would like to provide some further context. The main purpose of our theoretical results is to convey an intuition for the relation between the strength of inductive bias and the harmless interpolation phenomenon. We then show that this intuition transfers to more complex models via experiments. Hence, a simple theoretical setting allows us to gain clear insights, while we cover more sophisticated settings empirically.
>
> Nevertheless, generalizing the presented theoretical results to other settings is an exciting avenue for future work. The primary restriction of our proof is that it requires access to the kernel’s eigenfunctions and eigenvalues. This requirement stems from the high dimensionality of the input data, and stands in stark contrast to results for low-dimensional kernel learning. Deriving a general theoretical framework is hence challenging, and might require a radically different approach.
>
> **Comment on the first suggestion:**
> > “Please clarify the meaning of the $\Theta$ notation for quantities that are bounded from below by zero and go to zero such as writing $\sigma^2 = \Theta(d^{-\ell_\sigma})$. I personally read this as O notation but if there is more than that, please clarify in the paper.”
>
> The $\Theta$-notation is part of the Bachmann–Landau notation. We say that $f(x) \in \Theta(g(x))$ if there exists constants $c_1, c_2, x_0 > 0$ such that $c_1 g(x) \leq f(x) \leq c_2 g(x) $ for all $x \geq x_0$. Note that we allow $c_1$ and $c_2$ to depend on $\ell$ and $\beta$, but not on any of the other quantities in Theorem 1. Since the paper takes a non-asymptotic viewpoint, we intuitively consider the $\Theta$-notation as restricting quantities to an interval as long as $x$ (typically $d$) is sufficiently large. In particular, the notation allows a quantity to both grow and decay. We thank the reviewer for making us aware of our notation’s unclarity, and updated Footnote 2 in the revised version of the paper.
>
> **Comment on the second suggestion:**
> >“The statement of the theorems should be self-contained. Currently, there are hidden assumptions on the ground truth and eigenfunctions in the proof that are not stated in the main theorems. Please mention them explicitly.”
>
> We agree with the reviewer’s comment, and implement the suggestion in the updated version of the paper.

---

### Official Review · Reviewer_NQJb · 2022-10-24

**Confidence:** 4
**Correctness:** 4
**Technical Novelty And Significance:** 3
**Empirical Novelty And Significance:** Not applicable
**Recommendation:** 8

**Clarity, Quality, Novelty And Reproducibility:**

Clarity:
- Overall the paper is well-written and easy-to-follow.

Quality:
- The quality of this paper is overall good. Theorems and assumptions are clearly stated. Proof sketch is provided in the main text with full proof in the appendix. (I didn’t check the full proof).

Novelty:
- The harmless interpolation/benign overfitting is an active research direction and studying the impact of such biases is an important problem.
- The results are novel to my knowledge, which shows the impact of implicit bias on the performance in the noisy setting.

Reproducibility:
- This is a theoretical work with several experiments. Full proofs are provided in the appendix and codes are provided in the supplement material, but I didn’t check/run them.


**Strength And Weaknesses:**

Strength:
- The paper is overall clearly written and easy-to-follow.
- Understanding the generalization properties of interpolated models is an important and interesting direction of the theoretical machine learning, and this paper gives interesting results towards this direction.
- The results are interesting and novel to my knowledge, which show the relation between filter size and the performance of optimal-regularized and interpolated models.
- The results are clearly stated with assumptions. Proof sketch is provided in the main text with full proof in the appendix.

Weaknesses:
- The proof techniques may not be easy to generalize to other settings beyond kernel and linear settings.
- It would be great if authors could provide a formula (or the order) of some quantities in the main text or appendix, such as $\beta^*$ and $\lambda_{opt}$.

Minor:
- line 208: missing a space between descent and predicts


**Summary Of The Paper:**

This paper studies the harmless interpolation in the setting of high-dimensional kernel regression with convolutional kernels. Here, harmless interpolation means that the performance (measured by test error) gap between the interpolated models and optimal-regularized model is small. The authors gave the non-asymptotic bounds of test error (in terms of bias-variance decomposition) for both regularized and interpolated models. The authors further showed the impact of filter size of convolutional kernels: harmless interpolation will not happen for small filter size (strong inductive bias), but will occur for large filter size. Also, it is shown that for large filter size, the optimal-regularized models have to fit part of the noise, i.e., training error smaller than noise level. Experiments are provided to justify the theoretical results.

**Summary Of The Review:**

In summary, the paper studied the generalization error of optimal-regularized and interpolated models in high-dimensional kernel regression setting. The results showed impact of filter size to the test error, which appears to be new and interesting. Therefore, I’m leaning towards accept.

---

> ### Author Response · Authors · 2022-11-18
> **Response to Reviewer NQJb**
>
> We would like to thank Reviewer NQJb for appreciating our work. We now address some comments in detail.
>
> **Comment on the weaknesses, part 1:**
> > “The proof techniques may not be easy to generalize to other settings beyond kernel and linear settings.”
>
> We agree with the reviewer that the theoretic setting is restrictive, but would like to provide some additional context. The main purpose of our theoretical results is to convey an intuition for the relation between the strength of inductive bias and the harmless interpolation phenomenon. We then show that this intuition transfers to more complex models via experiments. Hence, a simple theoretical setting allows us to gain clear insights, while we cover more sophisticated settings empirically.
>
> Nevertheless, generalizing the presented theoretical results to other settings is an exciting avenue for future work. The primary restriction of our proof is that it requires access to the kernel’s eigenfunctions and eigenvalues. This requirement stems from the high dimensionality of the input data, and stands in stark contrast to results for low-dimensional kernel learning. Deriving a general theoretical framework is hence challenging, and might require a radically different approach.
>
> **Comment on the weaknesses, part 2:**
>  > “It would be great if authors could provide a formula (or the order) of some quantities in the main text or appendix, such as $\beta^*$ and $\lambda_{opt}$.”
>
> We thank the reviewer for this suggestion, which we address in the updated version of the paper (Appendix D.1, paragraph “Finding optimal parameters”). In short, multiple descent prevents insightful closed-form solutions for $\beta^*$ and the rate of $\lambda_{opt}$. For Figure 1, we use simple numerical optimization: First, we find the optimal regularization rate $\ell_{\lambda_{opt}}$ for a fixed inductive bias as the point where the bias and variance rates match. Since their signed difference is a monotonic function of $\ell_{\lambda_{opt}}$, we can do so using simple binary search. Second, we determine $\beta^*$ in Figure 1 via simple grid search. The submitted code contains an example in `theory.py`.

---

### Official Review · Reviewer_fKcA · 2022-10-24

**Confidence:** 4
**Correctness:** 4
**Technical Novelty And Significance:** 3
**Empirical Novelty And Significance:** 3
**Recommendation:** 8

**Clarity, Quality, Novelty And Reproducibility:**

- How is the optimal regularization strength found in the estimator of Thm 2? Is this an optimization that is performed exactly?
- There are no results on the bias and variance of the neural network trained in Sec 4. It should be possible to estimate this from experiments. Do the authors expect to see similar multiple descent curve?

**Strength And Weaknesses:**

**Strengths**
- The paper studies an important problem of benign overfitting and finds a new condition (strength of inductive bias) that it is predicated on.
- The paper is easy to follow and well written. It states the main claims clearly and provides evidence for them.

**Weaknesses**
- The difference between the strength of inductive biases and parameter count is not fully resolved in the neural network experiments. Is it possible to just control the number of parameters by adjusting the width to compensate for change in filter size?

**Summary Of The Paper:**

This paper studies how the strength of inductive biases affect harmless interpolation or benign overfitting, where models can generalize well despite interpolating a (possibly noisy) test set. To study this question, the authors adjust the filter size in convolutional kernels. Smaller filter sizes encode for functions that depend nonlinearly only on local neighborhoods of the input features, whereas this bias is weakened for larger filter sizes. The paper shows theoretically that for high-dimensional kernel regression with convolutional kernels, there is a phase transition between harmless and harmful interpolation in filter size (Theorem 1). The authors show that interpolation is harmless for weak inductive biases, and that for strong inductive biases, they show interpolation can be harmful. Further, they show fitting noise is necessary for strong inductive biases to achieve optimal performance. These theoretical results are supported by experiments in neural networks, where the same phenomenology is observed.

**Summary Of The Review:**

The paper makes a nice contribution and is of interest to the ICLR community. It is clearly written with nice figures. From my read of the paper's main text, the theorems seem correct.

---

> ### Author Response · Authors · 2022-11-18
> **Response to Reviewer fKcA - Part 1**
>
>
> We would like to thank Reviewer fKcA for appreciating our work and their thoughtful comments, some of which we now address in detail.
>
> **Comment on the summary:**
> > "The authors show that interpolation is harmless for weak inductive biases, and that for strong inductive biases, they show interpolation can be harmful. Further, they show fitting noise is necessary for **strong inductive biases** to achieve optimal performance."
>
> While the first sentence precisely summarizes the first main contribution of this paper (Theorem 1), we suspect that the second sentence contains a typo: could it be that “strong” was meant to be “weak”? Just to be safe, we would like to restate the second contribution of our paper: intuitively, Theorem 2 states that, for **weak inductive biases** and noisy training data, fitting noise is harmless and to some extent even necessary to achieve optimal generalization performance.
>
>
> **Comment on the weaknesses:**
> > “The difference between the strength of inductive biases and parameter count is not fully resolved in the neural network experiments. Is it possible to just control the number of parameters by adjusting the width to compensate for change in filter size?”
>
> We thank the reviewer for this interesting question, which we think is worth discussing in more detail. Based on our theoretical findings for kernel regression, the hypothesis we would like to test in the experimental section is that inductive bias strength is a second key ingredient that determines whether interpolation is harmless or harmful. In particular, we hypothesize that inductive bias behaves differently from the degree of overparameterization—which previous literature already identified to act as an implicit regularization mechanism by decreasing the variance. Hence, we ideally want to fix a degree of overparameterization and vary only the strength of the inductive bias.
>
> **What is a fixed degree of overparameterization?** Most previous literature considers filter size a fixed design choice, and uses width to measure the degree of overparameterization, coinciding with the # of parameters. However, for variable filter size, the reviewer noticed correctly that inductive bias, width, and the number of model parameters are inherently entangled: a stronger inductive bias (smaller filter size) corresponds to a smaller number of parameters for a fixed width. In this case, there are two possibilities to mimic a fixed degree of overparameterization: 1) consider width large enough so that the difference in parameter count between the weakest and strongest inductive bias has negligible effects on generalization properties, or 2) fix the number of parameters, as suggested by the reviewer.
>
> **Discussion of approach 1):** Our choices of widths cover case 1), supported by the additional experiments in Appendix E.3. For example, we would like to draw conclusions (such as gap between interpolating and optimally early-stopped estimator) when comparing filter size 5 vs. filter size 27 for width 256. The difference in # of parameters between filter size 5 and 27 for width 256 is comparable (or smaller) than the difference in # parameters for filter size 5 between widths 256 and 1024 or even 2048 (new experiments). However, the change in filter size has a much *larger* impact on the error gap than the change in network width, despite a similar or even smaller change in the # parameters. Therefore, the harmful interpolation phenomenon for strong inductive biases cannot be entirely explained by the parameter count difference.
>
> **New experiments:** We updated Section 4.1 and Appendix E.3 to clarify the reasoning behind our approach, and added two additional experiments: First, we repeat a subset of the experiments in Section 4.2 (rotational invariance) with increased network width factor (Figure 7). Second, we extended the additional filter size experiments (Figure 6) with CNN widths 1024 and 2048, with filter size 5 only for computational reasons (see Table 1). In all cases, we observe that increasing the number of parameters does not significantly influence the test error, while inductive bias strength does.
>
> **Discussion of approach 2) fixed # parameters:** Lastly, we would like to highlight some challenges we faced when exploring models with a fixed # parameters: First, exactly fixing the number of model parameters requires infeasible large networks. Keeping the number of parameters just approximately constant introduces large differences in network architecture. Those differences mean that no single optimization procedure works for all filter sizes (inductive bias strengths). Hence, different training procedures might confound any findings in our experiments. If the reviewer has a good suggestion to resolve these challenges, we would be happy to run the corresponding experiments.

---

> ### Author Response · Authors · 2022-11-18
> **Response to Reviewer fKcA - Part 2**
>
> **First comment on clarity:**
> > “How is the optimal regularization strength found in the estimator of Thm 2? Is this an optimization that is performed exactly?”
>
> We thank the reviewer for this profound question. Formally, for a fixed inductive bias, we define $\\lambda\_{opt}$ as the regularization strength that minimizes the population risk, that is, $\\lambda\_{opt} \\in argmin\_\\lambda \\mathbf{Risk}(\\hat{f}\_{\\lambda})$. While we cannot explicitly determine $\\lambda\_{opt}$ itself, the proof of Theorem 2 only requires a rate. The revised version of our paper contains a more thorough discussion in Appendix D.1, particularly in the paragraph “Finding the optimal parameters”.
>
> **Second comment on clarity:**
> > “There are no results on the bias and variance of the neural network trained in Sec 4. It should be possible to estimate this from experiments.”
>
> We thank the reviewer for this input. In contrast to regression, bias and variance cannot be defined straight-forwardly for classification (0-1 error), which is why we did not estimate those quantities. However, our experiments hint that neural networks with strong inductive biases suffer more from noise than networks with weak inductive biases. In Figures 2b and 3b in particular, fitting samples with perturbed labels hurts the test error for strong inductive biases, but not for weak inductive biases. Ultimately, we can still study the transferability of our hypothesis (“harmful interpolation for strong inductive biases”) by comparing the error curves of the optimally early stopped vs. interpolating solution, as we did in Figures 2 and 3.
>
> > “Do the authors expect to see similar multiple descent curve?”
>
> This is again an interesting question. Multiple descent is a very particular artifact that arises specifically from kernel regression in a high-dimensional setting. In particular, our theoretical analysis shows that multiple descent is a consequence of the kernel’s eigenvalue structure. Hence, we do not have much reason to believe that the phenomenon would carry over to neural network training beyond the kernel regime. However, we think the reviewer poses an intriguing question to investigate in future work.

---

### Author Response · Authors · 2022-11-18
**To all reviewers**

We thank all the reviewers including the area chair for their time to evaluate our paper and their valuable feedback.

---

### Decision · Program_Chairs · 2023-01-20

**Decision:**

Accept: poster

**Justification For Why Not Higher Score:**

The topic stated in the paper is very important and popular in the field of theoretical machine learning (including "overparametrization phenomenon" and  understanding of deep neural network generalisation). The results obtained in the paper (formal for a narrow setup and empirical for more realistic scenarios) are certainly significant and interesting, particularly to the aforementioned subfields. However, given the formal argument of the paper focuses on a rather artificial and narrow setup, I don't feel confident that at the moment the consequences of these results are broad enough to interest a wider audience.

**Justification For Why Not Lower Score:**

All the reviewers (and myself) agree that the paper provides a significant contribution to the field of theoretical machine learning, particularly to the research around overparametrization and benign overfitting.

**Metareview: Summary, Strengths And Weaknesses:**

The paper investigates the interplay between the strength of algorithm's inductive bias and the benign overfitting phenomenon. The "benign overfitting", "harmless interpolation", and "overparametrization" phenomena have recently gained a lot of attention in the field of theoretical machine learning, including the research focused on understanding generalization of deep neural networks. I believe that advances in these directions are very interesting and important for the field.

The authors approach the question by studying a very clear and controlled setup of high-dimensional (regularized) kernel regression, where the ground truth function is known and the inductive bias of the algorithm (the filter size of the convolutional kernel) has a clear interpretation. The paper provides (Theorem 1) exact rates for the bias and variance of two algorithms --- the min-norm (un-regularized) and ridge-regularized RKHS regressions. Comparing the rates, the authors clearly demonstrate that there is a critical size of the filter (or "critical strength of the inductive bias) that separates the benign and harmful overfitting: for the stronger inductive bias, the unregularized model that is allowed to interpolate the training data performs worse than the (optimally) regularized model; for the weaker inductive bias, both models have the same performance.

Even though the formal analytical argument holds only for the particular (and artificial) setup studied by the authors, all the reviewers agree that this does not diminish the significance of the contribution. More realistic general setups would be too difficult to deal with and by considering a simple, clear, controlled setup the authors managed to make their claim very accessible. Moreover, the authors provide experimental evidence (Section 4) suggesting that similar effects may take place in more realistic scenarios involving convolutional neural networks. Overall the reviewers agree that it is a very interesting contribution that has a potential to help future research in this direction.

All the reviewers highlighted that "the paper is easy to follow and well written" and that "it states the main claims clearly and provides evidence for them." They agreed that "both the theoretical and experimental results are insightful and novel".

I recommend the acceptance. However, I would ask the authors to carefully account for the feedback, provided by the reviewers. Doing so will further improve the readability of the paper.

**Note From Pc:**

if the above contains the word "oral" or "spotlight" please see: "oral" presentation means -> notable-top-5% and "spotlight" means -> notable-top-25%. As stated in our emails, we are disassociating presentation type from AC recommendations

**Summary Of Ac-Reviewer Meeting:**

N/A